# Pooled multicolour tagging for visualizing subcellular protein dynamics

Andreas Reicher[1,2], Jiří Reiniš[1,2], Maria Ciobanu[1], Pavel Růžička[1], Monika Malik[1], Marton Siklos[1], Victoria Kartysh[1], Tatjana Tomek[1], Anna Koren[1], André F. Rendeiro[1] & Stefan Kubicek [1] ✉

Imaging-based methods are widely used for studying the subcellular localization of proteins in living cells. While routine for individual proteins, global monitoring of protein dynamics following perturbation typically relies on arrayed panels of fluorescently tagged cell lines, limiting throughput and scalability. Here, we describe a strategy that combines high-throughput microscopy, computer vision and machine learning to detect perturbation-induced changes in multicolour tagged visual proteomics cell (vpCell) pools. We use genome-wide and cancer-focused intron-targeting sgRNA libraries to generate vpCell pools and a large, arrayed collection of clones each expressing two different endogenously tagged fluorescent proteins. Individual clones can be identified in vpCell pools by image analysis using the localization patterns and expression level of the tagged proteins as visual barcodes, enabling simultaneous live-cell monitoring of large sets of proteins. To demonstrate broad applicability and scale, we test the effects of antiproliferative compounds on a pool with cancer-related proteins, on which we identify widespread protein localization changes and new inhibitors of the nuclear import/export machinery. The time-resolved characterization of changes in subcellular localization and abundance of proteins upon perturbation in a pooled format highlights the power of the vpCell approach for drug discovery and mechanism-of-action studies.

Knowing the subcellular localizations of proteins is essential for understanding cellular biology[1,2]. Up to 50% of the human proteome resides in multiple compartments, membrane-less organelles or other subcellular structures and protein localization can be very dynamic and change across different cell states or in response to perturbations, thereby impacting protein function[2].

For analysing protein localizations and dynamics, various methods based on mass spectrometry (MS) or imaging are available, offering different degrees of proteome coverage, achievable spatiotemporal resolution and sample throughput[1,3]. While biochemical fractionation[4,5] or proximity labelling[6,7] followed by MS analysis can provide proteome-wide coverage, the complex sample processing and input material requirements limit throughput. On the other hand, imaging-based methods such as immunocytochemistry or endogenous tagging with fluorescent proteins provide inherent single-cell resolution and can be conducted in a multiwell plate format[8–17]. Particularly, fluorescent tagging approaches are well-suited for live-cell imaging at multiple time points to temporally resolve dynamic localization changes. Recently, large scale efforts such as the Human Protein Atlas[2] or the OpenCell[18] project have demonstrated how imaging-based methods can be scaled in an arrayed format for mapping the localizations of hundreds to thousands of proteins using either antibodies or large

[1]CeMM Research Center for Molecular Medicine of the Austrian Academy of Sciences, Vienna, Austria. [2]These authors contributed equally: Andreas Reicher, Jiří Reiniš. ✉e-mail: skubicek@cemm.oeaw.ac.at

cell line collections expressing endogenously tagged proteins. While these efforts resulted in invaluable resources describing subcellular protein localizations in steady state, screening these large, arrayed collections in many conditions remains challenging. Previous work with arrayed collections of fluorescently tagged lines in the yeast ORF green fluorescent protein (GFP) collection and subsets of human genes has been limited to individual or few selected perturbations[11,12,14], with challenges in scalability precluding wider application to systematically analyse subcellular protein dynamics.

We had previously used GFP-tagged cell pools for monitoring hundreds of proteins in a single well, and in situ sequencing of the intron-tag specifying single guide RNAs (sgRNAs) enabled us to determine the identity of the tagged protein in each cell[19]. Building on this method and drastically improving its throughput by multicolour tagging and computational clone identification, we here develop an approach for studying protein localization changes in response to genetic or environmental perturbations. We present a platform for the generation of cell pools expressing different fluorescently tagged proteins in every cell that enable simultaneous monitoring of subcellular localizations and abundance for large protein sets in live cells (Fig. 1a). Our strategy involves the following steps: (i) Pooled multicolour tagging with genome-wide or focused intron-targeting sgRNA libraries to generate a vpCell pool containing five complementary fluorescent tags on two endogenous proteins, two structural markers and a diversity channel. (ii) Using computer vision to learn clone identities from the localization patterns of tagged proteins in every cell. (iii) Assembling a custom cell pool of clones representing proteins of interest to monitor. (iv) Imaging this cell pool before and after perturbation. (v) Image-based identification of clones and quantification of localization changes in response to perturbations in a pooled format. As a proof of concept, we screen one vpCell pool covering 61 cancer-related proteins against a compound library of 1,059 antiproliferative compounds and observe multiple specific drug–protein interactions. We identify a previously uncharacterized compound as an inhibitor of exportin 1 (XPO1), demonstrating how these cell pools can be used for monitoring protein localizations at scale. Additionally, we show that vpCell pools can be used for the rapid generation of large, arrayed clone collections and we isolate and image clonal cell lines covering 1,158 endogenously tagged proteins. This collection can be interactively browsed in our web atlas of intron-tagged proteins at https://vpcells.cemm.at/.

## Results

### Genome-scale pooled protein tagging
For the generation of cell pools, we used a pooled intron-targeting strategy that we developed previously for the tagging of metabolic enzymes[19]. The strategy relies on the CRISPR/Cas9-mediated generation of double-strand breaks in the introns of genes, followed by the integration of a synthetic GFP-containing exon flanked by splice-donor and splice-acceptor sites[20]. The use of an sgRNA library enables the generation of a cell pool where every cell expresses a different tagged protein, with the sgRNA specifying the identity of the tagged protein. Here, we designed a genome-wide intron-targeting sgRNA library of 90,657 sgRNAs targeting 73,817 introns of 14,158 genes (Supplementary Table 1). We transduced HEK293T cells with that library and after co-transfection with minicircle donor DNA and a Cas9-expressing plasmid and following NHEJ-mediated integration of a synthetic GFP-containing exon at the intronic target sites, we sorted GFP+ cells. In the resulting cell pool, every cell contains a different protein endogenously tagged with GFP, as specified by the intron-targeting sgRNA that is expressed in that cell (Fig. 1b). Using minicircle DNA[17,21] instead of a GFP-containing plasmid as a DNA donor improved the tagging efficiency twofold. Notably, it decreased the chances of integrating additional sequences such as the plasmid backbone at the target sites, which we had previously observed to potentially effect the expression of the tagged gene (Extended Data Fig. 1a). The overall tagging efficiency in cells transduced with the library was approximately 10–20-fold lower compared with positive control sgRNAs (Extended Data Fig. 1b). This is expected, as not every sgRNA in the library results in protein tagging and there is a selection for proteins, tag positions and sgRNAs with higher tagging efficiencies. To determine which proteins were successfully tagged, we performed amplicon sequencing to analyse the sgRNA abundance in the pool of GFP-tagged cells. In line with the notion that only a small fraction of sgRNAs leads to successful protein tagging, we observed an enrichment of approximately 5% of all sgRNAs in the library (Fig. 1c). We termed these top 5,113 sgRNAs targeting 4,553 introns of 2,384 genes as high-efficiency sgRNAs (Supplementary Table 2). As a control, our library contained 1,000 non-targeting sgRNAs that should not lead to GFP tagging, and we found these to be depleted from the high-efficiency sgRNA pool compared with their abundance in the whole library (0.10% versus 1.11%; Extended Data Fig. 1c).

When testing the most abundant sgRNAs in the GFP pool individually, we could confirm very high tagging efficiencies up to 18.7%. Tagging the same genes with sgRNAs that were not highly enriched in the cell pool was less effective, showing that there was indeed a selection for sgRNAs and presumably tag positions within the protein that result in higher tagging efficiencies (Fig. 1d and Extended Data Fig. 1d). Similar to our previous observation[19], genes targeted by high-efficiency sgRNAs have higher average expression[18], indicating that a certain expression level is required for detection and sorting of GFP-tagged cells (Fig. 1e and Supplementary Table 3). Additionally, the set of high-efficiency sgRNAs contained a larger fraction of sgRNAs with tag sites at early introns compared with the whole library, indicating that tagging closer to the N terminus is more likely to be successful (Fig. 1f and Extended Data Fig. 2a). We further observed much weaker but statistically significant enrichment of tagging sites with lower hydrophobicity as measured by the Kyte–Doolittle scale[22] (Extended Data Fig. 2b–d), and within less-structured domains as indicated by AlphaFold[23] confidence scores (Extended Data Fig. 2e,f). Finally, we could not observe a clear enrichment dependent on sgRNA cutting efficiency score[24,25], likely because the library was designed by selecting sgRNAs for high cutting efficiency (Extended Data Fig. 2g,h).

### Generating a multicolour intron-tagged cell pool
Cell pools generated with the procedure described above can be used for monitoring subcellular localizations and abundance of proteins in a pooled format by in situ sequencing of the expressed sgRNA for identification of responding clones and proteins[19]; however, a purely image-based recognition of clones and proteins based on the localization patterns and intensity levels of tagged proteins is so far not possible, as proteins localizing to the same compartment cannot be easily discriminated. We hypothesized that tagging two different proteins with different colours in each cell might create enough diversity to discriminate hundreds of clones by using the combination of the two localization patterns and intensity values as visual barcodes. For tagging a second protein in every cell, we performed a second round of intron tagging using an orthogonal sgRNA library targeting introns of a different intron frame and a matching minicircle with the coding sequence of the red fluorescent protein mScarlet (Fig. 2a and Extended Data Fig. 3a). Targeting introns of a different frame in the second round ensures that only the target of the second transduced sgRNA can be tagged with mScarlet and there cannot be any tagging of an unedited allele targeted by the sgRNA introduced into cells in the first round of intron tagging. Furthermore, this strategy allows for easy identification of the two tagged proteins in every cell based on the two expressed sgRNAs, as the sgRNA originating from the frame 0 library corresponds to the GFP-tagged protein and the sgRNA originating from the frame 1 library corresponds to the mScarlet-tagged protein. To further increase the visual diversity of clones in the pools, we transduced double-positive

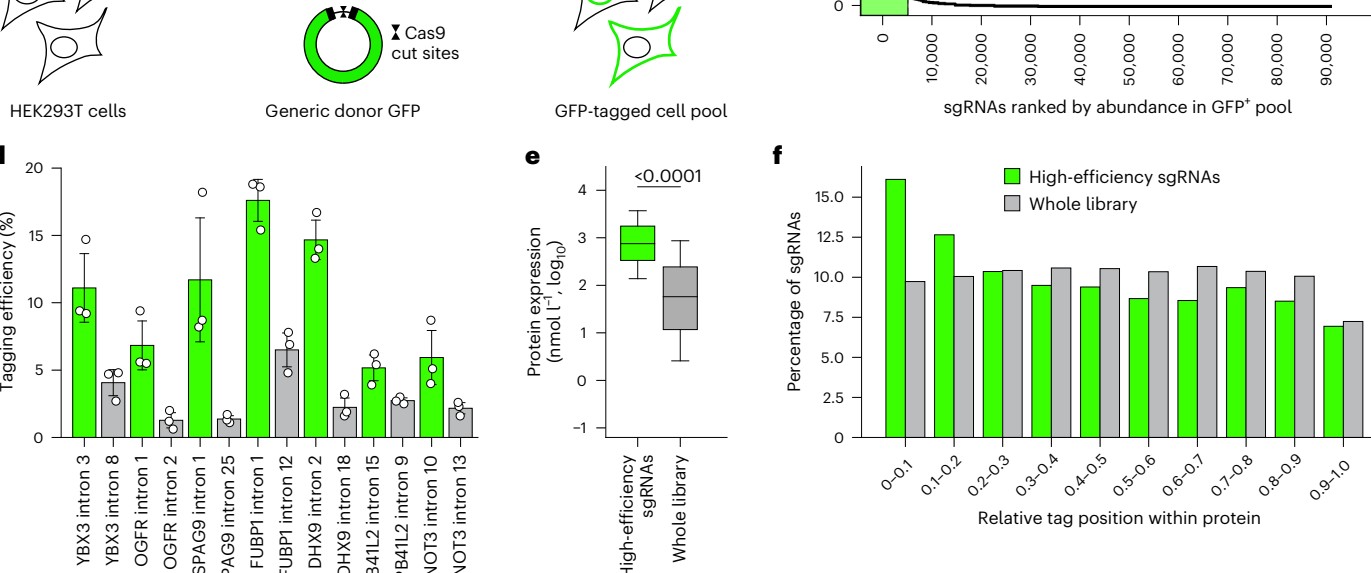

**Fig. 1 | Pooled protein tagging for the generation of vpCell pools. a,** Overview of the visual proteomics vpCell approach. **b,** Schematic of the pooled intron-tagging approach. **c,** Abundance of sgRNAs in the GFP-positive cell pool after protein tagging. **d,** Comparison of tagging efficiencies when targeting different genes with high-efficiency sgRNAs or other sgRNAs from the library. Data are shown as mean ± s.d.; n = 3 biologically independent experiments. **e,** Expression level of proteins targeted by high-efficiency sgRNAs and proteins targeted by all sgRNAs in the library. Boxes represent 25th, 50th and 75th percentiles, and whiskers represent 10th and 90th percentiles. P value < 10⁻²⁰⁰, two-sided Student's t-test; n = 58,472 sgRNAs with available expression levels of target proteins, examined over one pooled protein tagging experiment. **f,** Relative position of the tag sites in the protein of high-efficiency sgRNAs and all sgRNAs in the library.

cells with constructs expressing blue fluorescent protein (BFP) fused to different localization signals as additional visual barcodes (Fig. 2b). Finally, cells were transduced with a membrane and a nuclear marker (mAmetrine-CAAX and NLS-miRFP670-miRFP670nano; Extended Data Fig. 3b), facilitating cell segmentation during image analysis, to eventually generate a visually highly diverse cell pool (Fig. 2c,d).

By determining the sgRNA abundance in the multicolour pool, we estimated that approximately 2,500 different proteins were tagged with either GFP or mScarlet (Supplementary Table 2). We performed this procedure not only with HEK293T cells, but also with HAP1 cells and using two smaller intron-targeting sgRNA libraries targeting 287 cancer-associated genes (Supplementary Tables 1 and 2). This confirms

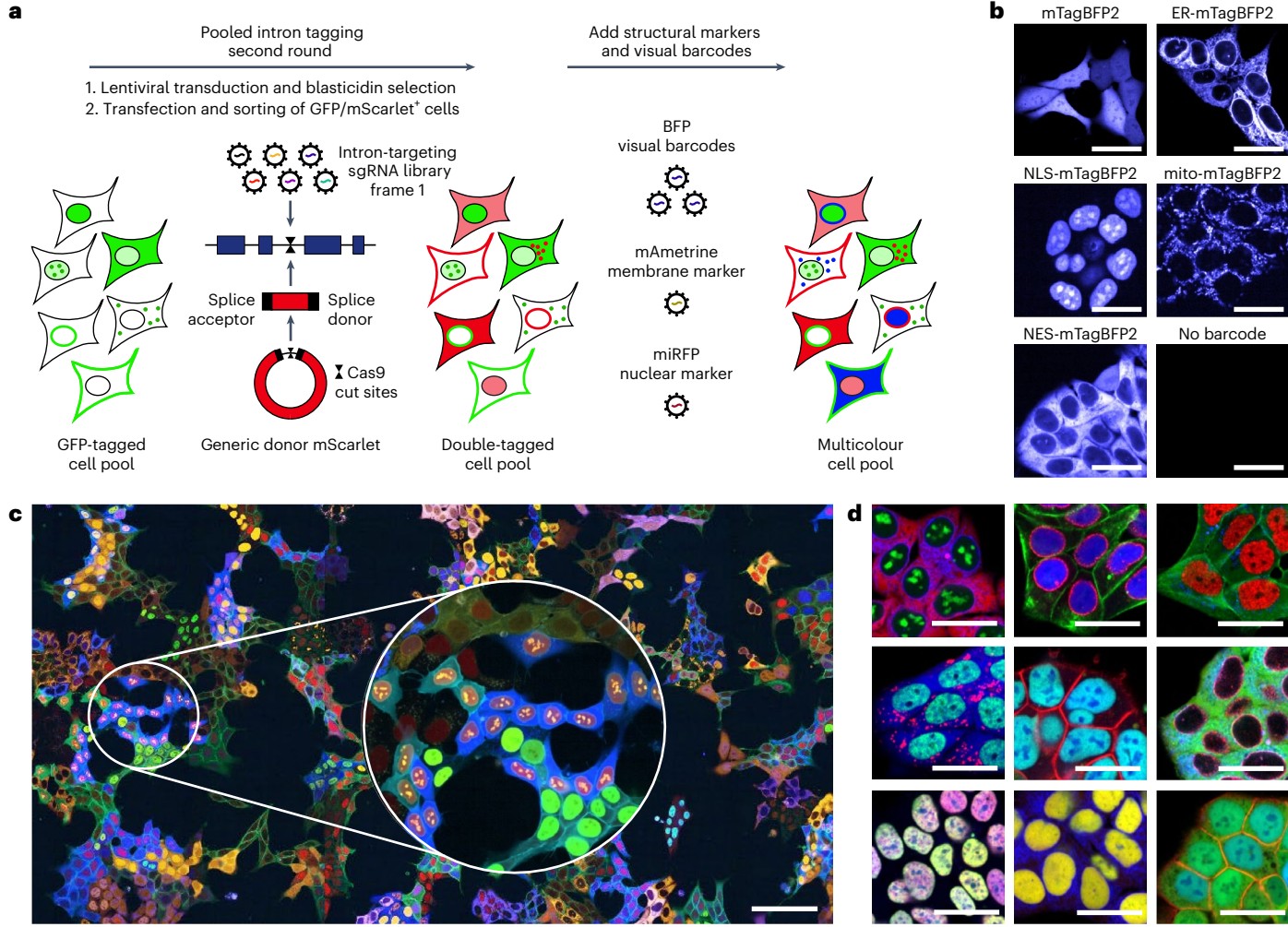

**Fig. 2 | Generation of multicolour cell pools. a**, Schematic of the generation of multicolour cell pools by performing a second round of intron tagging using a sgRNA library targeting frame 1 introns and a matching DNA donor followed by lentiviral transduction with fluorescent markers. **b**, Visual barcodes. Fluorescence microscopy images of HAP1 cells transduced with constructs expressing BFP localizing to different subcellular compartments. Scale bars, 25 μm. NLS, nuclear localization signal; NES, nuclear export signal. **c**, Stitched image from multiple FOVs of a multicolour pool, 2 days after seeding. Representative of $n$ = 3 tagging experiments. Scale bar, 100 μm. **d**, Examples of clonal cell lines isolated from the multicolour pool. Scale bars, 25 μm.

the applicability of the approach to all cell models that enable robust transduction and genome editing efficiencies and its scalability from focused gene sets to the genome scale.

## The vpCell clone collection of endogenously tagged proteins

Before using any of our cell pools in pooled screening applications, we generated a large clone collection from these cell pools and confirmed the correct subcellular localization of tagged proteins in the majority of individual clones. Clonal cell lines were isolated from pools by single cell dilution in 384-well plates, followed by imaging using confocal fluorescence microscopy and multiplexed sgRNA amplicon sequencing to identify the two tagged proteins in each clone (Fig. 3a). In total, we isolated 4,576 clonal cell lines to obtain a clone collection covering 1,158 proteins, with 1,272 clones being isolated from a HEK293T cell pool tagged using the genome-wide libraries, 1,601 clones from the HAP1 genome-wide tagging effort and 1,703 clones from the HAP1 pool with tagged genes that are associated with cancer covering 170 proteins of the 287 that were initially targeted by that library (Fig. 3b and Supplementary Table 4). We manually annotated the subcellular protein localization of each protein present in our clone collection with 12 different localizations, using an established methodology[2,18]. As many proteins localize to multiple compartments, each protein

was annotated with up to two localizations, with the most common combination of localizations being 'cytoplasm/nucleoplasm'. Comparing our annotations with the Human Protein Atlas (HPA)[2] confirms the correct localization of the majority of intron-tagged proteins, with disagreeing annotations mostly being differences within similar compartments (for example, nucleoplasm versus nuclear speckles; Fig. 3c and Supplementary Table 4). Overall, 79.4% of proteins have at least one localization annotation in common with the HPA, including 37.1% of proteins that share the same set of annotations, which are similar rates that were observed when comparing N- or C-terminally tagged proteins in the OpenCell[18] collection with the HPA. For proteins and clones with completely discordant annotations, it is possible, as with every tagging event, that the subcellular localization is affected by the protein tag. Deciding on the ideal tag position for each protein is only possible by individually comparing the observed localizations with multiple resources or literature. Therefore, we provide images of all clones including information on the tag position and sgRNA sequences at https://vpcells.cemm.at (Fig. 3d). The high diversity of this clone collection (Fig. 3e) enables image-based clustering of the different tagged proteins by their subcellular localization patterns (Fig. 3f and Extended Data Fig. 4). These data confirm that clones generated by pooled intron tagging can be used as reporters for subcellular protein localizations.

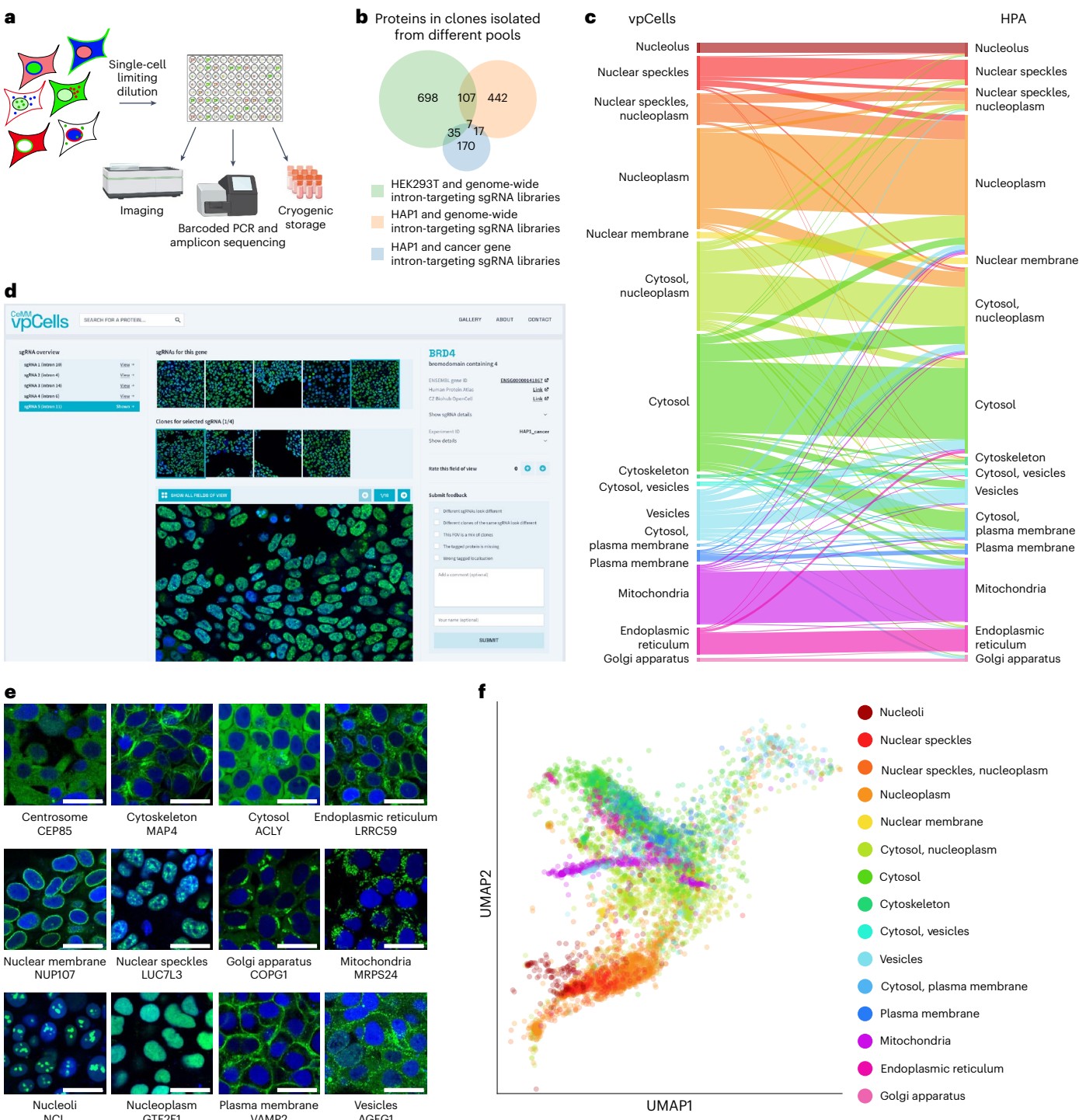

**Fig. 3 | Generation of a clone collection from cell pools. a**, Schematic of the isolation, imaging and genotyping of clonal cell lines for the cell pools. **b**, Number of different proteins that are tagged in clones isolated from different cell pools. **c**, Comparison of annotations of subcellular localizations in proteins in clones isolated from cell pools and in the HPA. Every band in the flow-diagram represents a group of proteins with a certain localization annotation in vpCells and the same or a different annotation in the HPA. The width of each band represents the number of proteins in each group. Some localization annotations are combinations of multiple subcellular localizations such as cytosol and nucleus. Only the 15 most common annotations are shown. **d**, Screenshot of the vpCells website for exploration of subcellular localization of intron-tagged proteins in isolated clonal cell lines. **e**, Examples of proteins and detected subcellular localizations. Tagged protein in green, nuclear channel in blue. Scale bars, 25 μm. **f**, Uniform Manifold Approximation and Projection (UMAP) representation of vpCells proteins. Each dot represents a protein in a particular clone, averaged across all its cells. Only the 15 most common annotated localizations are shown.

## Computer vision for identifying clones in vpCell pools

We built a computational pipeline to directly determine clone identity from microscopy images without the need for sequencing-based sgRNA identification (Fig. 4a,b). A key prerequisite is a robust method for cell segmentation[26,27] in images of cell pools despite potential clone-specific differences in the intensities of the different channels. We segmented whole cells based on the mAmetrine-CAAX membrane marker and nuclei based on the NLS-miRFP670-miRFP670nano signal, performing

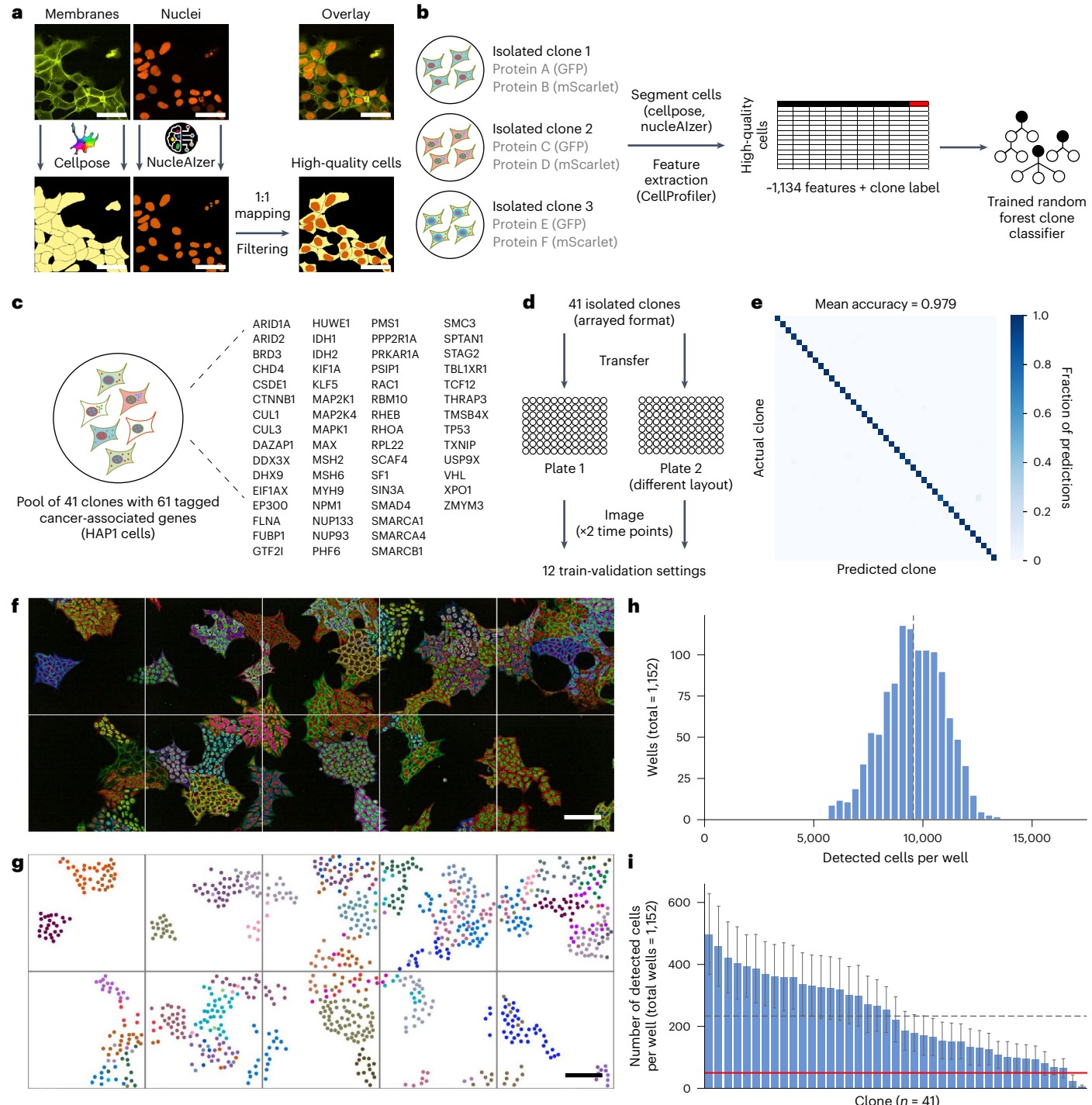

**Fig. 4 | Classification of intron-tagged clones in array and pool with machine learning. a**, Segmentation of whole cells (membranes) and nuclei is performed using cellpose and nucleAIzer. High-quality cells are obtained by 1:1 mapping of whole cells to nuclei and additional filtering. Scale bars, 50 μm. **b**, Schematic of training of the random forest model to distinguish clones at the single-cell level. **c**, Set of 61 cancer driver genes in the pool of 41 clones. **d**, Generation of the training dataset (and 12 train-validation settings) for the 41 clones in arrayed format. **e**, Assessment of predictive performance in arrayed format (confusion matrix), normalized by true labels (rows), average over 12 train-validation settings. **f**, Pool of 41 clones, ten FOVs from a representative well (total wells,

1,152). Overlay of GFP, mScarlet and BFP channels. Scale bar, 100 μm. **g**, Model predictions for **f**. Each dot corresponds to a detected high-quality cell, coloured by its predicted clone identity. Scale bar, 100 μm. **h**, Distribution of number of detected high-quality cells per well (100 FOVs imaged per well). The dashed line indicates the mean value of 9,843 cells. **i**, Distribution of number of detected high-quality cells per clone per well (total wells = 1,152). The mean value for each clone is shown, error bars indicate s.d. The dashed line indicates the overall mean of 233 cells per clone. The red line denotes the threshold of 50 cells per clone, which we consider as the minimum number to robustly detect perturbation-induced changes.

filtering to obtain high-quality cells (Fig. 4a; for details see Methods and Supplementary Fig. 2). The filtering step discards the large majority of mitotic cells.

To develop a computational image-based clone recognition algorithm, we first assembled a new cell pool from 41 vpCell clones covering 61 cancer-associated proteins to serve as ground truth (Fig. 4c). These

clones were also imaged in an arrayed format at two time points in two biological replicates (Fig. 4d). We took care to randomize clone positions between replicate (imaging plate) to prevent possible over-fitting to well position during model training. We trained a random forest classifier to discriminate the clones at the level of single cells based on CellProfiler features[28] extracted for each cell (Supplementary Tables 5 and 6). To estimate the predictive performance, we trained and validated the model on independent subsets of the dataset. We found that clones can be discriminated with accuracy and macro F1 scores over 97% (Fig. 4e shows the average over 12 train-validation settings; Extended Data Fig. 5a and Supplementary Table 7 show each train-validation setting separately).

To estimate the upper limit of the number of clones that our computational approach can distinguish, we also trained a model on the largest possible non-redundant set of vpCell clones. We selected 1,065 clones fulfilling the following criteria: (1) carrying a different combination of their two tagged proteins; and (2) with at least 200 high-quality cells available. Training and validating the model on measurements from separate time points, we achieved validation accuracy of over 93% (Extended Data Fig. 5b), suggesting that our approach could be applied to much larger sets of up to hundreds or even thousands of clones.

Finally, we evaluated the performance of our pipeline on the pool of 41 clones by training a model on the combined arrayed dataset (all replicates and time points) (Fig. 4d). We found very good performance based on (1) cells growing in colonies being assigned to the same clone (Fig. 4f,g and Extended Data Fig. 6); and (2) rarefaction analysis by manually assembled subpools containing fewer than the 41 clones in which omitted clones are also not found computationally (Supplementary Table 7). By imaging 100 fields of view (FOVs) per well in a 96-well plate with ×40 magnification, we on average cover 9,843 individual high-quality cells (Fig. 4h). For 39 of the clones in the cell pools, this results in more than 50 individual cells being imaged, a number we consider sufficient to detect perturbation-induced changes (Fig. 4i). The proportions of clones in the pool were not fully even, but we ruled out this was caused by widely different proliferation rates (Extended Data Fig. 6h). We therefore applied these conditions for detection of perturbation-induced localization changes.

### Monitoring abundance and localization of large protein sets

To detect perturbation-induced changes on cell pools, we first imaged the unperturbed cell pool, then induced the perturbation and imaged the cell pool again after 6 h (Fig. 5a). The computational protocol developed before can unambiguously assign clone identity in the unperturbed image, but perturbation-induced localization changes may interfere with clone recognition following drug treatment. We therefore assign clone identity in the perturbed image based on the following assumptions: (1) considering the low migratory potential of the cell lines used (Fig. 5b), cells in the perturbed image can only be assigned to clones present within a radius of 100 μm in the unperturbed image; and (2) compound-induced changes will only impact one of the channels, enabling majority calling based on leave-one-channel-out computational experiments (for details, see Methods). With clone identities in perturbed images assigned, we can then determine changes by comparing CellProfiler features in perturbed cells to control cells.

We treated the cell pool of 41 clones covering 61 cancer-associated proteins with 1,059 compounds that we observed in previous

experiments to impair cell proliferation[29] (Supplementary Table 8). For a proof of concept, we first validated the computational protocol using the cell pool treated with the BRD4-degrader dBET6 (ref. 30). We observed robust and specific loss of mScarlet signal in a clone tagged with CUL3–GFP BRD4–mScarlet (Fig. 5c–e and Extended Data Fig. 7a–e). Manual annotation of cells of this clone in the entire well confirmed the high predictive performance of the post-perturbation clone detection model with an F1 score of 0.9603 (Extended Data Fig. 7f).

We then analysed the entire set of drug–protein interactions, testing whether any of 90 selected CellProfiler features were altered. We observed widespread compound-induced changes, which we further filtered for obtaining 44 confirmed drug–protein interactions (Fig. 5f–h, Extended Data Fig. 8 and Supplementary Table 8). These included a number of known or predicted drug–protein interactions, which thereby further validate the approach. Among these, we observed a drastic reduction in the intensity of the mTOR regulator RHEB following treatment with mycophenolic acid and pralatrexate, consistent with the known sensitivity of the protein to purine metabolite deficiency[31] and its drastic depletion following mycophenolic treatment in a recent MS-based approach[32]. We also observed the specific loss of SMARCA4 signal for two independent intron-tagged clones treated with the known SMARCA2/4-degrader ACBI1 (ref. 33). With the translation inhibitor cycloheximide we observed loss of intensity of KLF4, which is among the proteins with the shortest half-life in the cell pool[34]. In addition to these known and predicted drug–protein interactions, we also observed specific effects on subcellular localization and condensation of SMAD4, NPM1, DAZAP1, RAC1, XPO1, DDX3X and CUL3 with certain compounds (Fig. 5g,h), effects that we could validate in cell pools and in individual tagged cells lines. Overall, these results demonstrate the applicability of our approach for pooled screening in uncovering protein–drug interactions beyond changes in protein abundance.

### Monitoring of protein dynamics discovers exportin inhibitors

To further elucidate the mechanism of action of the discovered protein–drug interactions, we focused on the decrease in XPO1 levels in response to treatment with several of the uncharacterized, antiproliferative compounds (Figs. 5g and 6a and Extended Data Fig. 9a). XPO1 is responsible for the nuclear export of proteins containing a nuclear export signal, which also includes some tumour-suppressor genes, making this gene a target for cancer therapy[35]. There are two US Food and Drug Administration-approved XPO1 inhibitors selinexor and verdinexor, both of which have been shown to cause XPO1 degradation in addition to XPO1 inhibition[36,37]. Notably, for screening hit Z384372236, we not only detected a decrease in XPO1 fluorescence intensity, but, because we were simultaneously monitoring multiple proteins in parallel using our cell pools, we could also observe effects on other proteins in the pool. These included a decrease in the levels of nuclear HUWE1 and an increase in SMAD4 and CUL3 (Fig. 6b,c). As SMAD4 is a known cargo of the nuclear export machinery, we reasoned that some of the hit compounds might indeed be XPO1 inhibitors and degraders. To further characterize the compound-induced abundance and localization changes, we performed time-lapse microscopy over a period of 7 h in live cells with intron-tagged XPO1 and SMAD4 with the hit compounds tested at three concentrations, using the known XPO1 inhibitors selinexor, verdinexor and leptomycin B as positive controls (Fig. 6d–f

**Fig. 5 | Drug screen with pool of 41 clones. a**, Schematic visualization of the pooled screen strategy. **b**, Pooled clones imaged at $t = 3, 4$ and 5 days after seeding. Representative image from a single experiment. Scale bars, 100 μm. **c**, Degradation of BRD4 with dBET6, positive control. Representative image from pooled screen. Arrows point to example cells with intron-tagged BRD4 in mScarlet channel. Model predictions (bottom) are robust to perturbation (the BRD4 clone cells are assigned the correct label). Scale bars, 25 μm. **d**, Change in median nuclear intensity of BRD4 in response to compounds in the pooled

screen compared with the average of negative control dimethylsulfoxide (DMSO) wells. **e**, The effect of dBET6 treatment on CellProfiler features for each tagged protein in the pool, quantified by counting significantly changed variables with an absolute $z$-score above 0.5. **f**, Overview of the hit calling workflow. **g**, Heat map visualization of proteins and compounds of the 44 visually confirmed hits. **h**, Top visually confirmed hit protein and compound combinations before and after treatment. Visualization of a single colony in the pool. Scale bars, 25 μm.

and Extended Data Fig. 9b). While the decrease in fluorescence intensity in XPO1 was detectable after 2–3 h, the nuclear accumulation of SMAD4 in response to some, but not all, hit compounds was already visible after 1 h. This suggests that some hit compounds inhibit XPO1 before degradation happens and that the degradation is a secondary effect that is not required for the nuclear accumulation of SMAD4.

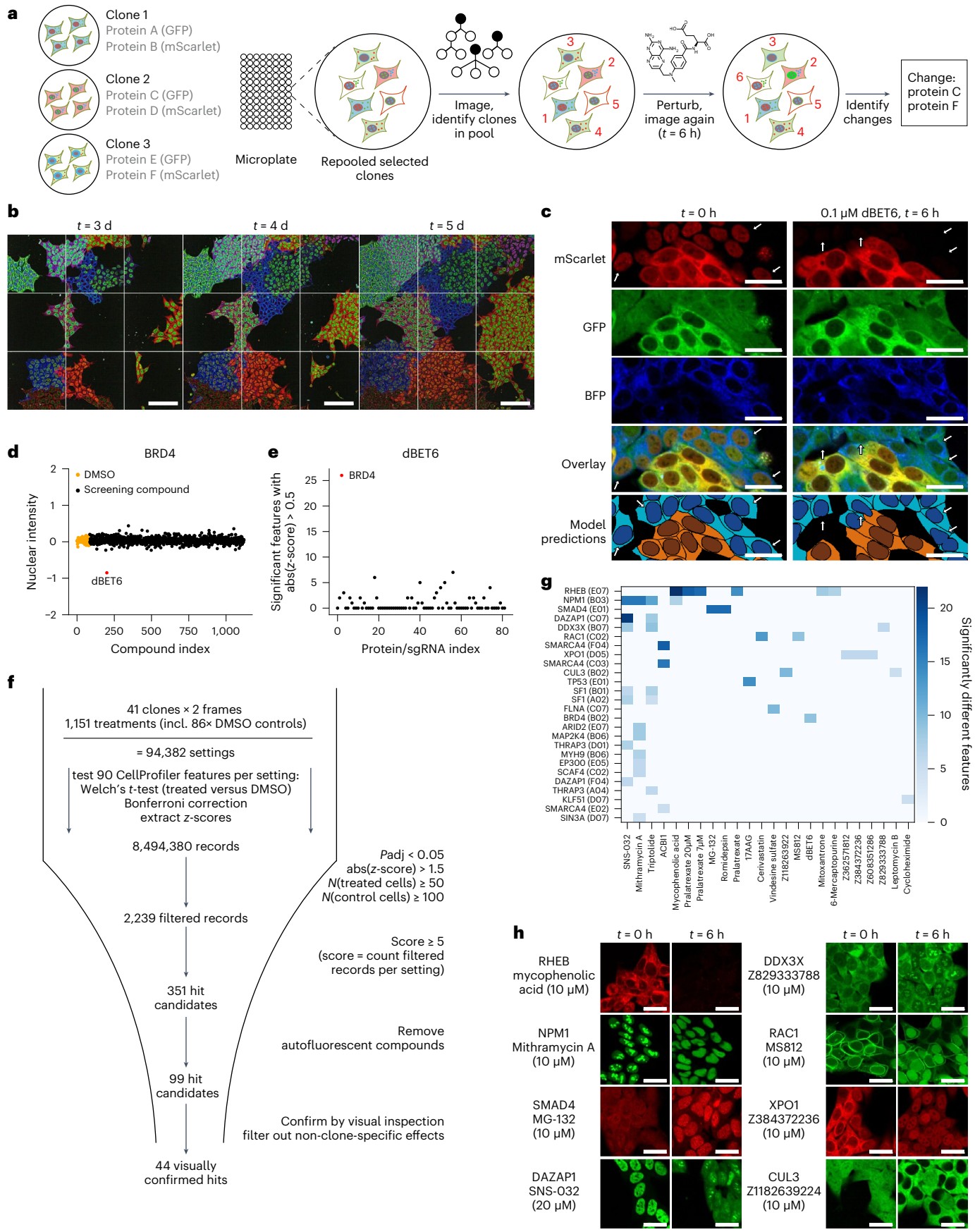

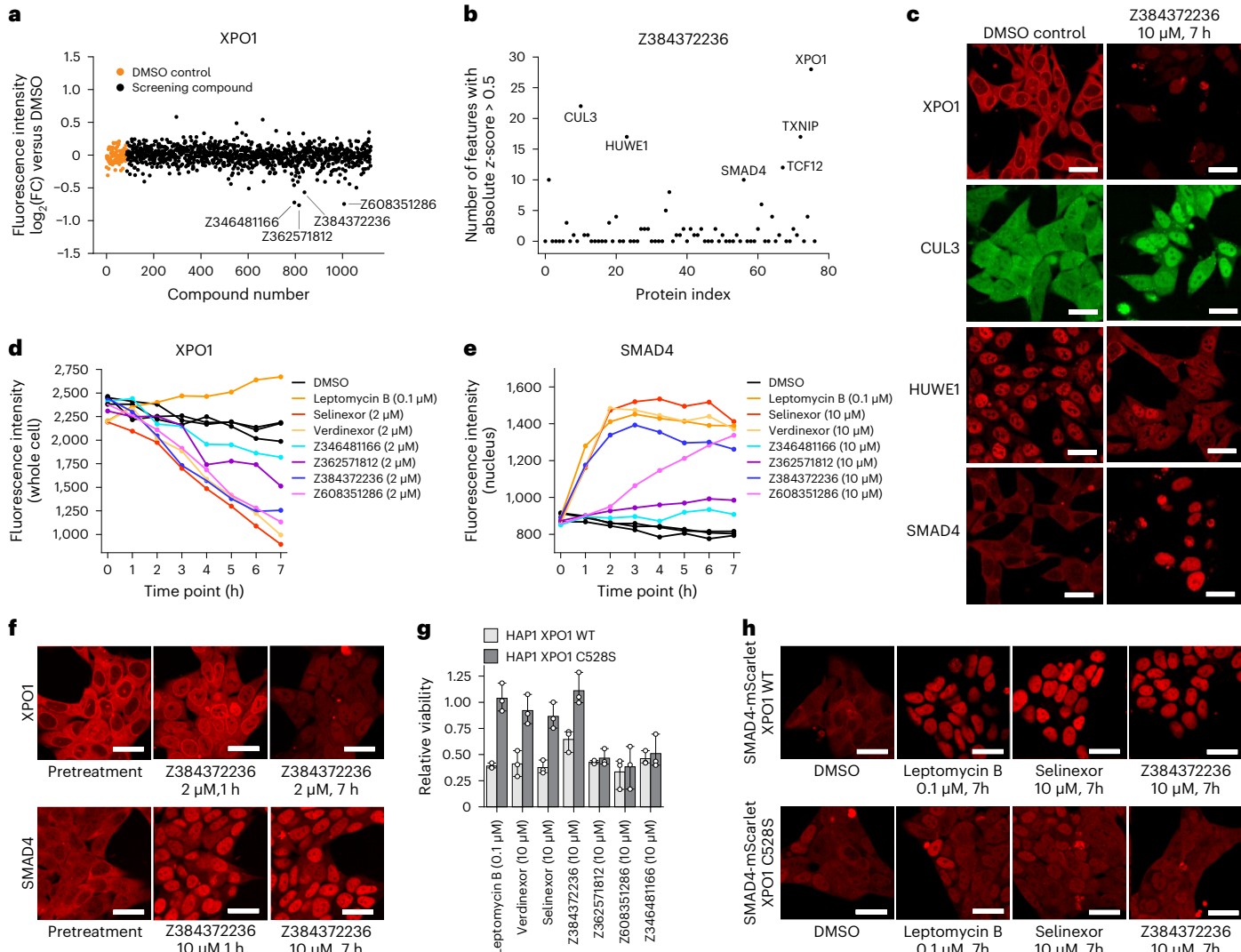

**Fig. 6 | Z384372236 inhibits nuclear export by binding to XPO1. a**, Fluorescence intensities of tagged XPO1 in the cell pool across all tested compounds. FC, fold change. **b**, Compound activity of Z384372236 on all proteins represented in the cell pool. **c**, Validation of drug–protein interactions in an arrayed format using clonal cell lines; *n* = 1 experiment. Scale bars, 25 μm. **d**, Fluorescence intensity of tagged XPO1 in response to selected hit compounds measured by 7-h time-lapse microscopy of a clonal cell line. **e**, Fluorescence intensity of tagged SMAD4 in response to selected hit compounds measured by 7-h time-lapse microscopy

of a clonal cell line; *n* = 1 experiment. **f**, Representative images of the 7-h time-lapse microscopy of cell lines with tagged XPO1 and SMAD4 treated with the hit compound Z384372236. XPO1 images correspond to **d** and SMAD4 images correspond to **e**. Scale bars, 25 μm. **g**, Effect of the XPO1 C528S mutation on the cytotoxicity of hit compounds and known covalent binders of XPO1. Data are shown as mean ± s.d.; *n* = 3 biologically independent experiments. **h**, Effect of the XPO1 C528S mutation on the nuclear translocation of SMAD4 in response to compound treatment; *n* = 1 experiment. Scale bars, 25 μm. WT, wild type.

For leptomycin B we did not observe any degradation of XPO1, but saw a strong nuclear accumulation of SMAD4. The most potent hit compound Z384372236 showed a compound-induced effect at a concentration as low as 0.5 μM. To test whether the observed effect is specific for the XPO1 clone that was included in the pool, or whether the degradation can be observed independent of tag position, we monitored XPO1 fluorescence intensity in nine clones from our collection that are all tagged at different introns. In seven out of nine clones, we observed the degradation by Z384372236 (Extended Data Fig. 10a,b). In the two clones without a degradation phenotype, the fluorescence intensities were already very low in the unperturbed state, indicating that these tag positions might result in destabilized protein. Of note, degradation of XPO1 in response to Z384372236 was not only observed in the majority of tagged clones but also in cells expressing endogenous, untagged XPO1 (Extended Data Fig. 9c).

Selinexor, verdinexor and leptomycin B are covalently binding Cys-528 in the cargo binding pocket of XPO1 (refs. 38,39). To test whether

Z384372236 also has the same binding mode despite being structurally different (Extended Data Fig. 9a), we introduced the XPO1 C528S mutation in HAP1 cells. We found that this mutation indeed conferred resistance not only to selinexor and leptomycin B, but also to the antiproliferative effects of Z384372236 (Fig. 6g). Furthermore, in cells with the XPO1 C528S mutation and intron-tagged SMAD4, there is no nuclear accumulation of SMAD4 as observed in cells with wild-type XPO1 following compound treatment (Fig. 6h). Together, these data suggests that the mode of action of the antiproliferative compound Z384372236 is inhibition of nuclear export via binding to XPO1. These findings confirm the power of the vpCell approach as a framework to simultaneously follow multiple proteins within a pathway for mechanistic studies.

## Discussion

Scalable pooled methods can drastically expand opportunities for systematically probing cellular mechanisms[40]. For monitoring subcellular

protein localization using imaging-based methods, this is because testing one additional compound in arrayed format requires the thawing, expansion, counting, seeding and profiling of hundreds to thousands of cell lines or the use of large antibody panels. Such efforts are, at best, feasible in few specialized platforms, whereas cell pools can be handled efficiently by individual laboratories. The vpCell technology has the potential to democratize subcellular localization studies, enabling the rapid generation, sharing and characterization of multicolour cell pools in different cell models.

We here show that a pooled intron-tagging strategy can be efficiently scaled to generate highly diverse cell pools of endogenously tagged proteins. Using genome-wide sgRNA libraries we identify proteins and tag positions that are more favourable for tagging, resulting in enrichment in our cell pool. High expression levels and targeting an early intron are the main factors for successful tagging, with hydrophilic and unstructured regions giving a minor benefit. While in most of these successfully tagged clones protein localization is consistent with current reference datasets, it is still possible that tagging impairs protein function, interactions or modifications, as is the case for any modification, including at the N or C termini[41]. Future developments of the overall strategy, together with pooled N- or C-terminal tagging strategies[42] will further boost the number of proteins that can be tagged and increase the confidence in observed localizations.

Of particular importance for the scalability of pooled localization studies is a method for rapidly assigning the identity of tagged proteins. Discriminating hundreds of proteins only based on microscopy images is so far not feasible, because magnifications compatible with rapid imaging of thousands of cells result in near-identical localization patterns for proteins residing in the same organelle, many of which have similar expression levels, and heterogeneity, for example, caused by cell cycle effects further complicates matters. We here show that multicolour tagging enables automated clone identification in cell pools. Rather than using specific tags solely for barcoding[43], we directly use the combination of localization patterns and intensities of the tagged proteins as visual barcodes. We estimate that this method enables discriminating thousands of clones, with the option to further increase this number up to proteome-scale cell pools by expanding the set of BFP barcodes and modulating structural channel intensities. Improved computational methods for feature extraction, including deep-learning-based protein localization profiling[44–47] will further increase the number of clones that can be discriminated and enable the identification of more subtle localization changes. We believe that, rather than in computational detection, the limiting factor for multiplexing ultimately lies in ensuring a well-balanced representation of clones in the pool and the imaging time required to capture enough cells for each clone.

Any area of biology that involves the characterization of cellular perturbations can benefit from pooled approaches to study subcellular protein localization, including the deep characterization of cellular processes, compound mechanism-of-action studies and the discovery of bioactive compounds. Our study provides a proof of concept for all of these aspects.

First, we show that the vpCell approach can successfully identify bioactive compounds. For such efforts, focused vpCell pools that cover desired targets, their upstream and downstream regulators and potential off-targets might be best suited. The options of customizing sgRNA libraries or assembling such a cell pool from the wider vpCell collection enable the rapid implementation of this approach. Here, we assembled a cell pool of cancer drivers and used it to probe the effects of antiproliferative compounds. Despite the early 6-h time point used for profiling, we find widespread compound-induced changes to protein abundance and subcellular localization, including known effects of chemical probes dBET6 and ACBI1 on their target proteins as well as other effects of previously uncharacterized compounds.

Of the pathways targeted by these compounds, we focused on the nuclear export machinery. Using the approved exportin inhibitor selinexor highlights the potential of time-resolved imaging of cell pools for mechanistic studies. We find that selinexor causes the rapid nuclear accumulation of both its direct target XPO1 and its cargo SMAD4 within 1–2 h. In contrast, XPO1 degradation only occurs at later time points (and not at all with leptomycin B), suggesting that it does not contribute to primary effects.

Finally, we show that vpCell pools can identify compound mechanism of action. The success of the Cell Painting[48,49] assay highlights the power of morphological features to discover compound mechanism of action. While the rapid dye-based staining of subcellular structures is a key advantage of Cell Painting, the vpCell approach offers the opportunity to monitor tens or hundreds of individual proteins in each of these organelles in live cells, suggesting the two approaches to be highly synergistic. Here, we identify a set of compounds that show similar behaviour in vpCell pools to the known exportin inhibitors selinexor, verdinexor and leptomycin B. One class of these compounds (exemplified by Z346481166, Z362571812 and Z608351286) strongly resembles Michael acceptors, previously described as covalent XPO1 inhibitors[50]. Notably, only the most potent of these compounds also leads to nuclear accumulation of XPO1 cargo proteins, and also only at later time points when XPO1 is already degraded and its antiproliferative effect is independent of the selinexor resistance mutation XPO1$^{C528S}$. In contrast, a structurally different compound that we identified, Z384372236, shows effects on both XPO1 and its cargo proteins with similar kinetics as the clinical compounds. These data identify the compound as a distinct chemotype of exportin inhibitors.

In summary, the vpCell approach that we developed enables the efficient monitoring of subcellular protein localization to identify the kinetics of perturbation-induced changes.

## Online content

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

## Methods

### Intron-targeting sgRNA library design

Intron-targeting sgRNA libraries were generated following a strategy we described previously[19]. In brief, the Ensembl BioMart data-mining tool was used to select transcripts with a Consensus Coding Sequence (CDS) ID of 19,035 human genes and to obtain chromosomal coordinates of introns for sgRNA design. For genes with multiple transcripts with a Consensus CDS ID, the transcript with the longest CDS was chosen. For each of the selected transcripts, the information on CDS start, exon frame and the exon start and end chromosomal coordinates were used to define intronic regions. To avoid selecting sgRNAs close to the exon/intron junction and disrupting splice-acceptor and splice-donor sites, only the intronic region that is at least 20 bp away from exon/intron junctions was used for sgRNA design. The GuideScan tool[25] was used to select up to 20 Cas9 sgRNAs with the highest on-target scores for each intron (GuideScan database v.1.0). These guides were ranked for each intron based on a combined on- and off-target score and annotated with gene name, Ensembl transcript ID, intron number and intron frame to generate a database of 2,979,354 sgRNAs targeting 162,261 introns of 16,279 genes (available for download at vpcells.cemm.at). For the frame 0 genome-wide intron-targeting sgRNA library, the top ranked sgRNA for each intron in frame 0 was included in the library. Additionally, for genes with only two or three targetable introns, the second ranked sgRNA was also included and for genes with only one targetable intron, the third ranked sgRNA for that intron was also included in the library. Approximately 0.5% of sgRNAs in the library target regions of overlapping genes and map to introns of two primary transcripts. These sgRNAs were annotated with two introns and genes (separated by '|' in their sgRNA name) and were excluded from any further analysis, because they cannot be unambiguously assigned to a single target. The frame 0 genome-wide intron-targeting sgRNA library also includes 1,000 non-targeting sgRNAs for a total number of 90,657 sgRNAs targeting 73,817 introns of 14,158 genes. The frame 1 genome-wide intron-targeting sgRNA library was generated using the same database and consists of 72,580 sgRNAs targeting 51,939 introns of 14,011 genes. Two smaller libraries were generated, targeting frame 0 and frame 1 of a total of 287 genes associated with cancer biology and consist of 2,511 and 1,763 intron-targeting sgRNAs. The genome-wide libraries have been deposited with Addgene (Human Genome-wide Intron Tagging Library, Frame 0 and Human Genome-wide Intron Tagging Library, Frame 1).

### Cloning of intron-targeting sgRNA libraries

Libraries were synthesized as oligonucleotide pools by Twist Biosciences and cloned into the CROPseq vector using Gibson assembly. For frame 0 libraries, the CROPseq-Guide-Puro vector was used, and for frame 1 libraries, the puromycin resistance in the vector was replaced with a blasticidin resistance before library cloning. Oligonucleotide pools were PCR amplified and the vectors were digested with *BsmBI* and purified. Multiple Gibson assembly reactions were performed and electroporated into Endura electrocompetent cells (Lucigen), plated on multiple bioassay dishes and plasmid DNA was isolated using multiple columns of a midiprep DNA purification kit (QIAGEN Plasmid Plus Midi kit). Library coverage was determined by counting colonies on dilution plates and was between ×200 and ×500 for the different libraries.

### Minicircle production

For the production of minicircle DNA containing a single generic sgRNA target site followed by a splice acceptor, a 20-amino acid linker sequence, the CDS of EGFP, a 20-amino acid linker sequence and a splice-donor site, the required DNA fragment was amplified from Intron-Tagging-EGFP-Donor plasmid (Addgene, #159740) and cloned into the pMC.BESPX-MCS1 parental minicircle production plasmid (System Biosciences) by EcoRV digest and Gibson assembly. Parental plasmid was transformed into ZYCY10P3S2T *Escherichia coli*

minicircle production strain (System Biosciences MC-Easy Minicircle DNA Production kit) and a colony containing the correct parental plasmid was used for minicircle production as described by the manufacturer. In brief, bacteria were grown overnight in TB medium and on the next day, induction medium containing l-arabinose was added to induce *att* recombination and parental plasmid backbone degradation. Minicircle DNA was isolated from bacterial pellets using multiple columns of an endotoxin-free midiprep DNA purification kit (QIAGEN Plasmid Plus Midi kit) and the produced minicircle was analysed by restriction enzyme digest and gel electrophoresis. For generating a minicircle that is compatible with frame 1 introns and contains the CDS of mScarlet, the Intron-Tagging-EGFP-Donor plasmid (Addgene, #159740) was modified by adding 2 nucleobases after the splice acceptor and 1 nucleobase before the splice donor for in-frame splicing when targeting frame 1 introns. EGFP in that plasmid was replaced with the CDS of mScarlet-I[51] before cloning the respective DNA fragment into the minicircle parental plasmid and minicircle production as described above. The parental minicircle plasmids for EGFP and mScarlet have been deposited with Addgene.

### Cell culture

HEK293T (ATCC CRL-3216) cells were grown in Dulbecco's modified Eagle's medium (Sigma-Aldrich, D5796) supplemented with 10% fetal bovine serum, sodium pyruvate (final concentration of 1 mM) and penicillin–streptomycin. HAP1 cells (Haplogen, now Horizon Discovery, C631) were grown in Iscove's modified Dulbecco's medium (Sigma-Aldrich, I6529) supplemented with 10% fetal bovine serum.

### Transfection

For pooled intron-tagging experiments in HEK293T cells, $7.0 \times 10^6$ cells were seeded per 15-cm dish on the day before transfection. Each 15-cm dish was transfected with 12 μg Intron-Tagging-pX330-Cas9-Blast (Addgene, #159741) and 300 ng minicircle DNA using PEI. For pooled intron-tagging experiments in HAP1 cells, $9.0 \times 10^6$ cells were seeded per 15-cm dish, 6 h before transfection. Each 15-cm dish was transfected with 8 μg Intron-Tagging-pX330-Cas9-Blast (Addgene, #159741) and 300 ng minicircle DNA using PolyJet (SignaGen), as described by the manufacturer. For tagging of individual sgRNAs in an arrayed format, $5 \times 10^5$ HEK293T cells were seeded in a six-well plate on the day before transfection. Cells were co-transfected using PEI with 750 ng CROPseq-Guide-Puro for intron-targeting sgRNA expression, 750 ng Intron-Tagging-pX330-Cas9-mCherry and 60 ng minicircle DNA.

### Pooled protein tagging

For lentivirus production, HEK293T cells were co-transfected using PEI with sgRNA library, sPAX2 and pMD2.G. The medium was changed 12 h after transduction and virus-containing supernatant was collected after 48 h. For genome-wide tagging experiments in HEK293T or HAP1, cells were transduced with virus of the frame 0 genome-wide intron-targeting sgRNA library in CROPseq-Guide-Puro vector at a coverage of >500× to ensure library representation and at a multiplicity of infection of 0.1 to ensure single integration in most cells. After puromycin selection for 3 days, cells were expanded in puromycin-free medium for an additional 2 days before being transfected with Intron-Tagging-pX330-Cas9-Blast (Addgene, #159741) and GFP minicircle. GFP-positive cells were enriched 4 days after transfection by flow cytometry using a Sony SH800 sorter (Sony Cell Sorter Software v.2.1.6) and ultra-yield sorting settings for very high throughput at the expense of purity to obtain a cell population with ~30% GFP-positive cells. This cell population was sorted again after an additional 7 days using the standard sorting settings to obtain a pure GFP-positive cell population. For comparing editing efficiencies between cells transduced with the library and positive and negative controls, cells were only sorted 11 days after transfection, without enriching for GFP-positive cells 4 days after transfection. For genome-wide tagging experiments in HEK293T cells,

a total of $1.5 \times 10^8$ cells were transfected and approximately $1.0 \times 10^6$ GFP-positive cells were sorted.

## Construction of plasmids for fluorescent marker and visual barcode overexpression

Expression constructs for lentiviral integration and overexpression of fluorescent proteins fused to different localization signals were cloned using Gibson assembly in a vector for mammalian expression (Addgene, #52962)[52]. A list of all cloned plasmids for fluorescent protein overexpression are listed in Supplementary Table 9 and have been deposited with Addgene. For cloning, the vector was digested with AgeI and EcoRI, the vector backbone was gel purified and the fluorescent proteins mAmetrine[53], miRFP670 (ref. [54]), miRFP670nano[55] and mTagBFP2 (ref. [56]) were synthesized as gene fragments (Genewiz) and localization signals for nuclear localization[57], cytoplasmic localization[58], ER localization[59] and mitochondrial localization were added via PCR before using the fragments for Gibson assembly using HiFi DNA assembly mix (NEB) as described by the manufacturer. Lentivirus containing these plasmids were produced as described above.

## Multicolour cell pool generation

For the generation of multicolour cell pools, a second round of intron tagging was performed by transducing a pool of GFP-positive cells with virus of a frame 1 genome-wide intron-targeting sgRNA library in the CROPseq-Guide-Blast vector. After blasticidin selection for 5 days, cells were transfected with Intron-Tagging-pX330-Cas9-Blast (Addgene, #159741) and frame 1 mScarlet minicircle. GFP/mScarlet double-positive cells were sorted as described above. For expression of additional fluorescent markers, double-positive cell pools were transduced with lentivirus containing the expression cassettes for NLS-miRFP, membrane-mAmetrine and one (or none) of the five possible mTagBFP2 visual barcodes.

## sgRNA abundance in cell pools

To determine the sgRNA abundance in cell pools, genomic DNA was isolated with the DNA blood and tissue kit (QIAGEN). The sgRNA containing genomic region was amplified and Illumina adapters were added by PCR (see Supplementary Table 9 for primer sequences) and sequencing libraries were submitted for next generation sequencing (Amplicon-EZ, Genewiz). To quantify sgRNAs in the pools, sgRNA sequences were extracted from sequencing reads using Cutadapt and mapped to the sgRNA libraries and counted using MAGeCK.

## Isolation, imaging and genotyping of clonal cell lines

For the generation of a clonal cell line collection, multicolour cell pools were seeded in 384-well plates at a density of 0.7 cells per well and expanded for 7 days. Then, 70–150 clonal cell lines per 384-well plate were trypsinized and cell suspensions were transferred to 96-well imaging plates (PerkinElmer PhenoPlate) and corresponding cell culture plates. Clones on the cell culture plate were expanded for 2 days and frozen by trypsinizing cells and mixing with freezing medium for a final DMSO concentration of 10% before transferring cell suspensions to cryotubes in 96-well racks and storage in liquid nitrogen. Clones on the imaging plates were imaged after 24 h and 48 h with an Opera Phenix high-content confocal imaging system (PerkinElmer) using the ×63 water immersion objective and imaging 6–10 FOVs per well. To identify the tagged protein in each clonal cell line that was imaged, the intron-targeting sgRNA was determined by highly multiplexed amplicon sequencing. For that, cells on the imaging plates were lysed after the last imaging step and cell lysate was used for amplification of the sgRNA containing region by PCR. PCR was conducted in 384-well plates using 24 barcoded forward primers and 16 barcoded reverse primers using a unique primer combination for each well for processing four 96-well plates together on one 384-well plate. PCR products from wells of a 384-well plate were pooled and submitted for paired-end

sequencing (Amplicon-EZ, Genewiz). Sequencing reads were demultiplexed and assigned to each well using Cutadapt and mapped to sgRNA libraries using MAGeCK to obtain sgRNA read counts for each well. For assigning the identity of the GFP-tagged protein in each clone, the detected sgRNA mapping to the frame 0 libraries was used and for assigning the identity of the mScarlet-tagged protein, the detected sgRNA mapping to the frame 1 libraries was used. Only clones where an unambiguous assignment was possible were included in the clone collection. For wells with excluded clones there was either no sgRNA being detectable above background in any of the two frames or multiple sgRNAs for the same frame were detected. The criterion for a single unambiguous sgRNA in each frame was a read count more than four times that of the second most abundant sgRNA detected in a particular well. Based on our analysis of previous cell pools by integration site mapping[19], a small percentage of clones may harbour additional or aberrant integrations and therefore not be correctly annotated based on sgRNA sequencing.

## Comparison of localization annotations

The comparison between subcellular localization annotations based on images of our clone collection and HPA was conducted as described previously[18] for the comparison of N- or C-terminally tagged proteins with HPA. In brief, we manually annotated the protein localization of each protein present in our collection using 12 possible subcellular localizations (Supplementary Table 4). Proteins localizing to multiple compartments were annotated with up to two subcellular localizations. HPA localization data were downloaded from the HPA website and the 'main locations' and 'additional locations' were used for further analysis. To compare our annotations with the more diverse annotations in the HPA dataset, a set of consensus annotation labels were defined to make a comparison between the two sets of annotation labels possible. Exact matches were proteins with identical consensus annotations between the two datasets and partial matches were proteins annotated with two or three localizations in one dataset and only one or two of them matching with the other dataset.

## Properties of sgRNA target proteins and tag positions

A publicly available HEK293T protein expression dataset was used to obtain protein expression values for sgRNA target proteins[18]. For calculating the hydrophobicity scores at the tag sites of proteins, the Ensembl BioMart data-mining tool was used to obtain amino acid sequences of exons flanking the sgRNA target introns and the Kyte–Doolittle scale[22] was used to calculate the hydrophobicity score for a six-amino acid window comprising three amino acids before the tag site and three amino acids after the tag site. For calculating the AlphaFold confidence score, the Ensembl BioMart data-mining tool was used to obtain CDS positions of exons flanking the sgRNA target sites and to obtain UniProt IDs of the respective transcripts. The AlphaFold per-residue confidence scores (pLDDT) for the respective proteins were extracted from mmCIF files that were obtained from the AlphaFold DB website[23] (AlphaFold database UP000005640_9606_HUMAN_v4) and the average of the pLDDT scores of the residue immediately before and after the tag site was calculated. All scores for sgRNAs in the genome-wide frame 0 library are included in Supplementary Table 3.

## Assembly of a cell pool for pooled screening applications

To generate a cell pool in which every clone can be identified by computer vision, 41 HAP1 clones were selected from the clone collection and thawed individually, before being mixed together in equal proportions using a Sony SH800 cell sorter. The cell pool was expanded for 4 days and frozen in multiple aliquots of $1 \times 10^6$ cells per cryotube that were thawed again for screening applications. Clones were also seeded in separate wells of a 96-well imaging plate to generate training data for building a computational model that can identify clones based on localization patterns and intensities in all channels. Each clone was

seeded in two wells that were imaged 24 h and 48 h after seeding with an Opera Phenix high-content confocal imaging system (PerkinElmer) using a ×40 water immersion objective.

## Compound library

A total of 1,059 screening compounds were provided by the Molecular Discovery Platform at CeMM. A total of 439 compounds in that library were approved drugs or well-annotated chemical probes and 620 compounds were antiproliferative, drug-like screening compounds with an unknown mechanism of action. Commercially available compounds were used without further purification. For compound Z384372236 we have evidence that an oxidation product acts as the active XPO1 inhibitor. Compounds dissolved in DMSO were provided in 12 compound plates at a final screening concentration of 10 μM for the majority of compounds (Supplementary Table 8).

## Pooled screening conditions

For screening of the multicolour cell pool of 41 HAP1 clones, cells were seeded in 12 96-well imaging plates (PerkinElmer PhenoPlate) at a concentration of 2,500 cells per well. At 56 h after seeding, 100 FOVs (approximately one-third of the entire well area; each FOV has a resolution of 1,080 × 1,080 pixels) were imaged per well with an Opera Phenix high-content confocal imaging system (PerkinElmer) using a ×40 water immersion objective. For the treatment of cells with compounds, medium was added to compound plates for pre-diluting compound stocks, before transferring pre-diluted compounds to imaging plates for a final compound concentration of 10 μM and 0.1% DMSO for the majority of compounds. At 6 h after treatment, the same FOVs in compound-treated wells were imaged again as described above.

## Calculation of the expected number of cells per well in pooled screen

The well diameter of a PerkinElmer PhenoPlate 96-well plate is 6.4 mm, corresponding to a well area of 32.17 mm². The size of a single FOV at ×40 magnification is 0.1027 mm² (1 pixel = 0.2967 μm). Therefore, 100 FOVs cover approximately 0.32 of the area of the entire well. Imaging after 56 h and using an estimated doubling time of HAP1 cells of 14 h, an entire well and 100 FOVs should contain 40,000 and 12,800 cells, respectively.

## Computational processing

Imaging datasets were analysed using Python v.3.9.15 at the CeMM high-performance computing cluster, using Slurm Workload Manager v.21.08.8. The code and detailed descriptions of the conda environments with package versions are deposited at https://github.com/reinisj/intron_tagging. Before analysis, flatfield correction was performed on the generated imaging data using Harmony software v.6 (PerkinElmer)[60].

## Segmentation of cells and nuclei, 1:1 mapping and filtering to high-quality cells

Cell masks were generated based on the mAmetrine-labelled membrane channel with the 'cyto' model of cellpose[27] v.0.6.1, setting the diameter to 80 pixels. Segmentation of nuclei was performed with nucleAIzer[26] (nucleaizer-backend 0.2.1) on the miRFP670-labelled nuclear channel employing the mask_rcnn_general model[61] with default_image_size parameter set to 2,048 and a diameter of 60. Using custom scripts in Python, masks of cells and nuclei were combined by 1:1 mapping, and additional filtering was performed to obtain high-quality cells. Nuclei larger than 750 pixels were assigned to cells larger than 1,500 pixels, if their overlap was at least 0.66 of the total area of the nucleus. Only cells with a single assigned nucleus were considered further. To remove artefacts and most apoptotic cells, cells were filtered based on their N:C ratio (defined as the area of the nucleus divided by the entire cell), using the minimal threshold of 0.20 and maximal threshold of 0.65.

For each cell, the number of immediate neighbours was determined by expanding its cell mask by 5 pixels and detecting the overlapping cells. To remove additional apoptotic cells, stricter filtering criteria were performed for mapped cells without any neighbours and nucleus and cell area below 2,000 and 5,000, respectively. Descriptors of solidity and eccentricity were calculated for the cell and nuclei objects using the measure.regionprops module of scikit-image[62], v.0.19.1. Mapped cells with cell solidity above 0.95 and the sum of cell and nuclear eccentricity above 1.4 were discarded. Finally, mapped cells with nuclei within 2 pixels of the FOV edge were removed. For the remaining high-quality cells, three object masks were saved: (1) entire cell, (2) nucleus and (3) cytoplasm, defined by subtracting the nucleus from the entire cell.

## Feature extraction, random forest models

For each of the three objects associated with a high-quality cell, 501 descriptor variables (Supplementary Table 5) were extracted with Cell-Profiler v.4.2.1 (ref. 28), using all five fluorescent channels as input for each FOV. A random forest model was trained on 1,455 intensity-based features, not including 48 features describing area and shape. The scikit-learn library[63] (v.1.1.3) implementation of random forest was used, with default hyperparameters. The dataset used for training consisted of four measurements (each clone seeded in two plates with a different layout, imaged at two time points). This corresponds to 12 combinations where the train and validation sets are different (Fig. 3d,e). The final model was trained on the entire dataset comprising all four measurements.

## Dimensionality reduction of CellProfiler features

Using the vpCells atlas dataset (3,469,778 cells), we first reduced the set of 274 CellProfiler features per single channel to 90 non-redundant variables by calculating Pearson's correlation between all pairs and iteratively discarding features with correlation above 0.9 to others (Supplementary Table 5). Second, a two-dimensional representation was obtained by running the UMAP algorithm[64] on all cells, using the Python implementation v.0.1.1 (ref. 65) with default hyperparameters. Finally, we calculated the mean UMAP dimensions for each protein across all its cells.

## Detection of clones in pool before and after perturbation

For each of the two time points imaged in the pooled screen (pre-treatment and $t = 6$ h post-treatment), a slightly different strategy was applied. For the pre-treatment measurement, a single random forest model trained on unperturbed clones using the full range of 1,455 features was employed. For the post-treatment measurement, where the phenotype of clones is poised to change, an ensemble of models was used. The first component was to use the predictions from the earlier time point to restrict the set of possible clone labels to those present within a radius of 350 pixels (104 μm) of the target cell (Extended Data Fig. 6f,g). For each target cell and clone class, a clone weight ($w$) score was calculated, aggregating the number and distances of cells of the given predicted clone class within the neighbourhood of the cell:

$$w_{c,l} = \sum_{i=0}^{n_{c,l}} \left( \frac{1}{0.1 d_i} \right)^2$$

where c is the target cell, l is the clone class (label), $d_i$ is the distance of a cell of the class in the previous time point within the considered radius. The second component were four random forest models trained on unperturbed clones but using different subsets of channels: (1 and 2) all channels but GFP/mScarlet, 1,134 features; (3) BFP and structural channels, 828 features; and (4) BFP barcodes only, 261 features (Supplementary Table 5). The final score for each target cell was calculated as follows:

$$s_{c,l} = w_{c,l} \sum_{rf} p_{rf}$$

where $s_{c,l}$ is the final score, $w$ is the spatial clone weight and $p_{rf}$ is the output probability of a channel subset-trained random forest model.

## Detection of hits in pooled screen

Hit calling was performed using the selected subset of 90 CellProfiler features. We defined a hit calling setting as the unique combination of compound treatment, sgRNA and clone. For each hit calling setting, perturbed cells were compared against unperturbed cells of the same clone and sgRNA (merged dataset DMSO controls wells from all plates) across all 90 features using two-sided Welch's unequal variances $t$-test implemented in the scipy library[66] v.1.9.3. Adjusted $P$ values were obtained by Bonferroni multiple testing correction. As a measure of effect size, $z$-scores were calculated. For each setting, an 'aggregating score' was calculated by counting the number of features with adjusted $P$ value below 0.05, absolute value of $z$-score above 1.5 and at least 50 treated and 100 control cells available. Settings with aggregating score of 5 or higher were considered hit candidates and inspected manually, as long as they did not involve an autofluorescent compound ($n = 30$) (Supplementary Table 8).

## Processing of images for visual inspection using quantile normalization and CLAHE

For visual inspection and the vpCells database, flat-field-corrected 16-bit TIFF images were quantile normalized and saved as eight-bit JPEG images. The quantile normalization was applied separately to each image and channel and consisted of two steps. First, intensity values for lower ($minq = 0.05$) and upper ($maxq = 0.9975$) quantile thresholds were calculated. Pixel values above the upper or below the lower threshold were set to the threshold. Second, the adjusted image was linearly rescaled to [0,1] range. For stitched images containing multiple FOVs, contrast limited adaptive histogram equalization[67] was applied before quantile normalization, using the opencv library in Python v.4.7.0. For extraction of CellProfiler features, the flatfield-corrected 16-bit TIFF files were used directly without any of the steps described in this paragraph.

## Western blot

Cell pellets were resuspended lysed for 30 min at 4 °C in RIPA buffer containing 1× Complete, EDTA-free protease inhibitor cocktail (Sigma-Aldrich) and 1× Phosphatase inhibitor (Thermo Scientific). After centrifugation for 10 min at 4 °C and 18,000$g$, the supernatant was collected and protein content was measured using a bovine serum albumin assay (Sigma). Equal amounts of protein were mixed with 4× Laemmli Sample buffer (1.0 M Tris, pH 6.8, 40% glycerol, 8% SDS, 0.2% bromophenol blue and 20% β-mercaptoethanol) and incubated for 10 min at 95 °C. Samples were loaded on an acrylamide gel together with a protein ladder (precision plus protein dual colour standards, Bio-Rad 1610394). After gel electrophoresis, proteins were transferred to an Immobilion-FL PVDF Membrane (Millipore Sigma). After blocking in TBST + 5% nonfat dry milk, the membrane was cut and incubated overnight at 4 °C with the respective primary antibodies (XPO1 antibody, Novus Biologicals, NB100-79802, 1:2,000 dilution; β-actin antibody, Abcam, ab8224, 1:1,000 dilution) in 2% milk in TBST. The next day, membranes were washed three times with TBST and then incubated for 1 h at room temperature with respective secondary antibodies in 2% milk in TBST. After washing three times with TBST, membranes were developed using Clarity Max western ECL substrate (Bio-Rad) and imaged on a Bio-Rad ChemiDoc MP.

## Statistics and reproducibility

The pooled drug screen was performed in a single experiment. Validation experiments in arrayed format were carried out with replicates and their numbers are indicated in the corresponding figures or their legends. Statistical tests were performed with GraphPad Prism v.9.0

unless described otherwise. Data distribution was assumed to be normal but this was not formally tested. No statistical methods were used to predetermine sample sizes but our sample sizes are similar to those reported in previous publications[10,18]. No data were excluded from the analyses. The experiments were not randomized. Data collection and analysis were not performed blind to the conditions of the experiments.

## Reporting summary

Further information on research design is available in the Nature Portfolio Reporting Summary linked to this article.

## Data availability

All data supporting the findings of this study are available within the paper, its supplementary information and the online resource https://vpcells.cemm.at/. All other data supporting the findings of this study are available from the corresponding author on reasonable request. Source data are provided with this paper.

## Code availability

The code for computational processing is deposited at Zenodo at https://doi.org/10.5281/zenodo.10598625 (ref. 68).

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

## Acknowledgements

We thank the members of the Kubicek laboratory and the CeMM Molecular Discovery Platform for critical input and discussions. Figures 1a and 3a were created with BioRender.com. CROPseq-Guide-Puro was a gift from C. Bock (Addgene, plasmid #86708; http://n2t.net/addgene:86708; RRID: Addgene_86708). lentiCas9-Blast was a gift from F. Zhang (Addgene, plasmid #52962; http://n2t.net/addgene:52962, RRID:Addgene_52962). A.R. was supported by a Boehringer Ingelheim Fonds PhD fellowship. A.F.R. is supported by Angelini Ventures, Rome, Italy. Research in the Kubicek laboratory has been supported by the Austrian Academy of Sciences, the European Research Council under the European Union's Horizon 2020 research and innovation programme (ERC-CoG-772437), the Austrian Science Fund (FWF) I 4768; and the Vienna Science and Technology Fund (WWTF) (10.47379/LS21010).

## Author contributions

A.R., S.K. and J.R. conceived the study. A.R., M.C., M.M., V.K., T.T., M.S. and J.R. performed experiments. J.R. analysed, curated and visualized the data with critical input from A.F.R. P.R. and J.R. built the website. A.K., A.F.R. and S.K. supervised the study. A.R., J.R. and S.K. wrote the manuscript with input from all coauthors.

## Competing interests

A.R. and S.K. have filed patent applications WO 2021/099273 A1 and WO 2023/046996 A1 related to methods described in this manuscript. S.K. is a co-founder and shareholder of Proxygen and Solgate. The remaining authors declare no competing interests.

## Additional information

**Extended data** is available for this paper at https://doi.org/10.1038/s41556-024-01407-w.

**Correspondence and requests for materials** should be addressed to Stefan Kubicek.

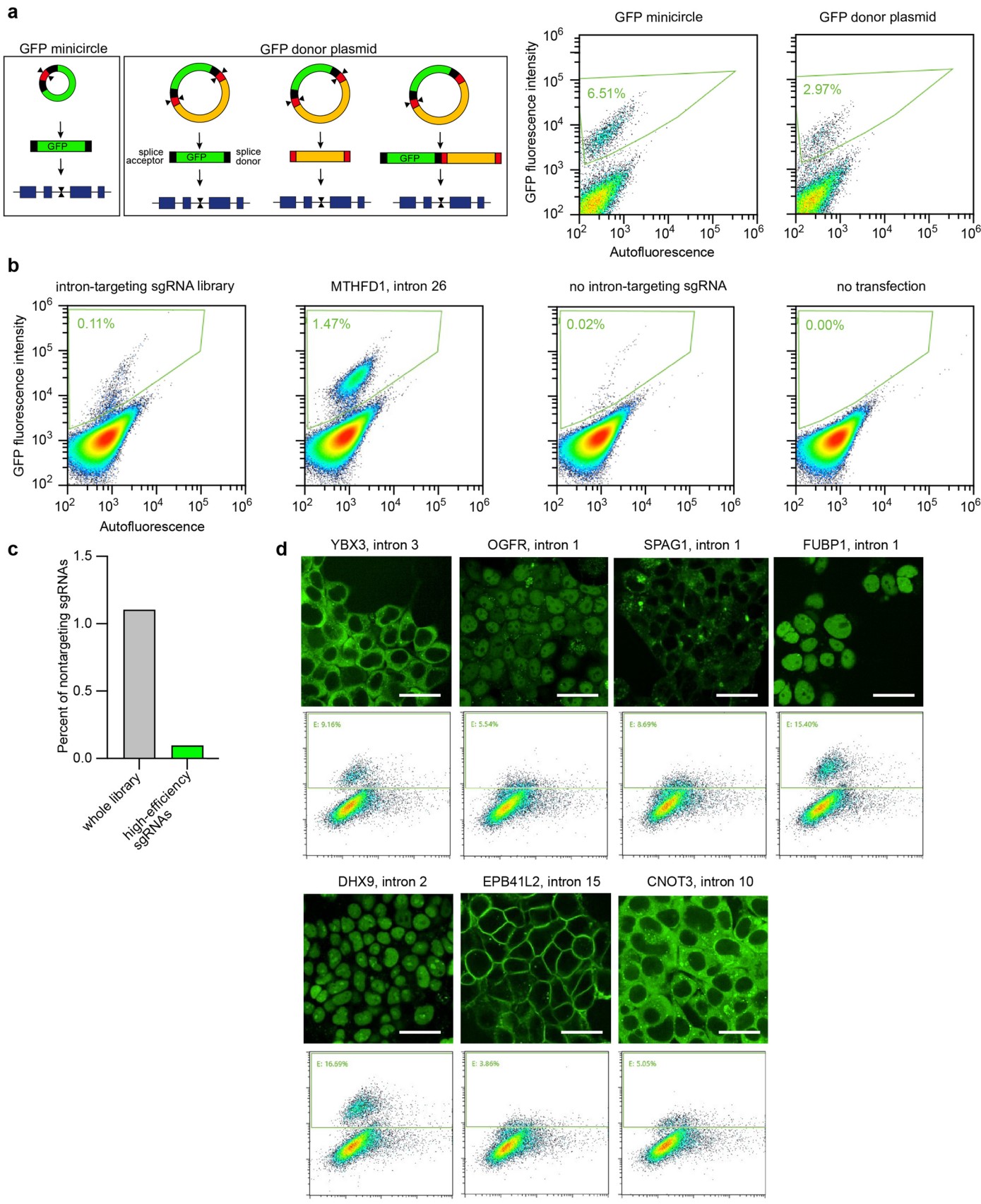

**Extended Data Fig. 1 | See next page for caption.**

**Extended Data Fig. 1 | Pooled protein tagging. a**, A minicircle containing the coding sequence of GFP is linearized at a single cut site before getting integrated at target sites. A GFP donor plasmid containing a plasmid backbone has two cut sites and undesired integrations of only the plasmid backbone or single cut plasmids are possible, resulting in lower tagging efficiencies or effecting target protein expression. Flow cytometry 2 d after transfection with intron tagging plasmids targeting MTHFD1 at intron 26 and using either a GFP minicircle or a GFP donor plasmid as a DNA donor. **b**, Flow cytometry of HEK293T cells that were transfected with intron tagging plasmids after being transduced with a genome-wide intron targeting sgRNA library, compared to HEK293T cells that were transfected with intron tagging plasmids and a plasmid expressing a sgRNA targeting intron 26 of MTHFD1, compared to HEK293T cells transfected with intron tagging plasmids, without any intron-targeting sgRNA present, compared to HEK293T that were not transfected after being transduced with a genome-wide intron targeting sgRNA library. **c**, Percentage of non-targeting sgRNAs in the set of high-efficiency sgRNAs and in the full library. **d**, Fluorescence microcopy and flow cytometry of HEK293T cells after intron tagging using selected intron targeting sgRNAs. Representative example of 1 out of 3 biologically independent experiments with similar results. Scale bars: 25 μm.

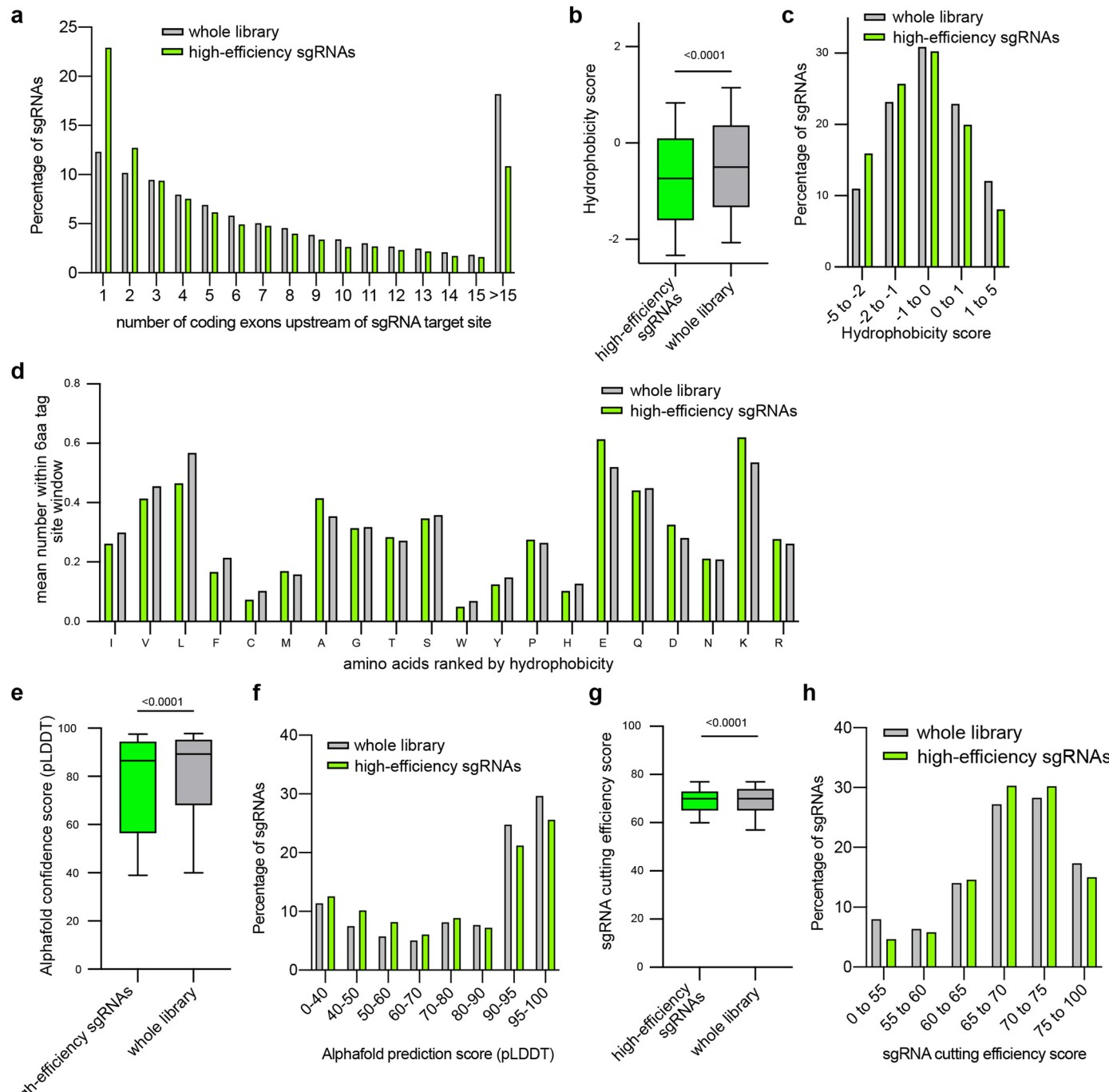

**Extended Data Fig. 2 | Contribution of sgRNA, tag site and protein features on successful tagging. a**, Percentage of high-efficiency sgRNAs and all sgRNAs in the library targeting different introns. **b**, Hydrophobicity score at tag sites of high-efficiency sgRNAs and all sgRNAs in the library. Boxes represent 25th, 50th, and 75th percentiles, and whiskers represent 10th and 90th percentiles. P value: 1.59 ×10$^{-46}$, two-sided Student's t test. n = 87,428 sgRNAs with available hydrophobicity scores for target sites, examined over 1 pooled protein tagging experiment **c**, Percentage of high-efficiency sgRNAs and all sgRNAs in the library with different hydrophobicity scores. **d**, Amino acid composition at tag sites of high-efficiency sgRNAs and all sgRNAs in the library. **e**, AlphaFold confidence score of high-efficiency sgRNAs and all sgRNAs in the library. Boxes represent 25th, 50th, and 75th percentiles, and whiskers represent 10th and 90th

percentiles. P value: 6.63 ×10$^{-21}$, two-sided Student's t test. n = 78,653 sgRNAs with available Alphafold confidence scores of target sites, examined over 1 pooled protein tagging experiment **f**, Percentage of high-efficiency sgRNAs and all sgRNAs in the library with different AlphaFold scores at the target sites. **g**, GuideScan sgRNA cutting efficiency score of high-efficiency sgRNAs and all sgRNAs in the library. Boxes represent 25th, 50th, and 75th percentiles, and whiskers represent 10th and 90th percentiles. P value: 2.82 ×10$^{-10}$, two-sided Student's t test. n = 88,700 sgRNAs with available sgRNA cutting efficiency scores, examined over 1 pooled protein tagging experiment **h**, Percentage of high-efficiency sgRNAs and all sgRNAs in the library with different cutting efficiency scores.

**a**

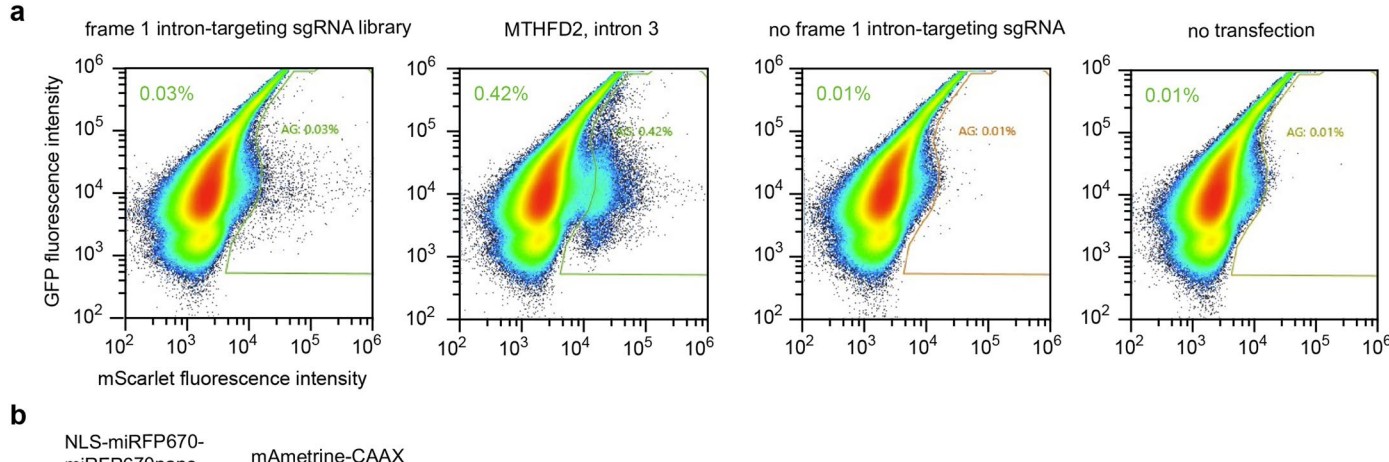

**b**

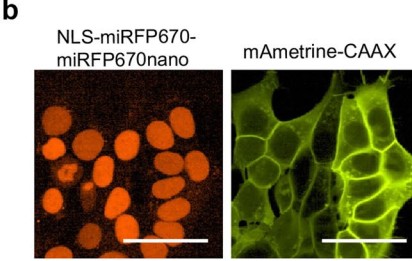

**Extended Data Fig. 3 | Generating a multicolour intron-tagged cell pool.**
**a**, Flow cytometry after the second round of intron tagging of HEK293T cells that were transfected with frame1 intron tagging plasmids after being transduced with a frame 1 genome-wide intron targeting sgRNA library, compared to HEK293T cells that were transfected with frame 1 intron tagging plasmids and a plasmid expressing a sgRNA targeting intron 3 of MTHFD2, compared to HEK293T cells transfected with frame 1 intron tagging plasmids, without any frame 1 intron-targeting sgRNA present, compared to HEK293T that were not transfected after being transduced with a frame 1 genome-wide intron targeting sgRNA library. **b**, Fluorescence microscopy images of HAP1 cells transduced with miRFP670-miRFP670nano localizing to the nucleus and mAmetrine localizing to the cell membrane. Scale bars: 25 μm.

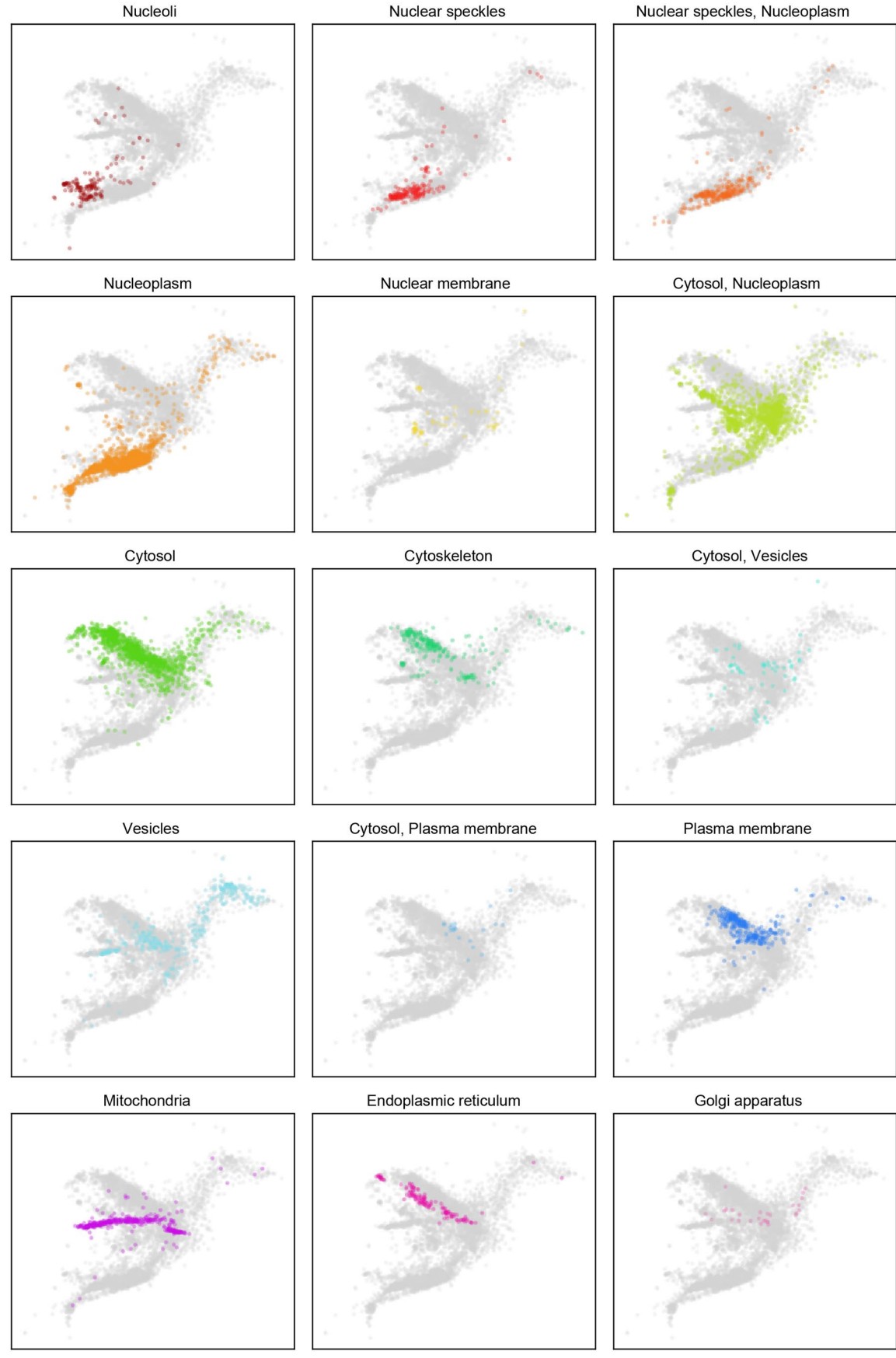

**Extended Data Fig. 4 | UMAP representation of the 15 most common annotated localizations in vpCells. a**, Each panel highlights one annotation class. Each dot represents a protein in a particular clone, averaged across all its cells.

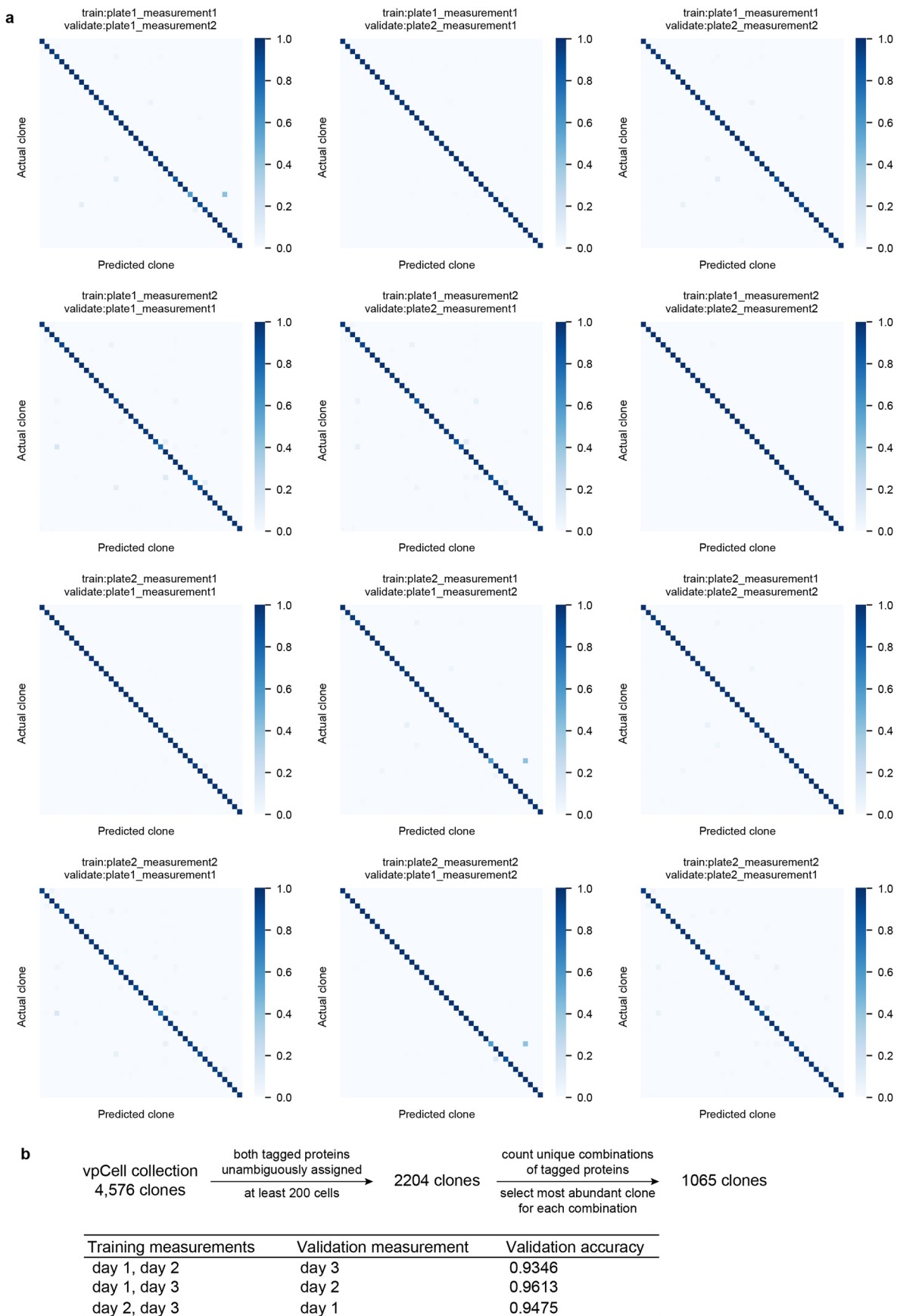

**Extended Data Fig. 5 | Training and validation of random forest models for classification of vpCell clones, arrayed format. a**, Set of 41 clones, confusion matrices, normalized by true labels (rows) for each of the 12 train-validation settings. **b**, Selection for the largest non-redundant vpCell clones; three train-validation settings and the validation accuracy scores.

| Training measurements | Validation measurement | Validation accuracy |
|---|---|---|
| day 1, day 2 | day 3 | 0.9346 |
| day 1, day 3 | day 2 | 0.9613 |
| day 2, day 3 | day 1 | 0.9475 |

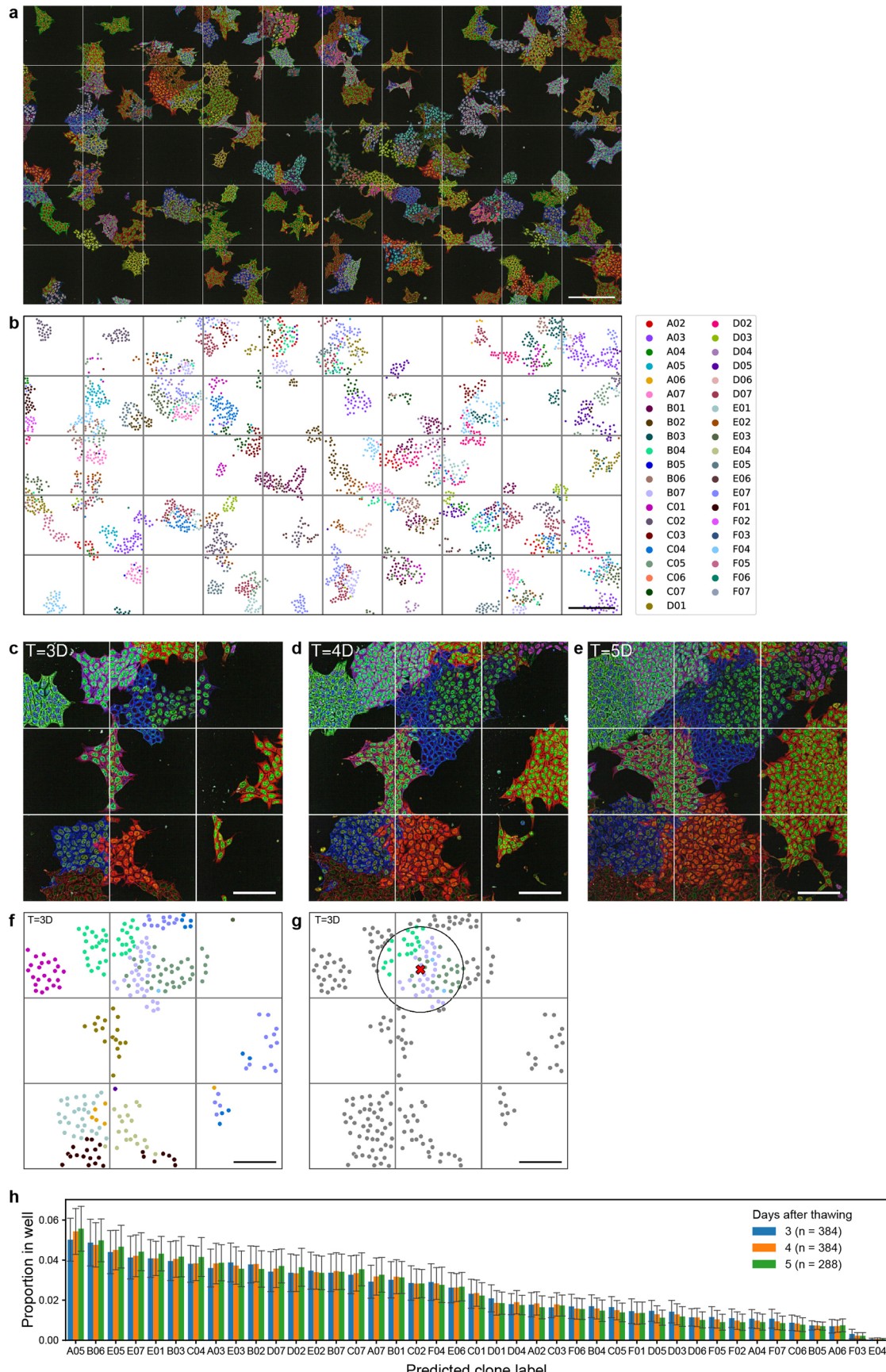

**Extended Data Fig. 6 | See next page for caption.**

**Extended Data Fig. 6 | Colony growth example, proportion of 41 clones in the pool. a**, A stitched image of a half (50 FOVs) of the imaged part of a representative well. Scale bar: 250 μm. **b**, Detected cells (dots) and predicted clones (colours) for the same 50 FOVs. Scale bar: 250 μm. **c-e**, Colony growth over a span of 2 days. The same 9 FOVs imaged at 3 time points (3, 4, 5 days after seeding). Representative images from a single experiment. Scale bars: 100 μm. **f**, Model predictions for the first time point. **g**, Clone neighbourhood within a radius of 100 μm of a given position. Scale bar: 100 μm. When predicting the clone label for a later time point (that is, T = 4D in this example) for a cell positioned at the red cross, only the clones present within the circle (in colour) would be considered. **h**, Mean proportion of each of the 41 clones in the pool after 3/4/5 days of expansion after thawing, seeding in imaging plates, and additional 56 hours of expansion. Error bars represent standard deviation, n = number of wells.

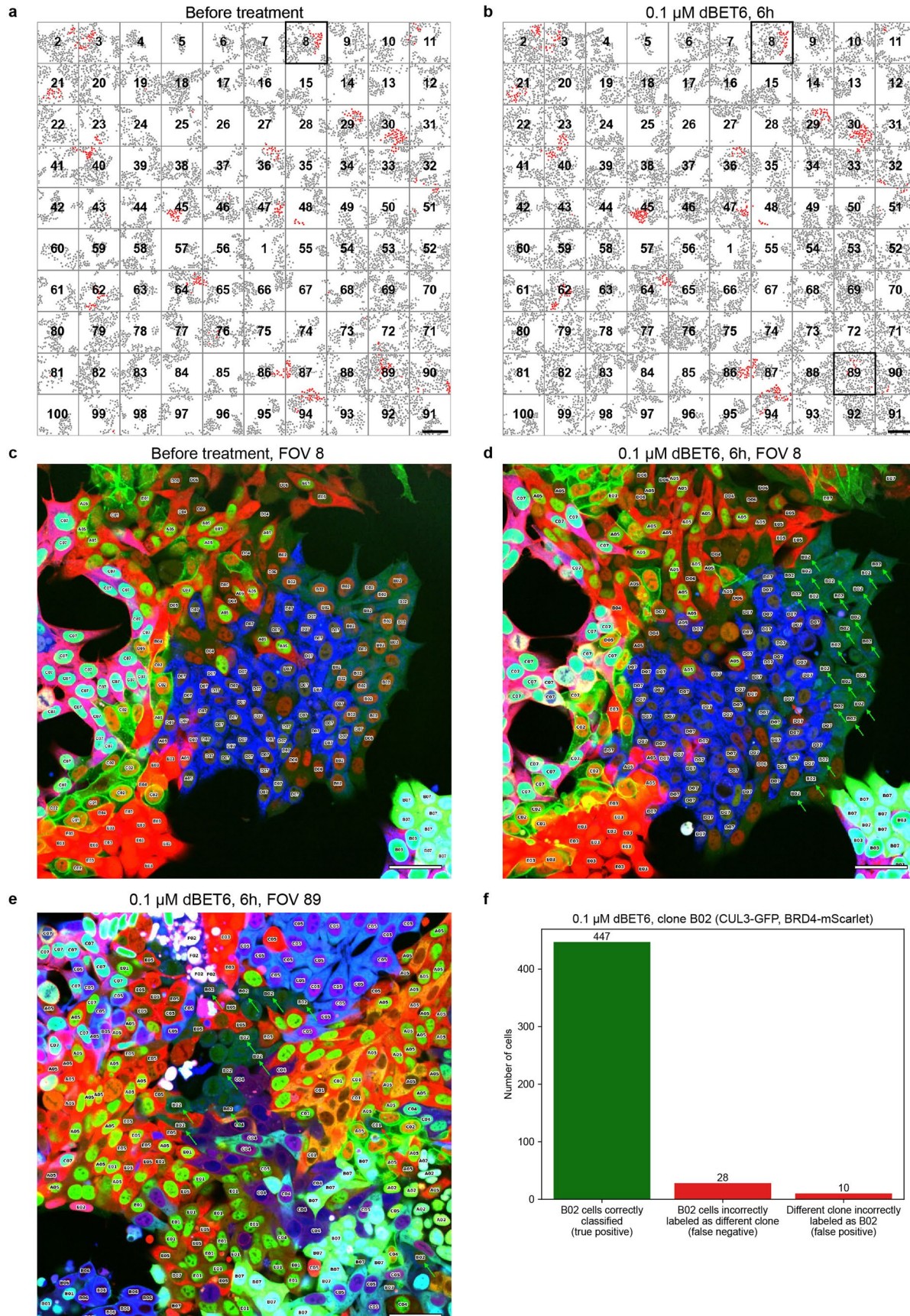

**Extended Data Fig. 7 | See next page for caption.**

**Extended Data Fig. 7 | Quantification of predictive performance of model for post-perturbation detection of clone with BRD4 protein after degradation by dBET6 treatment. a**, Detected cells in the entire well before treatment with dBET6. Cells coloured in red are predicted by the model to be the BRD4-mScarlet carrying clone B02. Field of view (FOV) numbers are in bold. Highlighted FOV 8 is shown in panel (c). Scale bar: 200 μm. **b**, The same well after treatment with dBET6. Cells coloured in red are predicted B02 by the post-perturbation model.

Highlighted FOVs 8 and 89 are shown in (d) and (e). **c**, FOV 8 with predicted clone labels, before treatment. Colour overlay of GFP (green), mScarlet (red) and BFP (blue). Scale bar: 25 μm. **d-e**, FOVs 8 and 89 with predicted clone labels, after treatment. Green arrows point to B02 cells correctly identified even after degradation of BRD4. Red arrows denote false positive or false negative predictions for B02. **f**, Counts of correctly and incorrectly classified cells of B02 clone in the entire well.

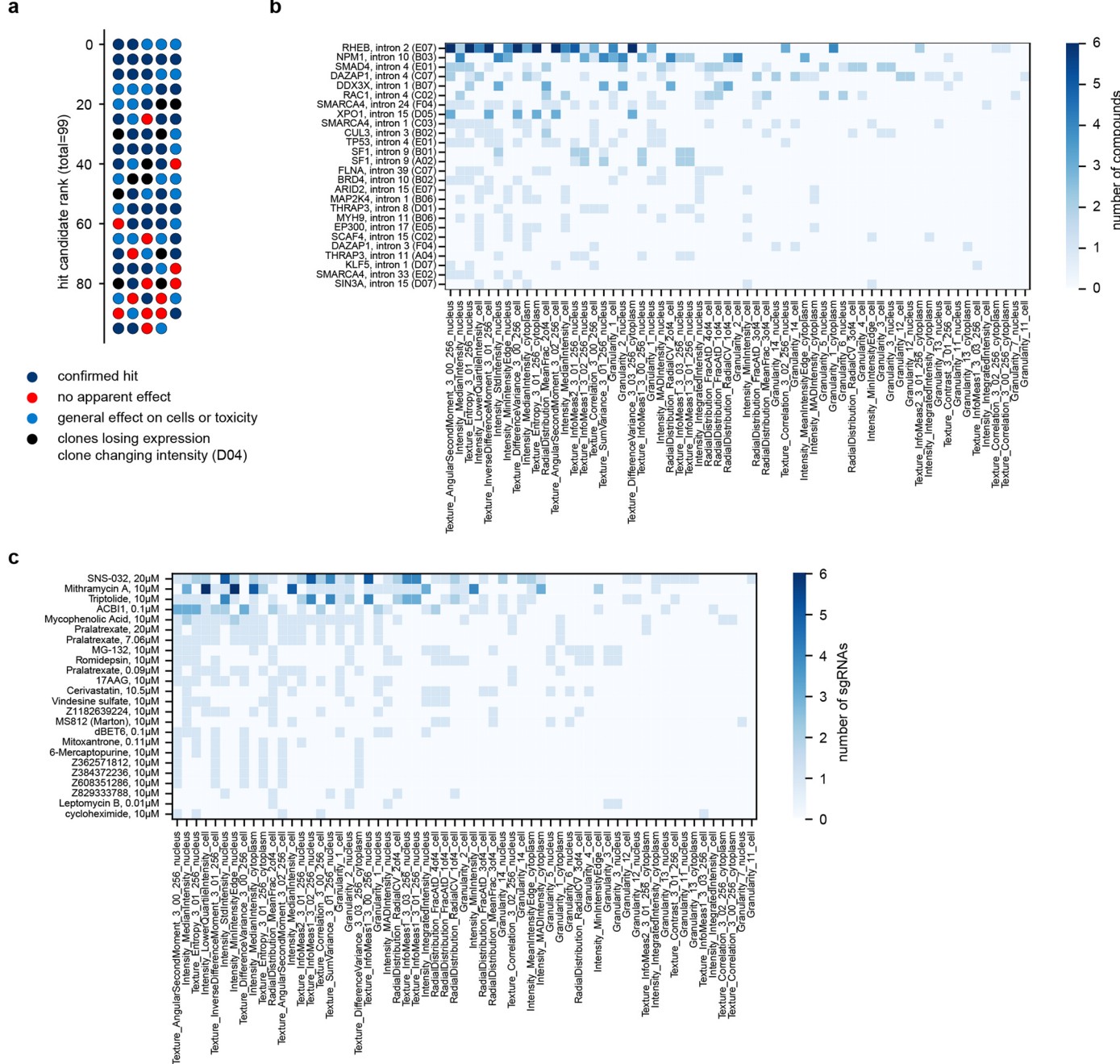

**Extended Data Fig. 8 | Hit candidates and confirmed hits from pooled screen. a**, Visual confirmation status for the 99 hit candidates after discarding autofluorescent compounds. **b**, Heatmap indicating significantly changed features for the hit proteins. Only the confirmed hits are shown. **c**, Heatmap indicating significantly changed features for the hit compounds. Only compounds from the confirmed hits are shown.

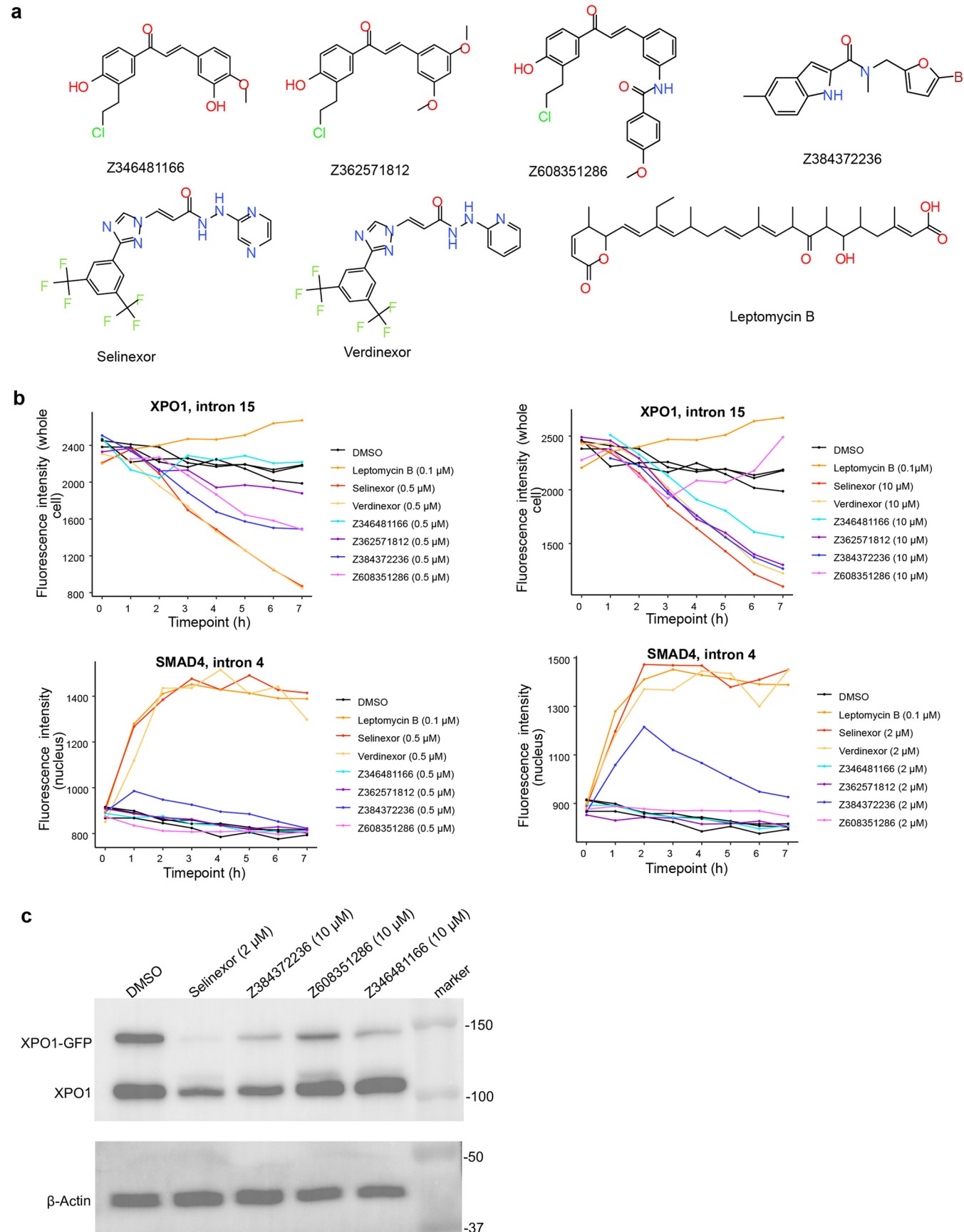

**Extended Data Fig. 9 | Inhibition of nuclear export by XPO1-binding compounds. a**, Chemical structures of hit compounds highlighted in Fig. 5a and known XPO1 inhibitors. **b**, 7 h time-lapse microscopy of XPO1 and SMAD4 in response to selected hit compounds tested at different concentrations. The same values for Leptomycin B (0.1 μM) and the DMSO controls and are shown in multiple plots of the same clones that were tested. **c**, Western Blot of XPO1 in HAP1 cells with tagged XPO1 at intron 15 treated for 16 h with the indicated compounds; n = 1 experiment.

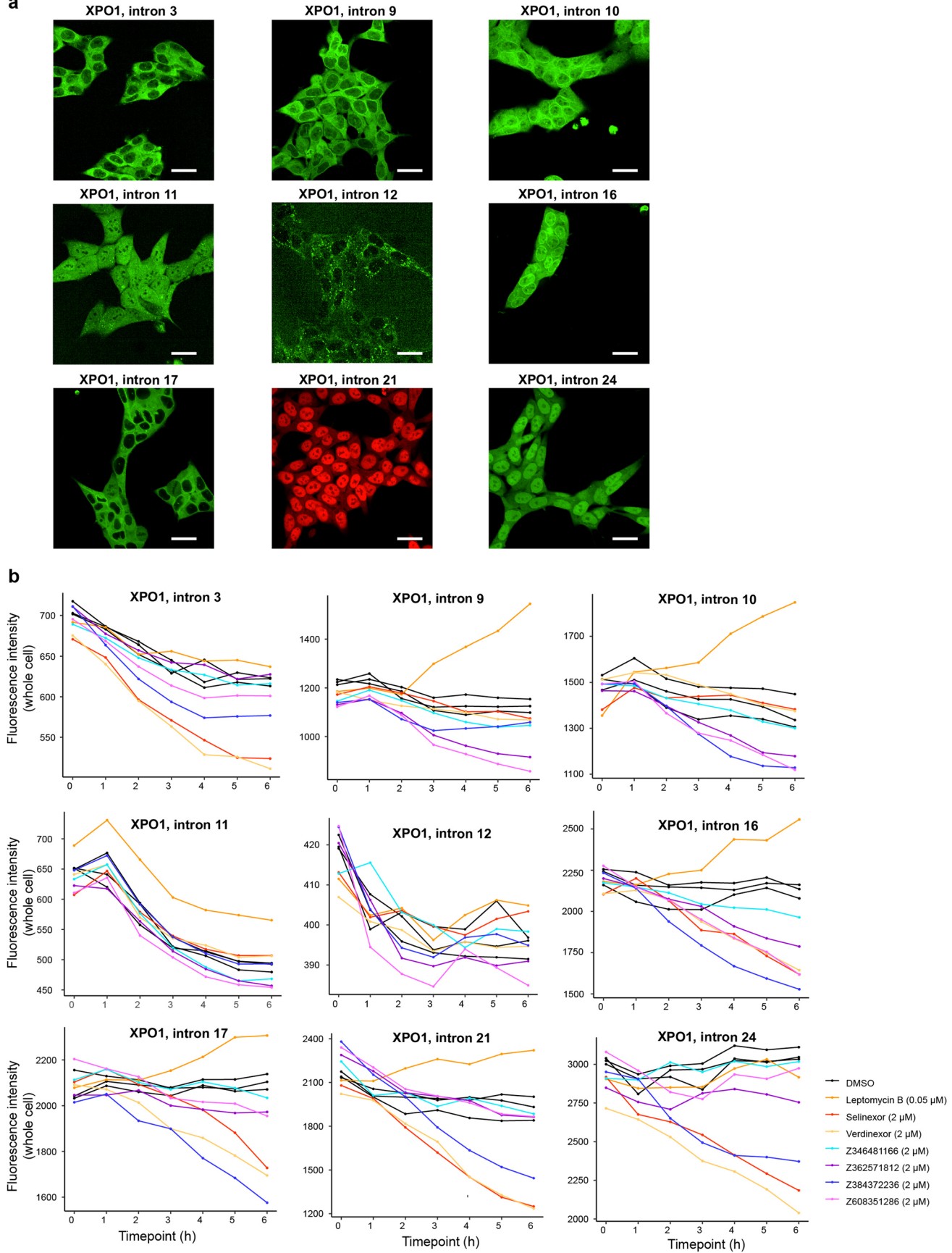

**Extended Data Fig. 10 | Degradation of XPO1 in clones tagged at different introns. a**, XPO1 tagged at different introns. The settings for brightness and contrast are different for images of each clone. Scale bars: 25 μm. **b**, 6 h time-lapse microscopy of different clonal cell lines with XPO1 tagged at different introns in response to selected hit compounds.

# Reporting Summary

## Statistics

For all statistical analyses, confirm that the following items are present in the figure legend, table legend, main text, or Methods section.

| n/a | Confirmed | |
|---|---|---|
| ☐ | ☒ | The exact sample size (*n*) for each experimental group/condition, given as a discrete number and unit of measurement |
| ☐ | ☒ | A statement on whether measurements were taken from distinct samples or whether the same sample was measured repeatedly |
| ☐ | ☒ | The statistical test(s) used AND whether they are one- or two-sided<br>*Only common tests should be described solely by name; describe more complex techniques in the Methods section.* |
| ☒ | ☐ | A description of all covariates tested |
| ☐ | ☒ | A description of any assumptions or corrections, such as tests of normality and adjustment for multiple comparisons |
| ☐ | ☒ | A full description of the statistical parameters including central tendency (e.g. means) or other basic estimates (e.g. regression coefficient) AND variation (e.g. standard deviation) or associated estimates of uncertainty (e.g. confidence intervals) |
| ☐ | ☒ | For null hypothesis testing, the test statistic (e.g. *F*, *t*, *r*) with confidence intervals, effect sizes, degrees of freedom and *P* value noted<br>*Give P values as exact values whenever suitable.* |
| ☒ | ☐ | For Bayesian analysis, information on the choice of priors and Markov chain Monte Carlo settings |
| ☒ | ☐ | For hierarchical and complex designs, identification of the appropriate level for tests and full reporting of outcomes |
| ☒ | ☐ | Estimates of effect sizes (e.g. Cohen's *d*, Pearson's *r*), indicating how they were calculated |

*Our web collection on statistics for biologists contains articles on many of the points above.*

## Software and code

Policy information about availability of computer code

| Data collection | Harmony High-Content Imaging and Analysis Software, version 6 (Perkin Elmer), Sony Cell Sorter Software 2.1.6, GuideScan sgRNA database version 1.0 |
|---|---|
| Data analysis | python 3.9.15, slurm 21.08.8, pandas 1.3.0, matplotlib-base 3.6.2, numpy 1.23.5, seaborn 0.11.2, cellpose 0.6.1, nucleAIzer-backend 0.2.1, scikit-image 0.19.1, CellProfiler 4.2.1, scikit-learn 1.1.3, umap 0.1.1, scipy 1.9.3, opencv 4.7.0, Prism 9.0<br><br>The image processing pipeline and hit calling code is deposited at https://github.com/reinisj/intron_tagging. DOI: 10.5281/zenodo.10598625 |

For manuscripts utilizing custom algorithms or software that are central to the research but not yet described in published literature, software must be made available to editors and reviewers. We strongly encourage code deposition in a community repository (e.g. GitHub). See the Nature Portfolio guidelines for submitting code & software for further information.

## Data

Policy information about availability of data

All manuscripts must include a data availability statement. This statement should provide the following information, where applicable:

- Accession codes, unique identifiers, or web links for publicly available datasets
- A description of any restrictions on data availability
- For clinical datasets or third party data, please ensure that the statement adheres to our policy

All data supporting the findings of this study are available within the article its Supplementary Information and the online resource vpcells.cemm.at.

## Research involving human participants, their data, or biological material

Policy information about studies with human participants or human data. See also policy information about sex, gender (identity/presentation), and sexual orientation and race, ethnicity and racism.

| | |
|---|---|
| Reporting on sex and gender | n/a |
| Reporting on race, ethnicity, or other socially relevant groupings | n/a |
| Population characteristics | n/a |
| Recruitment | n/a |
| Ethics oversight | n/a |

Note that full information on the approval of the study protocol must also be provided in the manuscript.

# Field-specific reporting

Please select the one below that is the best fit for your research. If you are not sure, read the appropriate sections before making your selection.

☒ Life sciences ☐ Behavioural & social sciences ☐ Ecological, evolutionary & environmental sciences

For a reference copy of the document with all sections, see nature.com/documents/nr-reporting-summary-flat.pdf

# Life sciences study design

All studies must disclose on these points even when the disclosure is negative.

| | |
|---|---|
| Sample size | No statistical methods were used to pre-determine sample sizes but our sample sizes are similar to those reported in previous publications (10, 18). Sample size was chosen to balance replication and efficiency in the experiments. The exact sample size of the associated experiments are described in the Method and Figure legends. There was in general good correlation between replicates for the different experiments justifying the chosen sample size. |
| Data exclusions | No data were excluded. |
| Replication | All experiments that were not performed in a pooled format, were successfully replicated in at least two biologically independent replicates as indicated in the figures and figure legends. The pooled protein tagging experiments using the genome-wide and focused sgRNA libraries were performed in multiple cell lines as indicated and as a single replicate per cell line. Conclusions derived from these pooled protein tagging experiments, including predicted sgRNA efficiencies were validated using individual sgRNAs in an arrayed format in experiments that were replicated three times as indicated in the corresponding figures. The pooled drug screen was performed in a single replicate. Validation experiments to confirm phenotypes observed in the pooled drug screen were successfully replicated in at least two biologically independent replicates, their numbers are indicated in the corresponding figures or their legend. |
| Randomization | All experiments were performed with molecular biological techniques. No randomization, but independent replicates were performed. |
| Blinding | Blinding was not used in this study. |

# Reporting for specific materials, systems and methods

We require information from authors about some types of materials, experimental systems and methods used in many studies. Here, indicate whether each material, system or method listed is relevant to your study. If you are not sure if a list item applies to your research, read the appropriate section before selecting a response.

## Materials & experimental systems

| n/a | Involved in the study |
|-----|----------------------|
| ☐ | ☒ Antibodies |
| ☐ | ☒ Eukaryotic cell lines |
| ☒ | ☐ Palaeontology and archaeology |
| ☒ | ☐ Animals and other organisms |
| ☒ | ☐ Clinical data |
| ☒ | ☐ Dual use research of concern |
| ☒ | ☐ Plants |

## Methods

| n/a | Involved in the study |
|-----|----------------------|
| ☒ | ☐ ChIP-seq |
| ☐ | ☒ Flow cytometry |
| ☒ | ☐ MRI-based neuroimaging |

# Antibodies

| | |
|---|---|
| Antibodies used | XPO1 (CRM1) polyclonal antibody: Novus Biologicals, NB100-79802<br>β-actin monoclonal antibody: Abcam, ab8224, clone number: mAbcam 8224 |
| Validation | XPO1 (CRM1) polyclonal antibody: Validated using other independent antibodies and with different biological strategies (https://www.novusbio.com/products/crm1-antibody_nb100-79802#datasheet)<br>β-actin monoclonal antibody: Frequently used loading control antibody with >470 references (https://www.abcam.com/products/primary-antibodies/beta-actin-antibody-mabcam-8224-loading-control-ab8224.html) |

# Eukaryotic cell lines

Policy information about cell lines and Sex and Gender in Research

| | |
|---|---|
| Cell line source(s) | HAP1: Haplogen (now Horizon Discovery, C631), HEK293T: ATCC CRL-3216 |
| Authentication | Cell lines were not authenticated. |
| Mycoplasma contamination | Cell lines were tested every two months for mycoplasma contamination using PCR - with negative results. |
| Commonly misidentified lines<br>(See ICLAC register) | No commonly misidentified cell lines were used in this study. |

# Flow Cytometry

## Plots

Confirm that:

☒ The axis labels state the marker and fluorochrome used (e.g. CD4-FITC).

☒ The axis scales are clearly visible. Include numbers along axes only for bottom left plot of group (a 'group' is an analysis of identical markers).

☒ All plots are contour plots with outliers or pseudocolor plots.

☒ A numerical value for number of cells or percentage (with statistics) is provided.

## Methodology

| | |
|---|---|
| Sample preparation | Adherent HAP1 or HEK293T cells were detached using trypsin and centrifuged (280 g, 5 min). Cell pellets were resuspended in PBS with 5% FBS before being transferred to polystyrene round bottom tubes with a cell strainer cap. |
| Instrument | Sony SH800S, model type: LE-SH800SZDCPL |
| Software | Sony Cell Sorter Software |
| Cell population abundance | Purity of post-sort fractions were analyzed by flow cytometry and live-cell imaging. |
| Gating strategy | The forward and sideward scatter was used to select the starting cell populations. Single cells were then selected using FSC-A and FSC-H and finally, cells were selected based on their green or red fluorescence as shown in the figures of the manuscript. |

☒ Tick this box to confirm that a figure exemplifying the gating strategy is provided in the Supplementary Information.

