## [Peer Review File · Nature Cell Biology]

Peer Review Information

Journal: Nature Cell Biology

Manuscript Title: Pooled multicolor tagging for visualizing subcellular protein dynamics

Corresponding author name(s): Dr Stefan Kubicek

Editorial Notes:

Reviewer Comments & Decisions:

Decision Letter, initial version:

*Please delete the link to your author homepage if you wish to forward this email to co-authors.

Dear Dr Kubicek,

Your manuscript, "Pooled multicolor tagging for visualizing subcellular proteome dynamics", has now been seen by 3 referees, who are experts in subcellular protein mapping (referee 1); CRISPR and imaging(referee 2); and high throughput cellular imaging (referee 3). As you will see from their comments (attached below) they find this work of potential interest, but have raised substantial concerns, which in our view would need to be addressed with considerable revisions before we can consider publication in Nature Cell Biology.

Nature Cell Biology editors discuss the referee reports in detail within the editorial team, including the chief editor, to identify key referee points that should be addressed with priority, and requests that are overruled as being beyond the scope of the current study. To guide the scope of the revisions, I have listed these points below. We are committed to providing a fair and constructive peer-review process, so please feel free to contact me if you would like to discuss any of the referee comments further.

In particular, it would be essential to:

- A) Validate the algorithm through more detailed analysis of the current clones (all Reviewers)
- B) Clarify concerns about methods for quantification, classification, and analysis (all Reviewers)
- C) Compare with other methods regarding efficiency, robustness, and applicability (reviewer #2)
- D) All other referee concerns pertaining to strengthening existing data, providing controls, methodological details, clarifications and textual changes, should also be addressed.
- E) Finally please pay close attention to our guidelines on statistical and methodological reporting (listed below) as failure to do so may delay the reconsideration of the revised manuscript. In particular please provide:

- a Supplementary Figure including unprocessed images of all gels/blots in the form of a multi-page pdf file. Please ensure that blots/gels are labeled and the sections presented in the figures are clearly indicated.
- a Supplementary Table including all numerical source data in Excel format, with data for different figures provided as different sheets within a single Excel file. The file should include source data giving rise to graphical representations and statistical descriptions in the paper and for all instances where the figures present representative experiments of multiple independent repeats, the source data of all repeats should be provided.

We would be happy to consider a revised manuscript that would satisfactorily address these points, unless a similar paper is published elsewhere, or is accepted for publication in Nature Cell Biology in the meantime.

- ensure that it conforms to our format instructions and publication policies (see below and www.nature.com/nature/authors/).
- provide a point-by-point rebuttal to the full referee reports verbatim, as provided at the end of this letter.
- provide the completed Editorial Policy Checklist (found here <https://www.nature.com/authors/policies/Policy.pdf>), and Reporting Summary (found here <https://www.nature.com/authors/policies/ReportingSummary.pdf>). This is essential for reconsideration of the manuscript and these documents will be available to editors and referees in the event of peer review. For more information see <http://www.nature.com/authors/policies/availability.html> or contact me.

Nature Cell Biology is committed to improving transparency in authorship. As part of our efforts in this

direction, we are now requesting that all authors identified as 'corresponding author' on published papers create and link their Open Researcher and Contributor Identifier (ORCID) with their account on the Manuscript Tracking System (MTS), prior to acceptance. ORCID helps the scientific community achieve unambiguous attribution of all scholarly contributions. You can create and link your ORCID from the home page of the MTS by clicking on 'Modify my Springer Nature account'. For more information please visit www.springernature.com/orcid.

[Redacted]

We would like to receive a revised submission within six months. We would be happy to consider a revision even after this timeframe, however if the resubmission deadline is missed and the paper is eventually published, the submission date will be the date when the revised manuscript was received.

We hope that you will find our referees' comments, and editorial guidance helpful. Please do not hesitate to contact me if there is anything you would like to discuss.

Best wishes,

Daryl

Daryl Jason Verzosa David, PhD

Senior Editor, Nature Cell Biology
Nature Portfolio

Heidelberger Platz 3, 14197 Berlin, Germany
Email: daryl.david@nature.com
ORCID: <https://orcid.org/0000-0002-9253-4805>

Reviewers' Comments:

Reviewer #1:

Remarks to the Author:

This manuscript describes the use of pooled intro-tagging to generate libraries of fluorescently labeled cell lines for imaging-based high-throughput screening discovery. A particular innovation is the use of dual-tagged cells and the image of the tagged proteins themselves to match cells in the pool with their individually measured identity. This approach conveniently avoided the need for additional identifiable "barcodes" for the labeled cells during screening. On the other hand, although the manuscript claims the potential to "democratize subcellular localization studies" and "enable the rapid generation, sharing and characterization... in different cell models", the creation of the cell pool itself is still somewhat laborious, requiring generating clonal cell lines for identifying the tagged targets before

pooling. While it is true that the process is more efficient than purely arrayed format because of the pooled tagging, the above mentioned claimed needs to be toned down. Overall, this manuscript has demonstrated a nice progress in the technological platform, and the reviewer could recommend its publication in Nature Cell Biology after addressing the following points:

1. Characterization of factors affecting sgRNA efficiency. Although the comparison of hydrophobicity scores (Fig 1e) and AlphaFold confidence scores (Fig 1f) between high efficiency sgRNA and the whole library shows statistical significance (likely because of the large sample number), it is hard to believe that the tiny differences between the median values have any biological significance, particularly considering the large standard deviation. Even considering the amino acid frequency statistics in Extended Data Fig 1d, the difference is so small that for the choice of target intron, the contribution of surrounding amino acid hydrophobicity would be vastly outweighed by other factors considered in this manuscript (e.g. order of intron) as well as those not considered (target sequence dependence of Cas9-sgRNA cutting efficiency, etc.). The claim that insertion to unstructured domains is more efficient than structured domains also potentially contradicts the hydrophobicity claim, as the former tend to have more hydrophilic residues. The reviewer thus suggests removing the aforementioned figure panels and associated discussions. Otherwise, the readers could be misled to miss a large number of legitimate high-efficiency target introns based on the hydrophobicity and structuredness criteria, at least if the manuscript cannot show significance level similar to that for protein expression level (Fig 1d). The manuscript also needs to explain why the comparison is high-efficiency sgRNA against the whole library instead of high-efficiency sgRNA against the rest of the library (low efficiency sgRNA).

2. Statistics with extremely low sample number. Fig 1c and 5g have a sample number of 2. In this case, the reviewer requests showing only the two raw data points, at most the mean value, and removing the bar and standard deviations, because standard deviation with a sample number of 2 is meaningless.

3. Drop out of cells from the pooled library. Although the pooled library is created by mixing equal numbers of clonal cells, the distribution of cell number in the library, as shown in Fig 3i, shows clear bias, with one clone nearly completely lost. If this dropout is the result of the short 4-day expansion and the 56 hr after plating as described in the methods, it is somewhat concerning. Either the intron tagging is perturbing the proliferation or viability of certain clones, or the process introduces random or non-random bias. Both cases could introduce uncertainty for the generation of other libraries and potentially affect some screening applications. The manuscript could benefit from showing the identity of the most over- and under-represented clones in the pool and provide even speculative explanations, which will alleviate such concerns regarding method generality.

4. Limitations of cell matching before and after screening. Although the use of the tagged protein images themselves to determine cell identity is a nice innovation to simplify the library construction, this approach has the problem if the target protein has substantial cell-to-cell heterogeneity of subcellular localization, and, more importantly, if the subcellular localization change substantially during screening perturbation. The manuscript nicely benchmarked the cell identification accuracy from the pool before perturbation. However, for the before and after matching, the manuscript did not show quantitative accuracy, except for mentioning that likely only one of the two tagged proteins in a cell will be affected by the perturbation. The fidelity in this case also needs to be quantified, as least when one of the two tagged proteins has dramatically changed subcellular distribution (as the ones in this manuscript).

5. Identification of mitotic cells. Based on the described method, the cell segmentation method in this manuscript will discard all mitotic cells because they do not have the proper shape of the nucleus and nuclear-to-cytoplasm ratio. Please clarify this point.

6. Minor point: Fig 2b, what are the overlaps between the pools?

Reviewer #2:

Remarks to the Author:

Summary. In this manuscript, Reicher et. al. demonstrated a method for generating cell pools with two rounds of CRISPR-based intron-FP-tagging. The authors subsequently sub-cloned the pool and generated three libraries of clonal cell lines (vpCell clones) that have a pair of proteins tagged by combination of GFP and mScarlet separately, and a BFP barcode. The authors later developed computational methods that can distinguish cell clones from high-content imaging data, based on the difference of tagged proteins' and BFP barcodes' subcellular localization signatures. The authors used this method to screen a library of proliferation-inhibiting compounds. From the screen, the author discovered that several uncharacterized compounds induced a decrease in the protein level of XPO1, a protein that is responsible for nuclear protein export. Further characterization of these compounds identified that Z384372236 can inhibit/degrade XPO1 through mechanisms independent of SMAD4, a common upstream pathway that a couple of FDA approved XPO1 inhibitors targets.

Novelty. There are two major novelties of this work: 1) The authors demonstrated that the dual-intron-tagged clonal cells can be identified and tracked by a non-destructive imaging-computation-based method, solely based on their fluorescent protein localization pattern and expression levels. This offers new opportunities that one could potentially use these libraries to track cell states and protein localization changes without the need to sequence the sgRNA barcodes. 2) The authors characterized a compound (Z384372236) that can potentially inhibit/degrade XPO1 through mechanisms independent of SMAD4 perturbation.

However, the CRISPR-based tagging strategy is not totally new. The authors have published this strategy for inserting GFP to introns of target genes (Reicher et al. Genome Research 2020). This work expands their previous method to two color cell pool generation via two-round sequential tagging, which expands the pool diversity and image-based clone identification.

Data & methodology presentation. The data and methods are generally presented clearly and the approaches are valid for the claims, though further improvements to clarify certain critical technical aspects are needed (see in the "Suggested improvements").

Appropriate use of statistics and treatment of uncertainties. This research used statistics properly in most parts. Please see the "Suggested improvements" part for parts that we think can be improved.

Conclusions: robustness, validity, reliability. The conclusions from the research are valid. Please see the "Suggested improvements" part for parts that we think can be improved.

Suggested improvements: experiments, data for possible revision.

Major points

a. The authors haven't validated their imaging-based, clone-recognition algorithm outside of the pool of 41 selected clones (Fig. 3c and related text on page 7), while they constructed over 4000 clonal lines. This seems a wasted opportunity to not test this algorithm on the bigger pool. Additionally, considering that the algorithm was trained on the 41 clones, it is questionable how generalizable the algorithm is. While it might be possible to recognize the thousands of clones by just looking at the two subcellular localization signatures, the author should demonstrate this, which stands out as a major innovation from this study. The authors should also estimate the upper limit of the number of cell clones that this algorithm can distinguish. In general, this reviewer finds the method for imaging-based cell identify confusing. It is a bit surprising that the authors can distinguish each protein's

tagging solely based on the localization pattern and expression. The authors should provide additional text and description to fully explain how this part works.

b. Related to point a, we are wondering if the computational algorithm is superior, in terms of capacity, to alternative “traditional methods” for the classification of the FP-protein localization.

- As an example of an alternative method, one could identify clones based on if the tagged protein is localized in nucleolus, nucleoplasm, cytosol, mitochondria, or ER (5 of the most robust properties shown in Fig 3c in the manuscript). With a pair of tagged proteins, the above method in theory has an upper limit of identifying $(2^5)^2 \sim 1000$ clones. This number can be further extended by the three digits of BFP barcodes the author integrated, to ~ 8000 . It is reasonable that the real-world capability is lower due to various reasons, but the validation on a set of 41 clones in this study is not enough to convince their method is superior.
- Even if the computation method is not as good as the manual classification, the authors should try to explore its capacity limit and discuss its advantages and disadvantages.

c. The vpCell library can be a valuable resource for other researchers to study protein localization, we would like to see deeper characterization of the clones in the larger vpCell library (covering ~ 1000 genes, Figure 2), outside the selected 41 cell clones (covering only 61 genes). For example, the authors can take the protein hits from 41-pool (described in Figure 4), and pull out the related clones from the larger pool, harboring tags on the same protein but with a different secondary protein tag, to see if it has similar effect. This could validate that screen results are not clonal dependent, or specific to certain intron insertion (if different intron targeting labels the protein in the original screen vs additional validation).

d. It remains a question regarding the broad utility of the approach. How many proteins can be tagged after two rounds of CRISPR transduction? Given the low efficiency of tagging for each protein, it is likely the approach can only generate a cell pool with random combination of proteins tagged out of a large sgRNA pool, not exactly the desired proteins. This ‘shotgun’ approach to generate randomly tagged proteins in clonal cell lines may be useful in certain contexts but less in others. Because the current method necessitates a tedious and long procedure and cannot generate a comprehensive pool to visualize any or many desired genes, it remains a major concern if the method will be broadly used by other labs.

Minor points

- Fig 1e and 1f. The difference does not look significant, even though they are due to the large sample size. The authors can plot this differently, e.g. in an overlapping histogram to further clarify their claims.
- Related to the last one. The author might want to show some positive controls for Fig 1e and 1f. For example, for 3e, how residues on known structures’ surface vs inside distribute along the same axis?
- Fig 2b. The author might want to elaborate why there is no overlapping of the proteins that are covered by three individual libraries.
- Fig 2f and extended fig 3. I suppose the multidimensional quantification in these UMAPs are the cell-profilers’s output features. I am not sure if UMAPs are the proper way, and if necessary, to de-dimensionize this type of data, as UMAPs are normally for RNAseq data. We are wondering if a PCA would serve a similar purpose in this case, without bringing in the ambiguity of the UMAP method.
- The website detailing the sgRNA sequences and clones (<https://vpcells.cemmm.at>) is not available to the public (reviewers included).
- Fig 4b. I suppose the point here is to show the cells from the same clone did not move 100 microns apart during the assay. The figure should indicate how big is 100 microns in the images. Also see the next comment.
- Most images do not have scale bars. Please add them.

h. The authors might want to at least discuss the potentials of this system to capture/screen protein dynamics. The screening example in the manuscript focused mostly on "static features" like decrease of a certain protein. It would be interesting to discuss if the authors think this system can be used for screens on more subtle dynamic features, like rate of degradation, relocalization, or even non-monotonic behavior.

References: The manuscript gives appropriate credit to previous work.

Clarity and context: The abstract, introduction and conclusion are generally appropriate. For potential improvement in this area, see minor point h.

Reviewer #3:

Remarks to the Author:

The authors present a pooled method for visualizing (and quantifying) proteins at scale and in living cells. The barcoding relies on the localization patterns of two tagged proteins per cell (vs prior in situ sequencing based methods).

They do a nice test where individual cell clones are isolated and tested for which tagged proteins they express, for the 1,178 protein collection. They used manual annotation to 12 compartments (allowing up to 2 compartments to be labeled for each protein) and assessed localization against previous annotation for each protein.

Other strengths of the paper are that it is a neat idea and they are addressing an important problem. They seem to discover a nice biological finding (I'm not qualified to judge that particular biological domain).

However, overall, the descriptions were not clear enough to understand the method readily, and the validation experiments and the discovery experiments seem to have been done on quite small-scale protein libraries (61 proteins), nowhere near proteome-scale nor genome-wide as stated and possibly not even in the mode that is recommended for use (they tested by manually combining isolated clones which is extremely tedious compared to the proposed use case of adding pooled genetic reagents to cells before a pooled experiment). I found this all quite misleading and frustrating.

MAJOR

- In the unperturbed state, >26% of proteins did not match the localization seen for endogenous protein in the Human Protein Atlas, indicating that tagging causes undesirable changes in localization (as also seen in OpenCell project). This needs more emphasis in the paper.

- I don't understand why "proteome-scale" and "genome-wide" is used in the abstract when the majority of validation steps are done on much smaller pools, e.g. 61 proteins or 1,178 proteins (and those were created but not actually tested/used in pooled format); even the sgRNA library they started with covers only 14k genes which is nowhere near genome-wide. This is misleading and should be changed to reflect the size of experiments that were actually validated. It is absolutely not clear from the data presented that thousands of proteins could be distinctly identified under perturbation conditions using the method, as errors in the method would not be expected to scale linearly. IIUC "approximately 2,500 different proteins were tagged with either GFP or mScarlet" so is that the maximum number (and those were not deconvoluted by imaging, but instead by sequencing, so not really helpful either for validating the method). As far as I can tell, 61 proteins are the maximum that are actually deconvoluted from a pool that is created the way that's recommended, by adding a pool

of genetic reagents to cells. The heading "large sets of proteins" is above the section with 61 proteins, which doesn't count as large in my view.

- it took an unacceptably long time to understand from the abstract what is being tagged and what is being visualized. At first I thought the entire proteome worth of proteins was being directly tagged and visualized, but then an sgRNA library is mentioned which triggered my thinking CRISPR is knocking proteins down and then some other protein is being monitored? And are the two tagged proteins per cell only for barcoding or also for reading out protein dynamics/localization? I couldn't answer these Qs from the abstract alone which is a pretty big problem. Overall, it needs to be more clear what is the (a) perturbation, (b) barcode strategy, (c) readout. Because here proteins could be all 3 of those, it led to confusion about which is feasible to be proteome-scale.

- The authors state: "We hypothesized that tagging two different proteins with different colors in each cell might create enough diversity to discriminate hundreds of clones by using the combination of the two localization patterns and intensity values as visual barcodes." But if the whole point is to monitor changes in localization in response to perturbation, then the barcoding fails for exactly those perturbations. They solve this by imaging cells before/after perturbation - this concept should be included in the abstract and much earlier in the main text.

- it was also not clear from the abstract whether individual clones are isolated, selected, and pooled, as opposed to more common (and convenient) methods where a pool of genetic reagents are introduced to a cell population.

- IIUC the experiment that does the former - separating individual clones - was only for validation and not representing how the method should be used. This should be clarified. IIUC "assembled a new cell pool from 41 vpCell clones covering 61 cancer-associated proteins" means that the individual clones were selected and manually pooled together which is not how the method is recommended to be used. This matters because the natural numbers of each clone in a pool will be quite different if cells are grown then combined, vs a pool being created directly from genetic reagents being added.

- accuracy of 97% in identifying clones is great, as is their analysis of manually assembled pools lacking a protein, which was then correctly not found in the pool, but that test was done in a pool of only 41 clones covering 61 proteins.

- the analysis that finds some expected gene-compound pairings is not quantitative and thus not convincing. One could imagine finding a few such relationships by chance in any list of a few dozen genes/compounds. It's easy in a high throughput experiment to look at a list of relationships identified and find literature support for them. The authors could either find a set of annotations for the gene-compound relationships and quantify how many were identified vs missed, or they could scramble the list and try to find literature support for the scrambled list (which would need a researcher who is kept in the dark about the goal so they give it a good try).

- the concept here is unclear "only a small fraction of sgRNAs leads to successful protein tagging, we observed an enrichment of approximately 5% of all sgRNAs in the library" - enrichment relative to what? Does this mean only 5% of sgRNAs (representing 17% of genes targeted) were successful? Or 5% were above some threshold (which was decided how)? This is crucial for understanding the rest of the paper.

- It should be noted that I got lost at points about whether an experiment was being done a certain way as a validation step or whether it's how it was intended to be used. There may therefore be errors in my above comments! But I will leave them because they point is that an educated reader who cares about this field didn't grasp what was going on and therefore the manuscript needs substantial improvement.

MINOR

- it was really nice to see this comment, it's quite valuable/important: "We took care to randomize clone positions between replicate (imaging plate), in order to prevent possible overfitting to well position during model training."

- the reference to "tagging efficiencies" means more definition. 18.7% of cells exposed to the reagent ended up with a visibly fluorescent GFP tag? or of cells that had an integration event?

- paper needs a read through for typos, like the two here "effects which we could validation not only in cell pools but also individual tagged cells lines." There were maybe a half dozen others in the main text.
- the authors should elaborate on the scale: a single multi well plate yielded 50 individual cells for most of the 41 clones. That I would think it a bare minimum - is this in each well of the 96 well plate or across the entire plate? The wording makes it unclear. This will allow assessing where is the tradeoff vs just testing each protein individually in arrayed format.

Methods should be written concisely, but should contain all elements necessary to allow interpretation and replication of the results. As a guideline, Methods sections typically do not exceed 3,000 words. The Methods should be divided into subsections listing reagents and techniques. When citing previous methods, accurate references should be provided and any alterations should be noted. Information must be provided about: antibody dilutions, company names, catalogue numbers and clone numbers for monoclonal antibodies; sequences of RNAi and cDNA probes/primers or company names and catalogue numbers if reagents are commercial; cell line names, sources and information on cell line identity and authentication. Animal studies and experiments involving human subjects must be reported in detail, identifying the committees approving the protocols. For studies involving human subjects/samples, a statement must be included confirming that informed consent was obtained. Statistical analyses and information on the reproducibility of experimental results should be provided in a section titled "Statistics and Reproducibility".

All Nature Cell Biology manuscripts submitted on or after March 21 2016 must include a Data availability statement at the end of the Methods section. For Springer Nature policies on data availability see <http://www.nature.com/authors/policies/availability.html>; for more information on this particular policy see <http://www.nature.com/authors/policies/data/data-availability-statements-data-citations.pdf>. The Data availability statement should include:

- Accession codes for primary datasets (generated during the study under consideration and designated as "primary accessions") and secondary datasets (published datasets reanalysed during the study under consideration, designated as "referenced accessions"). For primary accessions data should be made public to coincide with publication of the manuscript. A list of data types for which submission to community-endorsed public repositories is mandated (including sequence, structure, microarray, deep sequencing data) can be found here <http://www.nature.com/authors/policies/availability.html#data>.
- Unique identifiers (accession codes, DOIs or other unique persistent identifier) and hyperlinks for datasets deposited in an approved repository, but for which data deposition is not mandated (see here

for details <http://www.nature.com/sdata/data-policies/repositories>).

- At a minimum, please include a statement confirming that all relevant data are available from the authors, and/or are included with the manuscript (e.g. as source data or supplementary information), listing which data are included (e.g. by figure panels and data types) and mentioning any restrictions on availability.
- If a dataset has a Digital Object Identifier (DOI) as its unique identifier, we strongly encourage including this in the Reference list and citing the dataset in the Methods.

We recommend that you upload the step-by-step protocols used in this manuscript to the Protocol Exchange. More details can be found at www.nature.com/protocolexchange/about.

All imaging data should be accompanied by scale bars, which should be defined in the legend. Cropped images of gels/blots are acceptable, but need to be accompanied by size markers, and to retain visible background signal within the linear range (i.e. should not be saturated). The boundaries of panels with low background have to be demarked with black lines. Splicing of panels should only be considered if unavoidable, and must be clearly marked on the figure, and noted in the legend with a statement on whether the samples were obtained and processed simultaneously. Quantitative comparisons between samples on different gels/blots are discouraged; if this is unavoidable, it should only be performed for samples derived from the same experiment with gels/blots were processed in parallel, which needs to be stated in the legend.

- For line art, graphs, charts and schematics we prefer Adobe Illustrator (.AI), Encapsulated PostScript (.EPS) or Portable Document Format (.PDF). Files should be saved or exported as such directly from the application in which they were made, to allow us to restyle them according to our journal house style.
- We accept PowerPoint (.PPT) files if they are fully editable. However, please refrain from adding PowerPoint graphical effects to objects, as this results in them outputting poor quality raster art. Text

used for PowerPoint figures should be Helvetica (preferred) or Arial.

Unprocessed scans of all key data generated through electrophoretic separation techniques need to be presented in a supplementary figure that should be labelled and numbered as the final supplementary figure, and should be mentioned in every relevant figure legend. This figure does not count towards the total number of figures and is the only figure that can be displayed over multiple pages, but should be provided as a single file, in PDF or TIFF format. Data in this figure can be displayed in a relatively informal style, but size markers and the figures panels corresponding to the presented data

must be indicated.

The total number of Supplementary Figures (not including the “unprocessed scans” Supplementary Figure) should not exceed the number of main display items (figures and/or tables (see our Guide to Authors and March 2012 editorial <http://www.nature.com/ncb/authors/submit/index.html#suppinfo>; <http://www.nature.com/ncb/journal/v14/n3/index.html#ed>). No restrictions apply to Supplementary Tables or Videos, but we advise authors to be selective in including supplemental data.

GUIDELINES FOR EXPERIMENTAL AND STATISTICAL REPORTING

REPORTING REQUIREMENTS – To improve the quality of methods and statistics reporting in our papers we have recently revised the reporting checklist we introduced in 2013. We are now asking all life sciences authors to complete two items: an Editorial Policy Checklist (found here <https://www.nature.com/authors/policies/Policy.pdf>) that verifies compliance with all required editorial policies and a reporting summary (found here <https://www.nature.com/authors/policies/ReportingSummary.pdf>) that collects information on experimental design and reagents. These documents are available to referees to aid the evaluation of the manuscript. Please note that these forms are dynamic ‘smart pdfs’ and must therefore be downloaded and completed in Adobe Reader. We will then flatten them for ease of use by the reviewers. If you would like to reference the guidance text as you complete the template, please access these flattened versions at <http://www.nature.com/authors/policies/availability.html>.

We strongly recommend the presentation of source data for graphical and statistical analyses as a separate Supplementary Table, and request that source data for all independent repeats are provided when representative experiments of multiple independent repeats, or averages of two independent experiments are presented. This supplementary table should be in Excel format, with data for different figures provided as different sheets within a single Excel file. It should be labelled and numbered as one of the supplementary tables, titled “Statistics Source Data”, and mentioned in all relevant figure legends.

Author Rebuttal to Initial comments

RESPONSE TO REFEREES

Nature Cell Biology submission NCB-A51759

Pooled multicolor tagging for visualizing subcellular proteome dynamics

Andreas Reicher^{1,*}, Jiří Reiniš^{1,*}, Maria Ciobanu¹, Pavel Růžička¹, Monika Malik¹, Marton Siklos¹, Viktoriia Kartysh¹, Tatjana Tomek¹, Anna Koren¹, André F. Rendeiro¹, Stefan Kubicek¹

¹CeMM Research Center for Molecular Medicine of the Austrian Academy of Sciences, Lazarettgasse 14, 1090 Vienna, Austria.

*these authors contributed equally

We thank all reviewers and the editors for the constructive comments and interest in our manuscript. In this revised version, we have now comprehensively addressed these comments and added new experiments and analysis to further strengthen the manuscript.

The major changes and additions in the current version are the following:

- We applied our approach for purely computational assignment of clone identities to the entire non-redundant vpCell collection of more than 1,000 clones, and found a validation accuracy of more than 93%. These results both validate the algorithm and contribute to a more detailed analysis of the entire current clone collection.
- We compared the algorithm for computational clone identification to manual annotation for an entire well treated with a BRD4 degrader resulting in a complete loss of signal in one channel. The results confirm the high predictive performance (F1 score of 0.96) of the algorithm even after drastic perturbation-induced changes and thereby validate the efficiency and robustness of the method for classification and quantification compared to manual annotation.
- We added new experimental data showing effects of XPO1 inhibitors in all nine additional XPO1-tagged clones in our collection. The conservation of effects in all clones that robustly express XPO1 clones further validates the approach.

In addition, we have now added additional biological replicates, addressed all requests regarding text changes and editorial requirements. Please find our detailed point-by-point responses below.

Reviewer #1:

Remarks to the Author:

This manuscript describes the use of pooled intro-tagging to generate libraries of fluorescently labeled cell lines for imaging-based high-throughput screening discovery. A particular innovation is the use of dual-tagged cells and the image of the tagged proteins themselves to match cells in the pool with their individually measured identity. This approach conveniently avoided the need for additional identifiable “barcodes” for the labeled cells during screening. On the other hand, although the manuscript claims the potential to “democratize subcellular localization studies” and “enable the rapid generation, sharing and characterization... in different cell models”, the creation of the cell pool itself is still somewhat laborious, requiring generating clonal cell lines for identifying the tagged targets before pooling. While it is true that the process is more efficient than purely arrayed format because of the pooled tagging, the above mentioned claimed needs to be toned down. Overall, this manuscript has demonstrated a nice progress in the technological platform, and the reviewer could recommend its publication in Nature Cell Biology after addressing the following points:

We thank the reviewer for these positive comments. While it is true that the method also is highly efficient for the generation of cell pools, the comment regarding sharing particularly refers to already existing and characterized cell pools. Imagine a researcher wanting to test a novel perturbation in the cancer cell pool we use in Fig. 4 and 5. We would simply ship a single vial of that cell pool, the researcher could expand it and perform imaging and use our validated algorithm to call changes. Doing this with antibodies would require purchasing 61 antibodies. Doing the same experiment with arrayed collections like OpenCell would require shipping and expanding 61 individual clones. From that perspective, we believe that our approach truly has advantages in empowering the community.

We have now rewritten the manuscript to tone down these claims and make these aspects clearer.

1. Characterization of factors affecting sgRNA efficiency. Although the comparison of hydrophobicity scores (Fig 1e) and AlphaFold confidence scores (Fig 1f) between high efficiency sgRNA and the whole library shows statistical significance (likely because of the large sample number), it is hard to believe that the tiny differences between the median values have any biological significance, particularly considering the large standard deviation. Even considering the amino acid frequency statistics in Extended Data Fig 1d, the difference is so small that for the choice of target intron, the contribution of surrounding amino acid hydrophobicity would be vastly outweighed by other factors considered in this manuscript (e.g. order of intron) as well as those not considered (target sequence dependence of Cas9-sgRNA cutting efficiency, etc.). The claim that insertion to unstructured domains is more efficient than structured domains also potentially contradicts the hydrophobicity claim, as the former tend to have more hydrophilic residues. The reviewer thus suggests removing the aforementioned figure panels and associated discussions. Otherwise, the readers could be misled to miss a large number of legitimate high-efficiency target introns based on the hydrophobicity and structuredness criteria, at least if the manuscript cannot show significance level similar to that for protein expression level (Fig 1d). The manuscript also needs to explain why the comparison is high-efficiency sgRNA against the whole library instead of high-efficiency sgRNA against the rest of the library (low efficiency sgRNA).

The differences of the hydrophobicity scores and AlphaFold confidence scores are indeed very small and might be outweighed by more other factors, such as expression level and intron position, as indicated by the referee. Still, we believe there is a trend with biological significance that might be of interest for many readers. We therefore have moved these two panels to the Extended Data instead of removing them. In the text, we put more emphasis on the effect of gene expression on successful tagging and less on all other factors. In the Extended Data Fig. 2 we added a representation of that in histogram format, as suggested by reviewer 3.

We would also like to clarify that the more efficient tagging in unstructured regions does not potentially contradict the hydrophobicity claim, since we showed that tag sites with lower hydrophobicity scores (=more hydrophilic residues) are more efficiently tagged, which is in agreement with unstructured regions usually being more hydrophilic and surface exposed.

Additionally, we would like to thank the referee for pointing out other parameters that are potentially predictive of successful tagging, such as the cutting efficiency score of the sgRNAs. We have now analyzed whether also these scores are predictive of successful tagging, but we only see very minor differences and no clear trend in the histogram representation between high efficiency sgRNAs and the whole library. A very likely cause is that most sgRNAs in the library have very high cutting efficiency scores, since that parameter was considered when designing the sgRNA libraries (Extended Data Fig. 2g).

We agree with the referee that strong claims about predicting sgRNA efficiency might mislead readers to miss a large number of legitimate high-efficiency sgRNAs. For this reason we avoided calling everything that is not a high efficiency sgRNA a low efficiency sgRNA and instead did all comparisons between high-efficiency sgRNAs and the whole library. It is also common for sgRNA enrichment analysis in CRISPR knockout screens to compare enriched or depleted sgRNAs to the whole library, which is another reason why the comparison was done between these two groups.

2. Statistics with extremely low sample number. Fig 1c and 5g have a sample number of 2. In this case, the reviewer requests showing only the two raw data points, at most the mean value, and removing the bar and standard deviations, because standard deviation with a sample number of 2 is meaningless.

We now performed a third replicate for both figures. Therefore, we have kept the representation as a bar graph with standard deviation, but we also show all individual data points.

3. Drop out of cells from the pooled library. Although the pooled library is created by mixing equal numbers of clonal cells, the distribution of cell number in the library, as shown in Fig 3i, shows clear bias, with one clone nearly completely lost. If this dropout is the result of the short 4-day expansion and the 56 hr after plating as described in the methods, it is somewhat concerning. Either the intron tagging is perturbing the proliferation or viability of certain clones, or the process introduces random or non-random bias. Both cases could introduce uncertainty for the generation of other libraries and potentially affect some screening applications. The manuscript could benefit from showing the identity of the most over- and under-represented clones in the pool and provide even speculative explanations, which will alleviate such concerns regarding method generality.

Thank you for this comment. Uneven clonal representation is indeed a matter of concern. To identify whether over- and underrepresented clones were due to imperfect cell counting/initial viability when generating cell pools, or due to changed growth properties, we looked again at the numbers of cells per predicted clone in our data. The nature of the tagged proteins in the most abundant clone (MSH6/PPP2R1A) and least abundant clones (CHD4/MAX; RPL22/GTF2I) do not enable us to speculate on any biological causes underlying their altered abundance.

We therefore aimed to analyze the clonal growth properties from the pooled imaging data performed at multiple time points. In the drug screen with pool of 41 clones, before seeding in imaging plates and the 56 h expansion, the frozen pool was thawed and expanded for either 3, 4, or 5 days. Any severe differences in clone viability should lead to a corresponding considerable alteration of the relative fraction of that clone in the pool over these 3 days. For each clone, we calculated its proportion in the pool (based on the pre-treatment measurement only), comparing the 3 possible durations of expansion before seeding in imaging plates.

While we observed a mild increase in the proportion of the more abundant clones over time and a similar decrease in the less abundant, this change is not dramatic enough to explain the overall imbalance of the pool (Extended data Fig. 8f). Therefore, these data indicate that indeed an imbalance of clones, inaccurate cell counting, or clone-to-clone viability differences when generating the cell are causative for the uneven clone distribution indicating the disproportion is the result of initial differences when generating the pool.

4. Limitations of cell matching before and after screening. Although the use of the tagged protein images themselves to determine cell identity is a nice innovation to simplify the library construction, this approach has the problem if the target protein has substantial cell-to-cell heterogeneity of subcellular localization, and, more importantly, if the subcellular localization change substantially during screening perturbation. The manuscript nicely benchmarked the cell identification accuracy from the pool before perturbation. However, for the before and after matching, the manuscript did not show quantitative accuracy, except for mentioning that likely only one of the two tagged proteins in a cell will be affected by the perturbation. The fidelity in this case also needs to be quantified, as least when one of the two tagged proteins has dramatically changed subcellular distribution (as the ones in this manuscript).

In the manuscript, we benchmark this effect with some of the most dramatic changes caused by chemical degraders that lead to near-complete loss of the protein in one channel. Figure 4c shows a representative field of view with degradation of BRD4 by dBET6 treatment, including the clone labels predicted by the model. Importantly, for the large majority of cells of the BRD4 clone, the model outputs the correct clone label "B02" in the post-perturbation setting.

To further quantify the fidelity of clone identification following severe perturbation, we have manually identified and labeled all cells of the BRD4 clone after dBET6 treatment in the entire well (100 fields of view). Out of a total of 14,181 cells in the well, the model correctly identified 447 cells of the BRD4 clone (true positives). 27 cells were mislabeled as a different clone (false positives), and 10 cells of a different clone were wrongly labeled as the BRD4 clone (false negatives). This corresponds to a precision, recall, and F1 score of 0.9781, 0.9430, and 0.9603, respectively.

We have now added Extended Data Fig. 9 to describe these results and updated the corresponding section in the revised manuscript.

5. Identification of mitotic cells. Based on the described method, the cell segmentation method in this manuscript will discard all mitotic cells because they do not have the proper shape of the nucleus and nuclear-to-cytoplasm ratio. Please clarify this point.

Indeed, the filtering part of the segmentation pipeline generally removes mitotic cells due to their higher nuclear-to-cytoplasmic ratio (see new Extended Data Fig. 5e). In the pooled screen with 41 clones, we would only expect around 10 mitotic cells per clone per perturbation on average, too few to draw any robust conclusions. Therefore, we decided to keep mitotic cells excluded from our analysis. We have now clarified this point in the revised version.

6. Minor point: Fig 2b, what are the overlaps between the pools?

We have added numbers to the overlapping areas of the Venn diagram.

Reviewer #2:

Remarks to the Author:

Summary. In this manuscript, Reicher et. al. demonstrated a method for generating cell pools with two rounds of CRISPR-based intron-FP-tagging. The authors subsequently sub-cloned the pool and generated three libraries of clonal cell lines (vpCell clones) that have a pair of proteins tagged by combination of GFP and mScarlet separately, and a BFP barcode. The authors later developed computational methods that can distinguish cell clones from high-content imaging data, based on the difference of tagged proteins' and BFP barcodes' subcellular localization signatures. The authors used this method to screen a library of proliferation-inhibiting compounds. From the screen, the author discovered that several uncharacterized compounds induced a decrease in the protein level of XPO1, a protein that is responsible for nuclear protein export. Further characterization of these compounds identified that Z384372236 can inhibit/degrade XPO1 through mechanisms independent of SMAD4, a common upstream pathway that a couple of FDA approved XPO1 inhibitors targets.

Novelty. There are two major novelties of this work: 1) The authors demonstrated that the dual-intron-tagged clonal cells can be identified and tracked by a non-destructive imaging-computation-based method, solely based on their fluorescent protein localization pattern and expression levels. This offers new opportunities that one could potentially use these libraries to track cell states and protein localization changes without the need to sequence the sgRNA barcodes. 2) The authors characterized a compound (Z384372236) that can potentially inhibit/degrade XPO1 through mechanisms independent of SMAD4 perturbation. However, the CRISPR-based tagging strategy is not totally new. The authors have published this strategy for inserting GFP to introns of target genes (Reicher et al. Genome Research 2020). This work expands their previous method to two color cell pool generation via two-round sequential tagging, which expands the pool diversity and image-based clone identification.

We thank the referee for the accurate summary of our method and discoveries.

Data & methodology presentation. The data and methods are generally presented clearly and the approaches are valid for the claims, though further improvements to clarify certain critical technical aspects are needed (see in the "Suggested improvements").

Thank you for this assessment, we have now further improved the presentation of data and methods in the revised version.

Appropriate use of statistics and treatment of uncertainties. This research used statistics properly in most parts. Please see the "Suggested improvements" part for parts that we think can be improved.

Thank you for this assessment, we have now further improved the statistical analyses, added additional biological replicates and all source data in the revised version.

Conclusions: robustness, validity, reliability. The conclusions from the research are valid. Please see the "Suggested improvements" part for parts that we think can be improved.

Thank you for this assessment, in the revised version we have further improved these aspects.

Suggested improvements: experiments, data for possible revision.

Major points

a. The authors haven't validated their imaging-based, clone-recognition algorithm outside of the pool of 41 selected clones (Fig. 3c and related text on page 7), while they constructed over 4000 clonal lines. This seems a wasted opportunity to not test this algorithm on the bigger pool. Additionally, considering that the algorithm was trained on the 41 clones, it is questionable how generalizable the algorithm is. While it might be possible to recognize the thousands of clones by just looking at the two subcellular localization signatures, the author should demonstrate this, which stands out as a major innovation from this study. The authors should also estimate the upper limit of the number of cell clones that this algorithm can distinguish. In general, this reviewer finds the method for imaging-based cell identify confusing. It is a bit surprising that the authors can distinguish each protein's tagging solely based on the localization pattern and expression. The authors should provide additional text and description to fully explain how this part works.

Thank you for this comment. To estimate the generalizability of our clone recognition algorithm, we trained a classifier on a large set of all non-redundant clones from our vpCell collection.

Out of 4,582 clones in total, 3,525 have unambiguously assigned sgRNAs in both frames, with 1,522 unique combinations of tagged proteins. Further filtering for clones with more than 200 captured high-quality cells (minimal threshold we deem necessary for robust training and evaluation), 2,204 clones and 1,065 unique protein combinations remain. Selecting the clone with the most available cells for each of the 1,065 combinations, we trained the random forest classifier.

For each clone, we possess 3 measurements, corresponding to 3 independent imaging timepoints on consecutive days. Therefore, we trained three models, each using 2 timepoints as training data and the single remaining for validation. In all settings, we obtained over 93% prediction accuracy (new Extended Data Fig. 6b). This result indicates that the upper limit of our clone-recognition algorithm indeed is well in the range of over 1,000 clones.

We added a new paragraph in the Results section to describe this analysis in the updated manuscript. Supplementary Table 4 lists the vpCell clones included in the set of 1,065.

As a clarification, we do not distinguish the tagging of single proteins, but rather the combination of localization pattern and expression of multiple proteins to discriminate clones, giving us much more discriminatory power. The full explanation of the model is included in the Online Methods.

b. Related to point a, we are wondering if the computational algorithm is superior, in terms of capacity, to alternative "traditional methods" for the classification of the FP-protein localization.

As an example of an alternative method, one could identify clones based on if the tagged protein is localized in nucleolus, nucleoplasm, cytosol, mitochondria, or ER (5 of the most robust properties shown in Fig 3c in the manuscript). With a pair of tagged proteins, the above method in theory has an upper limit of identifying $(2^5)^2 \sim 1000$ clones. This number can be further extended by the three digits of BFP barcodes the author integrated, to ~ 8000 . It is

reasonable that the real-world capability is lower due to various reasons, but the validation on a set of 41 clones in this study is not enough to convince their method is superior.

Even if the computation method is not as good as the manual classification, the authors should try to explore its capacity limit and discuss its advantages and disadvantages.

Thank you for this suggestion. We hope our response to the previous point has demonstrated that the capacity of our computational approach is beyond 1,000 clones. Extracting features with CellProfiler and training a machine learning model on top for cell classification is a rather traditional approach that has been successfully used for more than 15 years. A random forest model finds the features best distinguishing the clones automatically and within a very reasonable training time (minutes to single-digit hours in our case). This is a large advantage over any methods that rely on manual steps.

Moreover, if our understanding of the proposed alternative method is correct, to reach the limit of 1,000 clones, all possible combinations of the 5 localizations *within a single channel* are used and are available in the set of clones to choose from. This means there would, for example, have to be a protein localizing to nucleoli, mitochondria, and the ER at the same time. However, the distribution of protein localizations in the cell in general and in vpCell collection is characterized by the large fraction of cytoplasmic and nuclear proteins, which severely limits our ability to choose an arbitrary selected combination of localizations (shown in Fig. 2c which we believe this comment relates to). While it might be possible to specifically engineer the combinations from the very beginning by performing targeted tagging in arrayed format, this would remove the key advantage of our pooled tagging approach, which is its high throughput.

c. The vpCell library can be a valuable resource for other researchers to study protein localization, we would like to see deeper characterization of the clones in the larger vpCell library (covering ~1000 genes, Figure 2), outside the selected 41 cell clones (covering only 61 genes).

For example, the authors can take the protein hits from 41-pool (described in Figure 4) and pull out the related clones from the larger pool, harboring tags on the same protein but with a different secondary protein tag, to see if it has similar effect. This could validate that screen results are not clonal dependent, or specific to certain intron insertion (if different intron targeting labels the protein in the original screen vs additional validation).

Thank you for this suggestion, which highlights the benefits of having the entire vpCell collection available for rapid hit validation.

We would like to point out that also within the 41 clone pool, same proteins were represented by multiple independent clones (3x SMARCA4, 2x SF1 2x, 2x DAZAP1), showing consistent effects irrespective of the clone.

For hit validation, we had included data for four independent XPO1 clones tagged at intron 15 (main figures) and introns 15, 21, 24 (Extended Data Fig. 8 in original submission). In addition to keeping these data (now Extended Data Fig. 11), we have now expanded all other available XPO1 clones contained in our collection (Extended Data Fig. 12), where we observe high consistency of perturbation effects irrespective of tagging location.

d. It remains a question regarding the broad utility of the approach. How many proteins can be tagged after two rounds of CRISPR transduction? Given the low efficiency of tagging for each protein, it is likely the approach can only generate a cell pool with random combination

of proteins tagged out of a large sgRNA pool, not exactly the desired proteins. This 'shotgun' approach to generate randomly tagged proteins in clonal cell lines may be useful in certain contexts but less in others. Because the current method necessitates a tedious and long procedure and cannot generate a comprehensive pool to visualize any or many desired genes, it remains a major concern if the method will be broadly used by other labs.

It is correct that we generated pools with a random combination of tagged proteins with our genome-scale intron targeting sgRNA libraries. The goal of those tagging experiments was to generate pools with the highest possible diversity by initially targeting as many genes as possible, without initially preselecting sgRNAs only for highly expressed or well annotated genes, but at the expense of a lower number of sgRNAs per gene in our genome-scale library. Because of this decision, our collection also includes many previously uncharacterized or poorly annotated proteins such as C9orf40 or C16orf87.

In contrast to those pooled tagging experiments with large libraries, we also used a smaller and more focused library targeting 287 genes associated with cancer and from the generated pool we could isolate clones with 170 of those genes tagged. This shows that the method can also be used to generate pools containing more comprehensive pools containing only desired genes and no unrelated, random proteins. The limited overlap between the genome-scale sgRNA pool and the cancer-focused library shows, that more focused approaches including more sgRNAs per targeted gene, FACs sorting closer to the background fluorescence levels and potentially (not shown) brighter fluorophores will be able to further expand the scope of the method.

Minor points

a. Fig 1e and 1f. The difference does not look significant, even though they are due to the large sample size. The authors can plot this differently, e.g. in an overlapping histogram to further clarify their claims.

We have now moved these planes to the supplement and have plotted them differently as suggested.

b. Related to the last one. The author might want to show some positive controls for Fig 1e and 1f. For example, for 3e, how residues on known structures' surface vs inside distribute along the same axis?

We now discuss the factors of hydrophobicity and unstructured domains as minor contributors to tagging efficiency and therefore have not further investigated these points.

c. Fig 2b. The author might want to elaborate why there is no overlapping of the proteins that are covered by three individual libraries.

We have added the numbers of overlapping proteins to Figure 2b and expanded the discussion on this point.

d. Fig 2f and extended fig 3. I suppose the multidimensional quantification in these UMAPs are the cell-profilers's output features. I am not sure if UMAPs are the proper way, and if

necessary, to de-dimensionize this type of data, as UMAPs are normally for RNAseq data. We are wondering if a PCA would serve a similar purpose in this case, without bringing in the ambiguity of the UMAP method.

UMAP is a method suitable for dimensionality reduction irrespective of the origin of high-dimensional data. Unlike PCA, UMAP is a nonlinear transformation. Therefore, it should per se not be restricted to RNAseq data. As an example, the Human Protein Atlas consortium uses an interactive UMAP to visualize and explore their large-scale subcellular protein localization dataset:

<https://www.proteinatlas.org/humanproteome/subcellular/location+umap>

For completeness, we have performed PCA on the same dataset shown below, but still prefer to keep the UMAP representations in the manuscript.

PCA representation of vpCells proteins calculated from CellProfiler features. Each dot represents a protein in a particular clone, averaged across all its cells. Only the 15 most common annotated localizations are shown.

PCA representation of vpCells proteins calculated from CellProfiler features. Each dot represents a protein in a particular clone, averaged across all its cells. Each subplot highlights one of the 15 most common annotated localizations.

e. The website detailing the sgRNA sequences and clones (<https://vpcells.cemm.at>) is not available to the public (reviewers included).

We apologize that the website was not available to reviewers. We had provided a reviewer login in the submission form during the initial submission:

<https://proud-wave-017631503.1.azurestaticapps.net/.auth/basicAuth/login>

password: Qq9Jr%v\$@ANJc

Please contact the editor to let us know in case you are still unable to access the website. Thank you.

f. Fig 4b. I suppose the point here is to show the cells from the same clone did not move 100 microns apart during the assay. The figure should indicate how big is 100 microns in the images. Also see the next comment.

Thank you, we have added a 100 μm scale bar to these images.

g. Most images do not have scale bars. Please add them.

We have added scale bars to all images.

h. The authors might want to at least discuss the potentials of this system to capture/screen protein dynamics. The screening example in the manuscript focused mostly on “static features” like decrease of a certain protein. It would be interesting to discuss if the authors think this system can be used for screens on more subtle dynamic features, like rate of degradation, relocalization, or even non-monotonic behavior.

Thank you for this suggestion. Indeed, we think that kinetic studies are one of the major advantages of the system. We actually show in Fig. 5d and e that the method is ideally suited for this purpose. There we analyze time-dependance of XPO1 degradation and nuclear accumulation of its cargo. In the revised version, we have put more emphasis on this point.

References: The manuscript gives appropriate credit to previous work.

Thank you.

Clarity and context: The abstract, introduction and conclusion are generally appropriate. For potential improvement in this area, see minor point h.

Thank you, we now specifically mention the time-resolved characterization of perturbation-induced changes in the abstract.

Reviewer #3:**Remarks to the Author:**

The authors present a pooled method for visualizing (and quantifying) proteins at scale and in living cells. The barcoding relies on the localization patterns of two tagged proteins per cell (vs prior in situ sequencing based methods).

They do a nice test where individual cell clones are isolated and tested for which tagged proteins they express, for the 1,178 protein collection. They used manual annotation to 12 compartments (allowing up to 2 compartments to be labeled for each protein) and assessed localization against previous annotation for each protein.

Other strengths of the paper are that it is a neat idea and they are addressing an important problem. They seem to discover a nice biological finding (I'm not qualified to judge that particular biological domain).

Thank you for these positive comments.

However, overall, the descriptions were not clear enough to understand the method readily, and the validation experiments and the discovery experiments seem to have been done on quite small-scale protein libraries (61 proteins), nowhere near proteome-scale nor genome-wide as stated and possibly not even in the mode that is recommended for use (they tested by manually combining isolated clones which is extremely tedious compared to the proposed use case of adding pooled genetic reagents to cells before a pooled experiment). I found this all quite misleading and frustrating.

We are sorry that we apparently could not provide clear enough descriptions of the method. We have now rewritten the abstract and introduction to clarify how the method works and is best applied. Both sgRNA libraries and pool generation are indeed on the genome-wide in the sense that all targetable genes are covered. The arrayed vpCell collection enables training computer vision models and the rapid generation of additional cell pools beyond the 61 proteins used as a proof of principle in this manuscript. In the revised version, we now added data to show high accuracy of the computational approach to discriminate also much large sets of more than 1,000 clones.

MAJOR

1. In the unperturbed state, >26% of proteins did not match the localization seen for endogenous protein in the Human Protein Atlas, indicating that tagging causes undesirable changes in localization (as also seen in OpenCell project). This needs more emphasis in the paper.

We have now double checked this number and slightly corrected them to 21% not matching the Protein Atlas annotation (annotations remaining the same, but overlap had been inaccurately calculated before due to error in annotation matching in the quantification pipeline). To discuss these discrepancies, we now added a paragraph discussing the limitations of tagging-based approaches that might alter protein localization. We would like to point out that such effects appear not specific to placing full-length fluorophores into intronic sites, as the tagging of protein termini with split mNeonGreen used in the OpenCell project results in near identical numbers of non-overlapping locations with the Human Protein Atlas (25%). While likely tagging-based effects are often responsible, in principle also other factors

like cell type specific differences and antibody issues of alternative approaches could cause these discordant localizations.

2. I don't understand why "proteome-scale" and "genome-wide" is used in the abstract when the majority of validation steps are done on much smaller pools, e.g. 61 proteins or 1,178 proteins (and those were created but not actually tested/used in pooled format); even the sgRNA library they started with covers only 14k genes which is nowhere near genome-wide. This is misleading and should be changed to reflect the size of experiments that were actually validated. It is absolutely not clear from the data presented that thousands of proteins could be distinctly identified under perturbation conditions using the method, as errors in the method would not be expected to scale linearly. IIUC "approximately 2,500 different proteins were tagged with either GFP or mScarlet" so is that the maximum number (and those were not deconvoluted by imaging, but instead by sequencing, so not really helpful either for validating the method). As far as I can tell, 61 proteins are the maximum that are actually deconvoluted from a pool that is created the way that's recommended, by adding a pool of genetic reagents to cells. The heading "large sets of proteins" is above the section with 61 proteins, which doesn't count as large in my view.

We here describe the first genome-wide intron tagging sgRNA libraries, that indeed cover every gene targetable with this approach in the human genome. In Fig. 1 we describe their application to protein tagging, and identify the major factors contributing to successful tagging events. Expression levels are the prime determinant, and we show that application in different cell lines results in different proteins being tagged. The limited overlap between proteins tagged with the genome wide library and the cancer library further shows that the genome-wide library has not been used to saturation, suggestion that editing larger cell pools and sorting to lower fluorescence levels (or potentially using brighter fluorophores) will result in deeper coverage. As for the computational method, to our knowledge the largest alternative approach (Kaufman et al. 2022) uses visual barcodes to multiplex 12 live cell reporters. Compared to that, discriminating 61 proteins computationally and using endogenous proteins as barcodes is indeed a significant advance. Moreover, we have now additionally tested our approach on a set of 1,065 clones in arrayed format, suggesting the algorithm can scale well beyond 61 proteins in a pool. In the revised version, we have rewritten abstract and manuscript to better represent these points, and clearly indicate which aspects are on a genome-scale in the current iteration of the method. We further added a description of the analysis on the maximum number of clones that can be discriminated computationally.

3. It took an unacceptably long time to understand from the abstract what is being tagged and what is being visualized. At first I thought the entire proteome worth of proteins was being directly tagged and visualized, but then an sgRNA library is mentioned which triggered my thinking CRISPR is knocking proteins down and then some other protein is being monitored? And are the two tagged proteins per cell only for barcoding or also for reading out protein dynamics/localization? I couldn't answer these Qs from the abstract alone which is a pretty big problem. Overall, it needs to be more clear what is the (a) perturbation, (b) barcode strategy, (c) readout. Because here proteins could be all 3 of those, it led to confusion about which is feasible to be proteome-scale.

We have now tried to revise the abstract to make it clearer. Given that our manuscript describes the first application of genome-wide intron tagging libraries, we feel it important to mention them here and feel that due to the multiple other application used currently (CRISPRa,

sensors, ...), the mentioning of sgRNAs should not only evoke associations to knock-outs. Due to abstract word limits, we could not include a full description of our method there, but we now included a detailed step-by-step overview of its application at the end of the introduction section.

4. The authors state: "We hypothesized that tagging two different proteins with different colors in each cell might create enough diversity to discriminate hundreds of clones by using the combination of the two localization patterns and intensity values as visual barcodes." But if the whole point is to monitor changes in localization in response to perturbation, then the barcoding fails for exactly those perturbations. They solve this by imaging cells before/after perturbation - this concept should be included in the abstract and much earlier in the main text.

We have now rewritten the abstract to hopefully make the concept clearer, word limits did not allow us to cover all aspects in the abstract. In addition, we now included an overview of the five steps comprising our method in the end of the introduction. Imaging before perturbation makes clone detection more robust, but, because typically a perturbation only changes one channel, most of the time clone recognition is highly accurate also on the perturbed cells.

5. It was also not clear from the abstract whether individual clones are isolated, selected, and pooled, as opposed to more common (and convenient) methods where a pool of genetic reagents are introduced to a cell population.

We have now rewritten the abstract to make it clearer which steps are conducted on cell pools, and that individual clones are used for training the computer vision algorithms.

6. IIUC the experiment that does the former - separating individual clones - was only for validation and not representing how the method should be used. This should be clarified. IIUC "assembled a new cell pool from 41 vpCell clones covering 61 cancer-associated proteins" means that the individual clones were selected and manually pooled together which is not how the method is recommended to be used. This matters because the natural numbers of each clone in a pool will be quite different if cells are grown then combined, vs a pool being created directly from genetic reagents being added.

The random combination of proteins in dual tagged pools requires one step of clone identification for training the computer vision approach. In principle this ground truth could be determined by in situ sequencing of the entire cell pool, as we have done in our previous publication. Here, we used single cell dilution and a PCR-based approach for sgRNA sequencing instead. This has the advantage of generating the large vpCell clone collection, including the opportunity to subselect rapidly sharable smaller clone pools as the one we have described for the cancer drivers. We have now rewritten these aspects of the manuscript to make this point clearer.

7. Accuracy of 97% in identifying clones is great, as is their analysis of manually assembled pools lacking a protein, which was then correctly not found in the pool, but that test was done in a pool of only 41 clones covering 61 proteins.

This description of our data is correct. In the revised version, we have now trained a computer vision model on the non-redundant set of more than 1,000 clones in our collection and also

these can be discriminated with more than 93% accuracy (Extended Data Fig. 6b) and corresponding description in the main text.

8. The analysis that finds some expected gene-compound pairings is not quantitative and thus not convincing. One could imagine finding a few such relationships by chance in any list of a few dozen genes/compounds. It's easy in a high throughput experiment to look at a list of relationships identified and find literature support for them. The authors could either find a set of annotations for the gene-compound relationships and quantify how many were identified vs missed, or they could scramble the list and try to find literature support for the scrambled list (which would need a researcher who is kept in the dark about the goal so they give it a good try).

To validate our method, we have included known chemical degraders of BRD4 and SMARCA4, dBET6 and ACBI1, in our library. For both compounds we had extensively validated their effects in the cell model used in our previous studies (Schick et al. 2021, Reicher et al. 2020). The data on vpCell pools show that we can robustly recover these compounds as hits. For BRD4 degradation, we show specific effects (Fig. 4 c-d), and now also included manual validation of the algorithm (Extended Data Fig. 9). For SMARCA4, two independent clones were included in the cell pool clones included and both showed robust SMARCA4 degradation with ACBI1 which was among the strongest hits picked up by the algorithm (Fig. 4g). We agree that the use of these compounds for validation remains anecdotal, but to our knowledge there is no ground truth data set on compound effects on subcellular localization in HAP1 cells which makes global efforts and estimation of false negative rates challenging. However, we do show that positive hits can be followed up, and validate our novel finding on XPO1 inhibition in multiple models including by genetic mutation and on the endogenous protein by Western blotting.

9. The concept here is unclear “only a small fraction of sgRNAs leads to successful protein tagging, we observed an enrichment of approximately 5% of all sgRNAs in the library” - enrichment relative to what? Does this mean only 5% of sgRNAs (representing 17% of genes targeted) were successful? Or 5% were above some threshold (which was decided how)? This is crucial for understanding the rest of the paper.

The description is correct. As described, enrichment refers to reads in sequencing sgRNAs amplified from successfully tagged cells, as shown in Fig. 1b. The threshold for high-efficiency sgRNAs was chosen based on read distribution in Fig. 1b as covering 70% of all reads in the GFP positive cell pool. The limited overlap with the cancer focused pool (Fig. 2b) shows that this experiment was not saturated, and that possibly sorting larger pools, using a lower threshold for fluorescence intensity in sorting or possibly brighter fluorophores will expand the pool of covered proteins.

10. It should be noted that I got lost at points about whether an experiment was being done a certain way as a validation step or whether it's how it was intended to be used. There may therefore be errors in my above comments! But I will leave them because they point is that an educated reader who cares about this field didn't grasp what was going on and therefore the manuscript needs substantial improvement.

We hope to have been able to clarify these points by rewriting abstract and introduction in the revised manuscript.

MINOR

1. It was really nice to see this comment, it's quite valuable/important: "We took care to randomize clone positions between replicate (imaging plate), in order to prevent possible overfitting to well position during model training."

Thank you for this assessment.

2. The reference to "tagging efficiencies" means more definition. 18.7% of cells exposed to the reagent ended up with a visibly fluorescent GFP tag? or of cells that had an integration event?

Tagging efficiencies refer to visibly fluorescent GFP tagged cells relative to transfected cells.

3. Paper needs a read through for typos, like the two here "effects which we could validation not only in cell pools but also individual tagged cells lines." There were maybe a half dozen others in the main text.

Thank you, we have corrected these and carefully proof-read the manuscript.

4. The authors should elaborate on the scale: a single multi well plate yielded 50 individual cells for most of the 41 clones. That I would think it a bare minimum - is this in each well of the 96 well plate or across the entire plate? The wording makes it unclear. This will allow assessing where is the tradeoff vs just testing each protein individually in arrayed format.

This refers to a single well in a 96-well plate that we used to study one perturbation. We have clarified this point in the revised version and also show complete overviews of the imaged well in Extended Data Figures 7 and 8.

Decision Letter, first revision:

Our ref: NCB-A51759A

4th January 2024

Dear Dr. Kubicek,

I apologize once again for the delay. As mentioned, Unfortunately, Reviewer #3 was unable to review your resubmission. However, we have approached Reviewer #2 to comment on your responses to Reviewer #3's previous concerns.

Thank you for submitting your revised manuscript "Pooled multicolor tagging for visualizing subcellular proteome dynamics" (NCB-A51759A). It has now been seen by the original referees and their comments are below. The reviewers find that the paper has improved in revision, and therefore we'll be happy in principle to publish it in Nature Cell Biology, pending minor revisions to satisfy the referees' final requests and to comply with our editorial and formatting guidelines.

In particular, we would require that you address the remaining points of the referees with further discussion and clarification of your figures such as with a schematic. However, we are now performing detailed checks on your paper and will send you a checklist detailing our editorial and formatting requirements in about a week. Please do not upload the final materials and make any revisions until you receive this additional information from us.

Thank you again for your interest in Nature Cell Biology Please do not hesitate to contact me if you have any questions.

Sincerely,
Daryl

Daryl Jason Verzosa David, PhD

Senior Editor, Nature Cell Biology
Nature Portfolio
Advisory Editor, npj Biological Physics and Mechanics

Heidelberger Platz 3, 14197 Berlin, Germany
Email: daryl.david@nature.com
ORCID: <https://orcid.org/0000-0002-9253-4805>

Reviewer #1 (Remarks to the Author):

The revised manuscript has addressed most of the comments previously raised by this reviewer.

Regarding the characteristics of high-efficiency sgRNAs, the revised manuscript now provides histograms for structuredness and hydrophobicity, which provides a clearer depiction of the statistical significance. Still, the effect size is very small - much smaller than the effect of expression level and early intron targeting. On the other hand, the statement in the main text (lines 146-151) over-emphasizes the statistical significance, which can mislead the audience, e.g. if a choice must be made between targeting a more structured early intron vs a less structured late intron. The effect size is also much smaller than the efficiency variability observed among the sgRNAs, so that prioritizing sgRNAs based on hydrophobicity score or AlphaFold prediction score may only bring in a benefit of a few percentage points, which is practically almost negligible. Therefore, the reviewer requests the manuscript to clarify that hydrophobicity score and AlphaFold prediction score only show weak correlations with targeting efficiency, and that they are not good predictors in the results section in addition to that in the discussion section already.

The reviewer appreciate the new efforts in identifying potential causes of underrepresentation and potential drop-out of certain clone in the pooled library, which is of high importance of any pooled libraries. Nevertheless, the related Extended Fig. 8f is not referenced anywhere in the text, nor has the consequence in pooled library construction been discussed (e.g. limiting the number of clones in a single well considering that the number of cells in a well is capped by the size of the well). The manuscript needs to address this point.

One minor point overlooked in the previous commons: line 212, OpenCell uses both N- and C-terminal tagging instead of just N-terminal tagging.

Reviewer #2 (Remarks to the Author):

The authors have addressed my previous concerns. However, I do find the description of their method is not totally clear to readers. It took a while to fully comprehend how their method works and what the uniqueness is. The manuscript can benefit greatly from an easy-to-understand schematic cartoon that illustrates the mechanism behind their method.

Reviewer #2's comments on authors' responses to the prior concerns of Reviewer #3:

This reviewer recognizes the potential impact and significance of the method in this study. Most comments concern two issues: the clarity of the method and the small scale of the protein libraries tested.

Half of this reviewer's comments (#3, 4, 5, 6, and 9) pertain to the clarity of the method's description. In response, the authors have revised the abstract, introduction, and methods. Despite these efforts, I find that the manuscript still lacks sufficient clarity (similar to my comments). This reviewer has a great point (#3): 'Overall, it needs to be more clear what is the (a) perturbation, (b) barcode strategy, (c) readout.' The authors chose to add some details to the Introduction. I believe this approach is insufficient. If readers cannot comprehend the aims and methods of the study within a brief reading period, the manuscript is unlikely to attract substantial interest. An effective solution could be to incorporate an overview figure (as Fig 1a) that clearly summarizes their whole methodology. Although current Fig. 1a and 1f are included to describe their tagging method, they fail to provide a holistic view of the approach.

Other comments focus on the small scale of the protein library tested, which the reviewer regarded as 'disappointing'. The authors have not expanded the library beyond the initial 61 proteins, though they have increased the number of clones to ~1,000. This modification does not adequately address the concerns raised about the library's scale ('the discovery experiments seem to have been done on quite small-scale protein libraries (61 proteins)'). The authors have adjusted the abstract to remove terms like 'proteome-wide', yet the issue of scale remains.

Overall, I don't feel the authors have fully addressed this reviewer's comments. Clarity could be further improved by revising the text and including visuals to facilitate understanding, which should be doable. I wonder if the authors' decision not to expand beyond 61 proteins was due to limitations inherent in their method. Nevertheless, expanding to a large scale of protein seems a significant effort. It will be an editorial call whether testing a library 61 protein is indeed a significant advance.

Decision Letter, final checks:

Our ref: NCB-A51759A

22nd January 2024

Dear Dr. Kubicek,

Thank you for your patience as we've prepared the guidelines for final submission of your Nature Cell Biology manuscript, "Pooled multicolor tagging for visualizing subcellular proteome dynamics" (NCB-A51759A). Please carefully follow the step-by-step instructions provided in the attached file, and add a response in each row of the table to indicate the changes that you have made. Please also check and comment on any additional marked-up edits we have proposed within the text. Ensuring that each point is addressed will help to ensure that your revised manuscript can be swiftly handed over to our production team.

In recognition of the time and expertise our reviewers provide to Nature Cell Biology's editorial process, we would like to formally acknowledge their contribution to the external peer review of your

manuscript entitled "Pooled multicolor tagging for visualizing subcellular proteome dynamics". For those reviewers who give their assent, we will be publishing their names alongside the published article.

Nature Cell Biology offers a Transparent Peer Review option for new original research manuscripts submitted after December 1st, 2019. As part of this initiative, we encourage our authors to support increased transparency into the peer review process by agreeing to have the reviewer comments, author rebuttal letters, and editorial decision letters published as a Supplementary item. When you submit your final files please clearly state in your cover letter whether or not you would like to participate in this initiative. Please note that failure to state your preference will result in delays in accepting your manuscript for publication.

Cover suggestions

COVER ARTWORK: We welcome submissions of artwork for consideration for our cover. For more information, please see our guide for cover artwork.

Nature Cell Biology has now transitioned to a unified Rights Collection system which will allow our Author Services team to quickly and easily collect the rights and permissions required to publish your work. Approximately 10 days after your paper is formally accepted, you will receive an email in providing you with a link to complete the grant of rights. If your paper is eligible for Open Access, our Author Services team will also be in touch regarding any additional information that may be required to arrange payment for your article.

Please note that *Nature Cell Biology* is a Transformative Journal (TJ). Authors may publish their research with us through the traditional subscription access route or make their paper immediately open access through payment of an article-processing charge (APC). Authors will not be required to make a final decision about access to their article until it has been accepted. Find out more about Transformative Journals

Authors may need to take specific actions to achieve compliance with funder and institutional open access mandates. If your research is supported by a funder that requires immediate open access (e.g. according to Plan S principles) then you should select the gold OA route, and we will direct you to the compliant route where possible. For authors selecting the subscription publication route, the journal's standard licensing terms will need to be accepted, including self-archiving policies. Those licensing terms will supersede any other terms that the author or any third party may assert apply to any version of the manuscript.

Please use the following link for uploading these materials:
[Redacted]

Best regards,

Kendra Donahue
Staff
Nature Cell Biology

On behalf of

Daryl Jason Verzosa David, PhD

Senior Editor, Nature Cell Biology
Nature Portfolio
Advisory Editor, npj Biological Physics and Mechanics

Heidelberger Platz 3, 14197 Berlin, Germany
Email: daryl.david@nature.com
ORCID: <https://orcid.org/0000-0002-9253-4805>

Reviewer #1:

Remarks to the Author:

The revised manuscript has addressed most of the comments previously raised by this reviewer.

Regarding the characteristics of high-efficiency sgRNAs, the revised manuscript now provides histograms for structuredness and hydrophobicity, which provides a clearer depiction of the statistical significance. Still, the effect size is very small - much smaller than the effect of expression level and early intron targeting. On the other hand, the statement in the main text (lines 146-151) over-emphasizes the statistical significance, which can mislead the audience, e.g. if a choice must be made between targeting a more structured early intron vs a less structured late intron. The effect size is also much smaller than the efficiency variability observed among the sgRNAs, so that prioritizing sgRNAs based on hydrophobicity score or AlphaFold prediction score may only bring in a benefit of a few percentage points, which is practically almost negligible. Therefore, the reviewer requests the manuscript to clarify that hydrophobicity score and AlphaFold prediction score only show weak correlations with targeting efficiency, and that they are not good predictors in the results section in addition to that in the discussion section already.

The reviewer appreciate the new efforts in identifying potential causes of underrepresentation and potential drop-out of certain clone in the pooled library, which is of high importance of any pooled libraries. Nevertheless, the related Extended Fig. 8f is not referenced anywhere in the text, nor has the consequence in pooled library construction been discussed (e.g. limiting the number of clones in a

single well considering that the number of cells in a well is capped by the size of the well). The manuscript needs to address this point.

One minor point overlooked in the previous commons: line 212, OpenCell uses both N- and C-terminal tagging instead of just N-terminal tagging.

Reviewer #2:

Remarks to the Author:

The authors have addressed my previous concerns. However, I do find the description of their method is not totally clear to readers. It took a while to fully comprehend how their method works and what the uniqueness is. The manuscript can benefit greatly from an easy-to-understand schematic cartoon that illustrates the mechanism behind their method.

Reviewer #2's comments on authors' responses to the prior concerns of Reviewer #3:

This reviewer recognizes the potential impact and significance of the method in this study. Most comments concern two issues: the clarity of the method and the small scale of the protein libraries tested.

Half of this reviewer's comments (#3, 4, 5, 6, and 9) pertain to the clarity of the method's description. In response, the authors have revised the abstract, introduction, and methods. Despite these efforts, I find that the manuscript still lacks sufficient clarity (similar to my comments). This reviewer has a great point (#3): 'Overall, it needs to be more clear what is the (a) perturbation, (b) barcode strategy, (c) readout.' The authors chose to add some details to the Introduction. I believe this approach is insufficient. If readers cannot comprehend the aims and methods of the study within a brief reading period, the manuscript is unlikely to attract substantial interest. An effective solution could be to incorporate an overview figure (as Fig 1a) that clearly summarizes their whole methodology. Although current Fig. 1a and 1f are included to describe their tagging method, they fail to provide a holistic view of the approach.

Other comments focus on the small scale of the protein library tested, which the reviewer regarded as 'disappointing'. The authors have not expanded the library beyond the initial 61 proteins, though they have increased the number of clones to ~1,000. This modification does not adequately address the concerns raised about the library's scale ('the discovery experiments seem to have been done on quite small-scale protein libraries (61 proteins)'). The authors have adjusted the abstract to remove terms like 'proteome-wide', yet the issue of scale remains.

Overall, I don't feel the authors have fully addressed this reviewer's comments. Clarity could be further improved by revising the text and including visuals to facilitate understanding, which should be doable. I wonder if the authors' decision not to expand beyond 61 proteins was due to limitations inherent in their method. Nevertheless, expanding to a large scale of protein seems a significant effort. It will be an editorial call whether testing a library 61 protein is indeed a significant advance.

Author Rebuttal, second revision:

RESPONSE TO REFEREES

Nature Cell Biology submission NCB-A51759A

Pooled multicolor tagging for visualizing subcellular protein dynamics

Andreas Reicher^{1*}, Jiří Reiniš^{1*}, Maria Ciobanu¹, Pavel Růžička¹, Monika Malik¹, Marton Siklos¹, Victoria Kartysh¹, Tatjana Tomek¹, Anna Koren¹, André F. Rendeiro¹, Stefan Kubicek¹

¹CeMM Research Center for Molecular Medicine of the Austrian Academy of Sciences, Lazarettgasse 14, 1090 Vienna, Austria.

*these authors contributed equally

We thank all reviewers and the editors for these further constructive comments and suggestions. In this revision, we have not addressed all remaining comments and reformatted the manuscript by editorial requirements:

The major changes and additions in the current version are the following:

- We added an overview figure (new Figure 1a) as an outline of the approach and to indicate the scale in terms of number of clones and tagged proteins at which each step is conducted.
- We more clearly indicated that although statistically significant, tagging site hydrophobicity and structuredness only have a weak impact on tagging efficiency.
- We changed the title to emphasize 'protein dynamics' over proteome.
- We reformatted the supplementary table of sgRNA libraries to more clearly indicate sgRNAs that target introns that are common to overlapping genes, and removed the resulting clones from the analysis.
- We rearranged the figures to have only 10 Extended Data figures and additional Supplementary Materials including an added sorting strategy.
- We addressed all requests regarding source data and statistics.
- We obtained a doi for the GitHub repository and made it and the vpcells.cemm.at webpage publicly available.

Please find our detailed point-by-point responses below.

Reviewer #1:

Remarks to the Author:

The revised manuscript has addressed most of the comments previously raised by this reviewer.

We thank the reviewer for these positive comments.

Regarding the characteristics of high-efficiency sgRNAs, the revised manuscript now provides histograms for structuredness and hydrophobicity, which provides a clearer depiction of the statistical significance. Still, the effect size is very small - much smaller than the effect of expression level and early intron targeting. On the other hand, the statement in the main text (lines 146-151) over-emphasizes the statistical significance, which can mislead the audience, e.g. if a choice must be made between targeting a more structured early intron vs a less structured late intron. The effect size is also much smaller than the efficiency variability observed among the sgRNAs, so that prioritizing sgRNAs based on hydrophobicity score or AlphaFold prediction score may only bring in a benefit of a few percentage points, which is practically almost negligible. Therefore, the reviewer requests the manuscript to clarify that hydrophobicity score and AlphaFold prediction score only show weak correlations with targeting efficiency, and that they are not good predictors in the results section in addition to that in the discussion section already.

We have now further toned down the text regarding preferential tagging at hydrophilic and unstructured sites. Both in the results section and the discussion, we now emphasize that while statistically significant, these only weakly impact tagging efficiency.

The reviewer appreciate the new efforts in identifying potential causes of underrepresentation and potential drop-out of certain clone in the pooled library, which is of high importance of any pooled libraries. Nevertheless, the related Extended Fig. 8f is not referenced anywhere in the text, nor has the consequence in pooled library construction been discussed (e.g. limiting the number of clones in a single well considering that the number of cells in a well is capped by the size of the well). The manuscript needs to address this point.

We added a reference to this panel (now Extended Data Fig. 6h) and mentioned the importance of establishing pools with a balanced representation of clones in the discussion.

One minor point overlooked in the previous commons: line 212, OpenCell uses both N- and C-terminal tagging instead of just N-terminal tagging.

We have now corrected this to 'N- or C-terminally tagged proteins in the OpenCell collection'.

Reviewer #2:

Remarks to the Author:

The authors have addressed my previous concerns. However, I do find the description of their method is not totally clear to readers. It took a while to fully comprehend how their method works and what the uniqueness is. The manuscript can benefit greatly from an easy-to-understand schematic cartoon that illustrates the mechanism behind their method.

Thank you for this suggestion. We have now added a new figure (Fig. 1a) to give an overview of the approach, including details on the scales at which each step is conducted in the current study.

Reviewer #2's comments on authors' responses to the prior concerns of Reviewer #3:

This reviewer recognizes the potential impact and significance of the method in this study. Most comments concern two issues: the clarity of the method and the small scale of the protein libraries tested.

Half of this reviewer's comments (#3, 4, 5, 6, and 9) pertain to the clarity of the method's description. In response, the authors have revised the abstract, introduction, and methods. Despite these efforts, I find that the manuscript still lacks sufficient clarity (similar to my comments). This reviewer has a great point (#3): 'Overall, it needs to be more clear what is the (a) perturbation, (b) barcode strategy, (c) readout.' The authors chose to add some details to the Introduction. I believe this approach is insufficient. If readers cannot comprehend the aims and methods of the study within a brief reading period, the manuscript is unlikely to attract substantial interest. An effective solution could be to incorporate an overview figure (as Fig 1a) that clearly summarizes their whole methodology. Although current Fig. 1a and 1f are included to describe their tagging method, they fail to provide a holistic view of the approach.

We have now followed this suggestion and added Fig. 1a to provide an overview of the study and to clearly indicate what constitutes perturbations, barcodes and readouts, as well as the scale at which each step is conducted in the current manuscript.

Other comments focus on the small scale of the protein library tested, which the reviewer regarded as 'disappointing'. The authors have not expanded the library beyond the initial 61 proteins, though they have increased the number of clones to ~1,000. This modification does not adequately address the concerns raised about the library's scale ('the discovery experiments seem to have been done on quite small-scale protein libraries (61 proteins)'). The authors have adjusted the abstract to remove terms like 'proteome-wide', yet the issue of scale remains.

As indicated in our earlier response, we conduct different steps at different scales. The sgRNAs constructed are indeed genome-scale as they comprehensively target available introns in the different reading frames. Using these libraries in a pooled format under the non-saturating sorting conditions results in tagging of approximately 2,500 proteins, and we isolate clonal cell lines for more than 1,100 proteins out of these pools. In all of these, we analyze localization in steady state, and thereby contribute a complementary resource of comparable scale to OpenCell, the biggest systematic tagging effort so far. We then monitor perturbation-induced changes in a cell pool of 61 proteins as a proof of concept, but in principle nothing limits the approach from being repeated throughout the entire collection. Even at the scale of 61 proteins, this is to our knowledge by far the biggest effort for pooled monitoring of protein localization in response to perturbations. To emphasize the different scales, we have now

added Fig. 1a including the detailed number of clones and tagged proteins involved at each step. We have further changed the title to 'Pooled multicolor tagging for visualizing subcellular protein dynamics'.

Overall, I don't feel the authors have fully addressed this reviewer's comments. Clarity could be further improved by revising the text and including visuals to facilitate understanding, which should be doable. I wonder if the authors' decision not to expand beyond 61 proteins was due to limitations inherent in their method. Nevertheless, expanding to a large scale of protein seems a significant effort. It will be an editorial call whether testing a library 61 protein is indeed a significant advance.

As suggested, we have now included the new Fig. 1a to visually depict the scale at which each step is conducted. Nothing inherent in the technology limits its application to larger protein sets. For very large protein sets of thousands of clones, imaging time becomes limiting for live-cell approaches, so that these are better suited for screening in multiple small pools of ~50 clones or to readouts on fixed cells clone identification by in situ sequencing. The main reason not to expand to additional cell pools was not due to inherent limitation of the method, but rather due to the fact that we had already identified more active compounds from this cell pool than we could functionally follow-up. For a first proof-of-concept study we deemed these validation experiments more important than conducting screens with larger cell pools.

Final Decision Letter:

Dear Dr Kubicek,

I am pleased to inform you that your manuscript, "Pooled multicolor tagging for visualizing subcellular protein dynamics", has now been accepted for publication in Nature Cell Biology.

You may wish to make your media relations office aware of your accepted publication, in case they consider it appropriate to organize some internal or external publicity. Once your paper has been scheduled you will receive an email confirming the publication details. This is normally 3-4 working days in advance of publication. If you need additional notice of the date and time of publication, please let the production team know when you receive the proof of your article to ensure there is sufficient time to coordinate. Further information on our embargo policies can be found here:

<https://www.nature.com/authors/policies/embargo.html>

Please note that *Nature Cell Biology* is a Transformative Journal (TJ). Authors may publish their research with us through the traditional subscription access route or make their paper immediately open access through payment of an article-processing charge (APC). Authors will not be required to make a final decision about access to their article until it has been accepted. Find out more about Transformative Journals

If you have not already done so, we strongly recommend that you upload the step-by-step protocols used in this manuscript to the Protocol Exchange (www.nature.com/protocolexchange), an open online resource established by Nature Protocols that allows researchers to share their detailed experimental know-how. All uploaded protocols are made freely available, assigned DOIs for ease of citation and are fully searchable through nature.com. Protocols and Nature Portfolio journal papers in which they are used can be linked to one another, and this link is clearly and prominently visible in the online versions of both papers. Authors who performed the specific experiments can act as primary authors for the Protocol as they will be best placed to share the methodology details, but the Corresponding Author of the present research paper should be included as one of the authors. By uploading your Protocols to Protocol Exchange, you are enabling researchers to more readily reproduce or adapt the methodology you use, as well as increasing the visibility of your protocols and papers. You can also establish a dedicated page to collect your lab Protocols. Further information can be found at www.nature.com/protocolexchange/about

With kind regards,

Daryl

Daryl Jason Verzosa David, PhD

Senior Editor, Nature Cell Biology

Nature Portfolio

Advisory Editor, npj Biological Physics and Mechanics

Heidelberger Platz 3, 14197 Berlin, Germany

Email: daryl.david@nature.com

ORCID: <https://orcid.org/0000-0002-9253-4805>
